# In vitro modeling of the human dopaminergic system using spatially arranged ventral midbrain–striatum–cortex assembloids

**A list of authors and their affiliations appears at the end of the paper**

Ventral midbrain dopaminergic neurons project to the striatum as well as the cortex and are involved in movement control and reward-related cognition. In Parkinson's disease, nigrostriatal midbrain dopaminergic neurons degenerate and cause typical Parkinson's disease motor-related impairments, while the dysfunction of mesocorticolimbic midbrain dopaminergic neurons is implicated in addiction and neuropsychiatric disorders. Study of the development and selective neurodegeneration of the human dopaminergic system, however, has been limited due to the lack of an appropriate model and access to human material. Here, we have developed a human in vitro model that recapitulates key aspects of dopaminergic innervation of the striatum and cortex. These spatially arranged ventral midbrain–striatum–cortical organoids (MISCOs) can be used to study dopaminergic neuron maturation, innervation and function with implications for cell therapy and addiction research. We detail protocols for growing ventral midbrain, striatal and cortical organoids and describe how they fuse in a linear manner when placed in custom embedding molds. We report the formation of functional long-range dopaminergic connections to striatal and cortical tissues in MISCOs, and show that injected, ventral midbrain-patterned progenitors can mature and innervate the tissue. Using these assembloids, we examine dopaminergic circuit perturbations and show that chronic cocaine treatment causes long-lasting morphological, functional and transcriptional changes that persist upon drug withdrawal. Thus, our method opens new avenues to investigate human dopaminergic cell transplantation and circuitry reconstruction as well as the effect of drugs on the human dopaminergic system.

Most of the ~500,000 dopaminergic neurons are located in the ventral midbrain of the human brain. Midbrain dopaminergic neuron signaling is neuromodulatory and controls motivation, reinforcement, motor control, voluntary movement, arousal and reward[1]. Midbrain dopaminergic neurons predominantly project anteriorly into the striatum (the

nigrostriatal pathway) and cortex (the mesocorticolimbic pathway)[2–5] (Fig. 1a). Midbrain dopaminergic neurons (mDA) are divided into two main populations: A9 midbrain dopaminergic neurons in the substantia nigra pars compacta (SNc), which form the nigrostriatal pathway, and A10 midbrain dopaminergic neurons in the ventral tegmental area

✉e-mail: Juergen.knoblich@imba.oeaw.ac.at

(VTA), which form the mesocorticolimbic pathway[6]. Degeneration of midbrain dopaminergic neurons, especially A9, results in one of the most common neurodegenerative disorders, Parkinson's disease[7].

Although the rodent dopaminergic system has been extensively studied[8,9], the development and the neurodegeneration of the human dopaminergic system are poorly understood. This is partly due to the limited availability of human tissue and the lack of appropriate human model systems, but also due to the extended timespan of human neurogenesis[10]. Importantly, some aspects of Parkinson's disease cannot be adequately modeled in rodents because they do not naturally develop Parkinson's disease, which makes artificial depletion of dopaminergic signaling necessary[11,12]. Thus, so far we do not fully understand how the human dopaminergic system develops, nor how genetic and environmental factors can alter its development and lead to midbrain dopaminergic neuron degeneration[1,13].

A novel approach in the context of Parkinson's disease is dopaminergic cell replacement therapy, in which depleted midbrain dopaminergic neurons are replaced with fetal or human pluripotent stem cell (hPSC)-derived midbrain dopaminergic progenitors[14]. Understanding how these grafts innervate and functionally integrate into existing circuits would thus benefit current and future clinical studies focused on midbrain dopaminergic neuron replacement therapy.

The dopaminergic mesocorticolimbic projections are the key reward pathway in the brain and have a key role in addiction, given that the dopaminergic system is the direct target of highly addictive drugs such as cocaine[15]. The effect of addictive drugs on human brain development is generally poorly understood and is mostly described in postnatal behavioral studies, as the extended developmental time window in humans renders comparison with in vivo model systems difficult[16]. Thus, the generation of appropriate models to study the human dopaminergic system might lead to improved treatment options for dopamine-related diseases.

Human midbrain organoid culture has become a widely adopted system for long-term maintenance of functionally mature midbrain dopaminergic neurons from stem cells or fetal tissue[17–20]. Brain organoid fusions (assembloids) have been previously reported and display interactions between different brain regions[21–23]. However, to our knowledge there is no integrated system that can be used to investigate dopaminergic circuit formation and function in a human context.

Here, we produced ventral midbrain, striatal and cortical organoids from hPSCs to generate ventral midbrain–striatum–cortical organoid assembloids (MISCOs) by positioning the organoids linearly in their anterior–posterior direction using custom polydimethylsiloxane (PDMS) embedding molds. MISCOs develop functionally active, dopamine-releasing axonal long-range projections from ventral midbrain into striatal and cortical tissue. MISCOs can be effectively applied in multiple ways: First, we show that dopaminergic progenitors used in Parkinson's disease clinical studies for cell replacement can be injected into MISCOs to study innervation and maturation properties in a human environment. Second, we show that artificially elevated dopamine exposure after the addition of the dopamine reuptake inhibitor cocaine induces morphological, functional and transcriptional changes that persist after withdrawal, indicating long-term neuronal circuit related changes by overstimulation of dopaminergic signaling during development.

## Results

### Generation of ventral midbrain organoids

Midbrain dopaminergic neurons originate from the ventral midbrain floor plate (FOXA2+), the formation of which relies on precise signaling of both ventralizing SHH and caudalizing Wnt factors[17,24–26]. To generate ventral midbrain organoids we modified existing ventral midbrain protocols and substituted the commonly used ventralization morphogen SHH (sonic hedgehog) with the small molecule and SHH-activator smoothened agonist (SAG) (Fig. 1b,c). When dose-curving organoids with SAG from day 4 to day 11, we found that treatment with 300 nM SAG, together with dual SMAD inhibition and Wnt activation, was sufficient to form organoids that had maximal FOXA2 expression levels by day 20 (Fig. 1b,d and Extended Data Fig. 1a–d). Organoids were positive for tyrosine hydroxylase (TH), a key enzyme for dopamine synthesis, and FOXA2 on day 44 (Fig. 1e,e′ and Extended Data Fig. 1e), while being negative for the forebrain marker FOXG1(ref. 27) (Extended Data Fig. 1f). TH-positive neurons widely expressed the midbrain dopaminergic markers LMX1A and EN1, indicating differentiation into midbrain dopaminergic neurons (Fig. 1f,g and Extended Data Fig. 1e). This protocol could be reproduced using three human induced pluripotent stem (iPS) cell and three human embryonic stem (hES) cell lines, highlighting its robustness (Extended Data Fig. 1g–j). Thus, our protocol successfully and robustly generates ventral midbrain organoids containing midbrain dopaminergic neurons.

### Generation of striatal organoids

The predominant contributor to striatal neurogenesis is the lateral ganglionic eminence (LGE) neuroepithelium[28]. Medial ganglionic eminence (MGE) identity in organoids was previously induced by Wnt pathway inhibition and SHH pathway activation[23,29]. Given that the LGE is situated less ventrally than the MGE (Fig. 1h), we generated a SAG dose–response curve in the presence of the Wnt inhibitor IWP-2 in an otherwise growth-factor free neural induction medium[30]. We found that only 10 nM SAG from day 0 until day 6 was sufficient to form organoids

---

**Fig. 1 | Generation of patterning protocols for ventral midbrain and striatal organoids. a**, The dopaminergic system during development (left) and in the adult human brain (right). The axons of midbrain dopaminergic (mDA) neurons innervating striatal and cortical tissue are highlighted in blue. **b**, Patterning timings and factors used for the generation of ventral midbrain, striatal and cortical organoids. **c**, Schematic diagram of ventral midbrain dopaminergic neurogenesis including key morphogenic gradients. **d**, Ventral midbrain patterned organoids treated with 300 nM SAG express the floor plate marker FOXA2 on day 20 (representative image, similar results in n = 20 of 20 organoids of five cell lines). **e**, 44-day-old ventral midbrain organoids show clusters of TH-positive and FOXA2-positive mDA neurons (representative image, similar results in n = 8 of 8 organoids of two batches and two cell lines). Yellow box: magnified view in e′. **f,g**, 80-day-old TH-positive neurons express the mDA neuronal markers EN1 and LMX1A. **h**, Schematic diagram of striatal neurogenesis including key morphogenic gradients. **i,j**, Day 30 striatal patterned organoids show GSX2-positive rosettes (representative images, similar results in n = 28 of 32 organoids of six cell lines with one cell line (178-5) failing to produce GSX2-positive rosettes (Extended Data Fig. 4b,c)) and produce DLX5-positive (**i**) and CTIP2-positive (**j**) neurons (representative image, similar results in n = 4 of 4

organoids). Yellow box in i: magnified view in i′. **k**, Day 60 striatal organoids still contain GSX2-positive LGE progenitors and are broadly composed of CTIP2-positive neurons (representative image, similar results in n = 9 of 9 organoids of two batches). **l**, Striatal patterned organoids produce DARPP32-positive clusters of neurons (representative image, similar results in 35 of 38 organoids of six cell lines with one iPS cell line (178-5) being mostly negative for DARPP32 (Extended Data Fig. 4c)). **m,n**, DARPP32-positive neurons broadly express the GABAergic marker GAD1 (n = 4 of 4 organoids of two batches). **o,p**, DARPP32-positive cells express the striatal markers CTIP2 and FOXP1 (n = 6 of 6 organoids of two batches). White arrows indicate triple-positive cells. **q**, PCA of the top 500 variable genes of RNA-seq of individual ventral midbrain, striatal and cortical, patterned organoids on day 60 (n = 3–4 organoids per group). **r**, Genes with the highest loading on PC1 (left) and PC2 (right), by PCA of the top 500 variable genes. **s**, VoxHunt spatial similarity mapping of bulk RNA-seq data of ventral midbrain, striatal and cortical organoids to E13.5 Allen Developing Mouse Brain Atlas data with sections colored by scaled expression similarity scores. Data given as mean ± s.d. Scale bars: **d,e,i,k,l**, 500 μm; **m**, 200 μm; **e′,i′**, 100 μm; **j**, 50 μm; **f,g,o**, 20 μm.

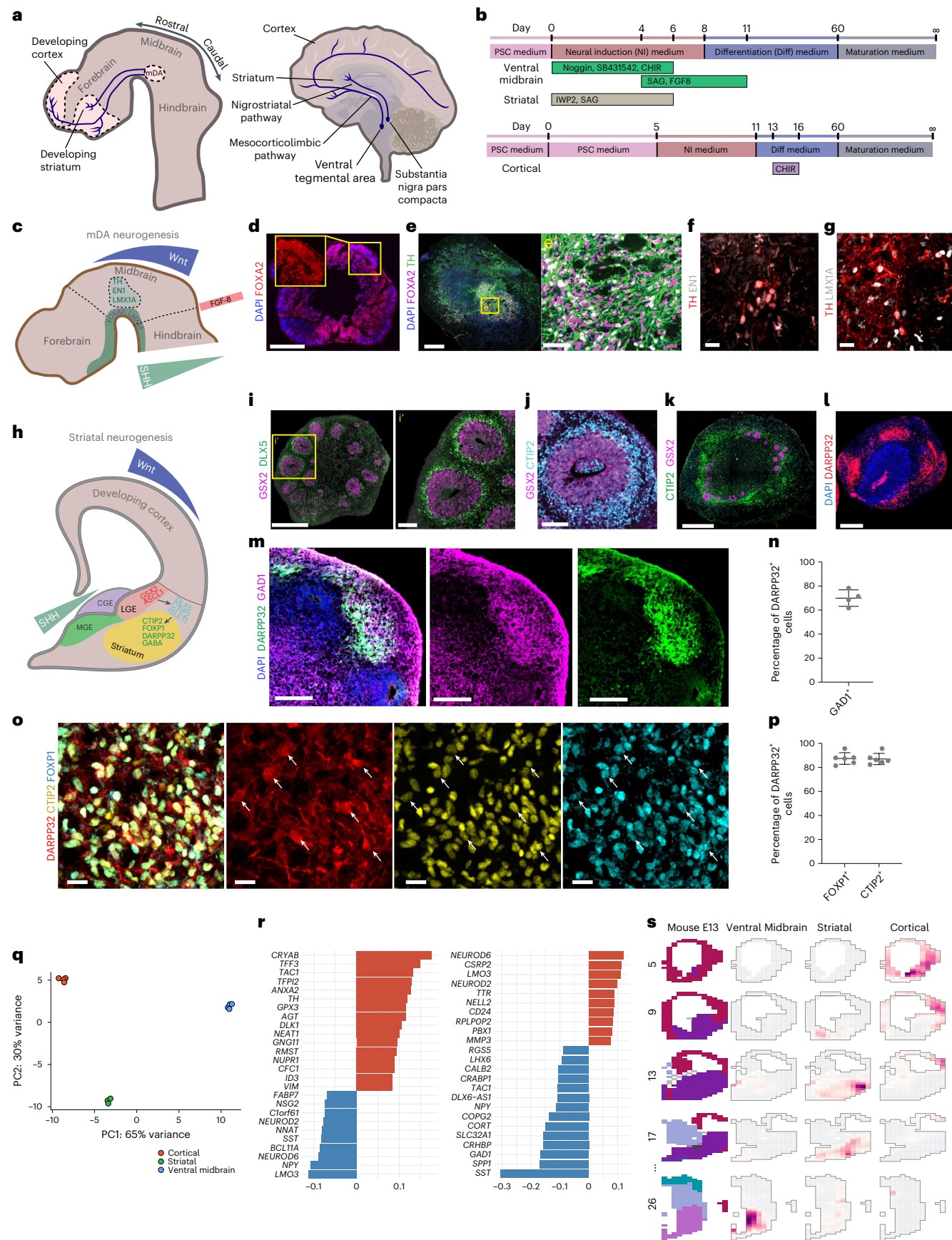

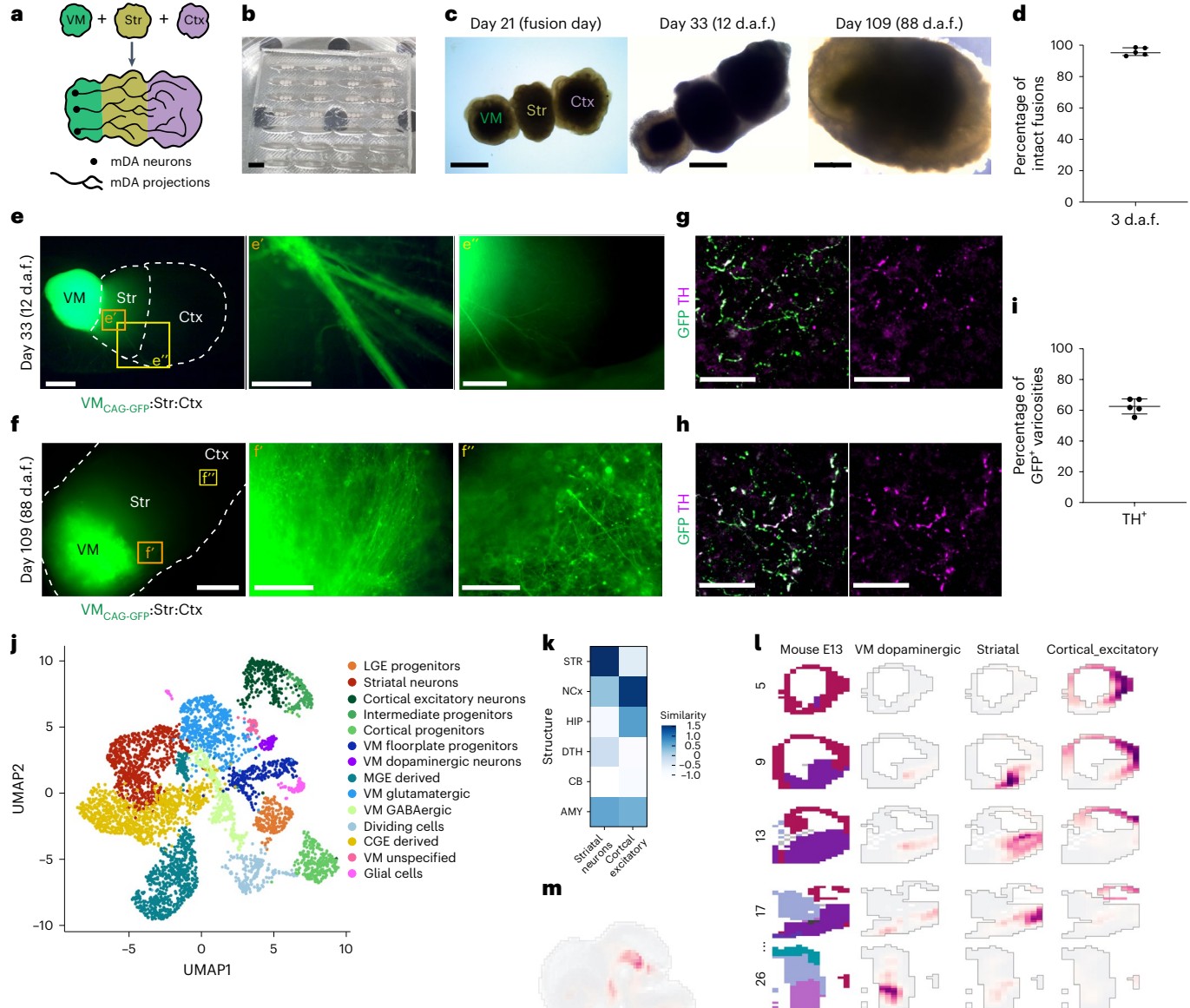

**Fig. 2 | PDMS mold-assisted spatially arranged fusion of ventral midbrain, striatal and cortical organoids enables the observation of dopaminergic innervation into striatal and cortical tissues. a**, Schematic diagram of the generation of linear anterior–posterior-positioned fused organoids from ventral midbrain (VM), striatal (Str) and cortical (Ctx) organoids. **b**, PDMS embedding molds with juxtaposed VM–striatal–cortical organoids. **c**, Triple-fused organoids readily grow together and form a homogeneous tissue. d.a.f., days after fusion. (Representative images; similar results in $n$ = 3–6 organoids per timepoint.) **d**, Quantification of fusion efficiency of five batches and a total of 365 fusions. A mean of 96% (±2.4% s.d.) of fusions remained intact after the fusion procedure across multiple batches. **e**, 12 days after fusion (day 33 of organoid age), the first neurite outgrowth can be observed in Ctx_{WT}-Str_{WT}-VM_{CAG-GFP}-labeled MISCOs (representative images, similar results in $n$ = 6 of 6 organoids of two batches). Orange and yellow boxes: magnified view in **e′** and **e″**. **f**, On day 109 the striatal (**f′**) and cortical (**f″**) tissue are strongly innervated by the VM tissue (representative images; similar results in $n$ = 3 of 3 organoids). Orange and yellow boxes: magnified view in **f′** and **f″**. **g**,**h**, GFP-positive neurites in cortical (**g**) and striatal (**h**) tissue were broadly double positive for the dopaminergic marker TH on day 90 (representative

images; similar results in $n$ = 3 organoids per region in cryosectioned tissue, 4–5 organoids of an independent batch in 3D tissue-cleared batch). **i**, Fraction of dopaminergic varicosities: 63% of all GFP-positive axons in the forebrain were of dopaminergic identity ($n$ = 5 organoids; s.d., 4.9%). **j**, scRNA-seq of day 60 MISCOs showing that all major populations of the dopaminergic circuit (mDA, striatal and cortical neurons) were present, together with clusters of cortical and LGE progenitors, MGE- and CGE-derived cells and VM GABAergic and glutamatergic neurons ($n$ = 3 pooled MISCOs of one batch). **k**, Correlation of the cortical excitatory neuron and striatal neuron cluster with the BrainSpan dataset of the developing human brain (postconception weeks 20–25). AMY, amygdala; CB, cerebellum; DTH, dorsal thalamus; HIP, hippocampus; NCx, neocortex; STR, striatum. **l**, VoxHunt spatial similarity mapping of the cortical excitatory neuron, striatal and VM dopaminergic neuronal clusters onto E13.5 Allen Developing Mouse Brain Atlas data with sections colored by scaled expression similarity scores. **m**, VoxHunt spatial similarity mapping of the VM dopaminergic cluster onto E13.5 Allen Developing Mouse Brain Atlas data (sagital), colored by scaled expression similarity scores. Data given as mean ± s.d. Scale bars: **c** (right),**f**, 1 mm; **b**,**c** (left, middle), **e**, 500 μm; **e″**, 250 μm; **f′**, 100 μm; **e′**,**f″**, 50 μm; **g**,**h**, 20 μm.

in which neuroepithelial rosettes predominantly expressed the LGE marker GSX2, but were negative for the cortical progenitor marker PAX6 (Fig. 1i, Extended Data Fig. 1b–d and Extended Data Fig. 2a–d). Rosettes expressed the proneural transcription factor ASCL1 (Extended Data

Fig. 2e) and produced postmitotic LGE-derived neurons that expressed DLX5, BCL11B (also known as CTIP2) and ISLA1A (ISLET1) on day 30 (Fig. 1i,j and Extended Data Fig. 2f). Sixty-day-old striatal organoids still contained GSX2-positive LGE neuroepithelium and large clusters

of cells expressing the striatal neuron marker DARPP32 (also known as PPP1R1B) (Fig. 1k,l). DARPP32-positive clusters were mainly GAD1 positive, indicating GABA (γ-aminobutyric acid)-ergic identity (Fig. 1m,n), and expressed CTIP2 and FOXP1, further evidence of a striatal neuron identity (Fig. 1o,p). We tested our protocol using a DLXi5/6-GFP cell line[31] and found that DARPP32-positive cells expressed green fluorescent protein (GFP), albeit at lower levels than DARPP32⁻ GFP⁺ forebrain interneurons, suggesting a ganglionic eminence origin[28] (Extended Data Fig. 2g,h). Striatal organoids contained scattered NKX2-1-positive cells, indicating the emergence of MGE-derived striatal interneurons[28] (Extended Data Fig. 2i). Striatal organoids were positive for FOXG1, a forebrain marker that is also expressed in cortical, but not ventral midbrain tissues (Extended Data Figs. 1f,2j). Notably, striatal organoids were negative for the cortical marker TBR1 (Extended Data Fig. 2k). We additionally confirmed the capability of this protocol to produce striatal neurons on three human induced PSC and three hES cell lines and found consistent formation and induction of striatal neurogenesis in five out of six cell lines (Extended Data Fig. 3). Our data show that our protocol successfully and robustly generates striatal organoids.

### RNA sequencing confirms ventral midbrain, striatal and cortical tissue identity

To validate our patterning protocols, we performed RNA sequencing (RNA-seq) at day 60. Cortical organoids were grown according to recently published protocols[32–34]. Notably, we found significant size differences between the three different protocols, with the cortical organoid protocol resulting in the largest, and the ventral midbrain organoid protocol resulting in the smallest organoids (Extended Data Fig. 1c,d). Principal component analysis (PCA) clustered individual organoids according to their respective patterning protocols (Fig. 1q). The strongest contribution to the first principal component (PC1) was made by anterior–posterior identity genes. Among these were *TH* and *RMST* for posterior (ventral) midbrain identity, as well as *NEUROD2*, *NEUROD6* and *BCL11A* for anterior (forebrain) identity. PC2 mostly indicated forebrain dorsoventral patterning, and included the dorsal markers *NEUROD6* and *NEUROD2*, as well as classic ventral forebrain markers *GAD1*, *DLX6-AS1* and LHX6 (Fig. 1r). In addition, comparison with previously published enrichment lists for developing ventral midbrain, striatal and cortical tissue indicated that the striatal- and cortical-enriched genes were enriched in the respective organoids[28] (Extended Data Fig. 4a,b,e,f). Comparison of cortical and ventral midbrain organoids also indicated enrichment of ventral midbrain and cortical markers in the respective organoids (Extended Data Fig. 4c,d,g,h). To further confirm correct patterning, we used VoxHunt with the E13.5 Allen Developing Mouse Brain Atlas dataset as a reference and found a strong correlation between the patterned organoids and their corresponding tissues in vivo[35] (Fig.

1s). Together, these data demonstrate that our protocols successfully generate ventral midbrain, striatal and cortical organoids.

### PDMS mold assists generation of triple fusion organoids

To recreate aspects of the dopaminergic system, we linearly fused ventral midbrain, striatal and cortical organoids, mimicking their anterior–posterior positioning in the brain (Figs. 1a, 2a and Extended Data Fig. 5a). To facilitate fusion, we designed PDMS embedding molds that fit day 20–25 organoids (Fig. 2b and Extended Data Fig. 5a–g). A PLA (polylactic acid) extrusion three-dimensional (3D) printed mold was used as a negative to cast the PDMS positive mold. Organoids were then positioned in the molds and attached to each other with Matrigel. The linear assembloids (MISCOs) were then rinsed out of the mold and cultured on an orbital shaker (Fig. 2c and Extended Data Fig. 5a). In consecutive batches, we achieved 96% (±2.4% s.d.) fusion efficiency after 3 days (Fig. 2d), after which we generally did not see tissue separation.

To determine ventral midbrain innervation into striatal and cortical tissue we used a constitutively GFP-expressing cell line (CAG-GFP) for ventral midbrain. Innervation was readily observable on day 33, only 12 days after fusion[23] (Fig. 2e). The extent of innervation rose drastically by day 109 (Fig. 2f), and approximately 60% of all GFP-positive projections were positive for the dopaminergic marker TH (Fig. 2g–i). We also confirmed dopaminergic projections into striatal (DARPP32⁺) and cortical (TBR1⁺) tissues (Extended Data Fig. 5h). Notably, the mature midbrain dopaminergic neuron marker dopamine transporter (SLC6A3/DAT) was prominent in midbrain dopaminergic neurons in the ventral midbrain, but also in their axons extending into striatal and cortical tissues (Extended Data Fig. 5i). In striatal neurons, dopamine receptor types 1 and 2 (DRD1 and DRD2) suggested the presence of distinct striatal neuron subtypes (Extended Data Fig. 5j). We found that the majority of TH-positive neurons expressed the midbrain dopaminergic subtype markers KCNJ6/GIRK2 and CALB1 as well as the A10 VTA midbrain dopaminergic specification marker OTX2. Furthermore, ~10% of midbrain dopaminergic neurons expressed SOX6, a marker associated with the A9 SNc midbrain dopaminergic specification. We also found midbrain dopaminergic neurons expressing the subtype markers ALDH1A1 and GABA, showing that MISCOs reflect the diversity of midbrain dopaminergic neuronal subtypes found in the human brain[36,37] (Extended Data Fig. 5k,l).

To confirm cellular identities, we performed single-cell RNA sequencing (scRNA-seq) of day 60 MISCOs. Manually dissected individual regions were barcoded using MULTI-seq and then sequenced[38] (Extended Data Fig. 6a). Although impurities from neighboring regions were present, a bias towards region-specific clusters was clearly detected (Extended Data Fig. 6b,c). Supervised clustering

---

**Fig. 3 | MISCOs form structural features of maturation. a**, 2Eci-cleared 60-day-old MISCOs with CAG-GFP expression in the VM tissue and immunolabeling for TH in gray (representative images; similar results in *n* = 7 of 7 organoids). **b**, Striatal tissue (left) had stronger innervation than cortical tissue (right). **c**, Quantification of peak fluorescence of 2Eci-cleared organoid recordings in striatal and cortical tissue (*n* = 5 organoids, *P* = 0.0125, unpaired two-sided *t*-test). **d**, Dopaminergic (TH⁺) axons form axon bundles that project into the forebrain tissue. **e**, TH-positive axons generally avoid neurogenic regions (representative images; similar results in *n* = 17 of 17 rosettes of 8 organoids with ≤1 TH⁺ axon per neural rosette). V, ventricle; VZ, ventricular zone. **f**–**h**, TH-positive axons in the striatum (**f**) and cortex (not shown) structurally mature over time, as indicated by an increase in varicosities of TH-positive axons between day 40 and 120 in the striatum (**g**) and cortex (**h**) (*n* = 5–7 organoids per timepoint). Statistical significance was tested using one-way ANOVA followed by Tukey's multiple comparison test (****$P_{adj}$ < 0.0001, **$P_{adj}$ = 0.0086, NS (not signficant), $P_{adj}$ = 0.0877 (**g**), ***$P_{adj}$ = 0.0002, ****$P_{adj}$ < 0.0001, *$P_{adj}$ = 0.0109 (**h**)). **i**,**j**, Organoids transduced with an AAV expressing Syn-Arch1-GFP, a membrane-bound variant of GFP, enable observation of dendrites (yellow arrows, magnified

view: Extended Data Fig. 7f–h) and axonal boutons (white arrows, magnified view: Extended Data Fig. 7f–h). Stereotypic morphologies of pyramidal neurons in cortical tissues (**j**, magnified view of a dendritic tree) and neurons resembling stereotypic multipolar morphology in the striatal tissue as well as heterogeneous morphologies in VM organoids were found (representative images; similar results in *n* = 12–20 neurons per region of 150-day-old organoids). **k**, Schematic diagram of monosynaptic rabies virus (RV) tracing for retrograde tracing of VM axons. Region-restricted helper virus (HV) transduction was achieved by injection of small volumes into the forebrain tissue. **l**, TVA-G helper virus (GFP⁺) transduction was locally constrained into the forebrain. Rabies virus signal (Crimson⁺) was predominantly found surrounding the injection site, but also in the periphery of the organoids (representative images; similar results in *n* = 5 of 5 organoids). Magnified views (yellow boxes) are given in Extended Data Fig. 7i (e7i) and 7j (e7j). **m**, In the VM, rabies virus broadly labeled neurons, among others dopaminergic (TH⁺) neurons (yellow arrows) (representative images; similar results in *n* = 5 of 5 organoids). Data given as mean ± s.d. Scale bars: **a**, 1 mm; **l**, 500 μm; **d**, 100 μm; **b**,**e**,**i**,**m**, 50 μm; **f**, 20 μm; **j**, 10 μm.

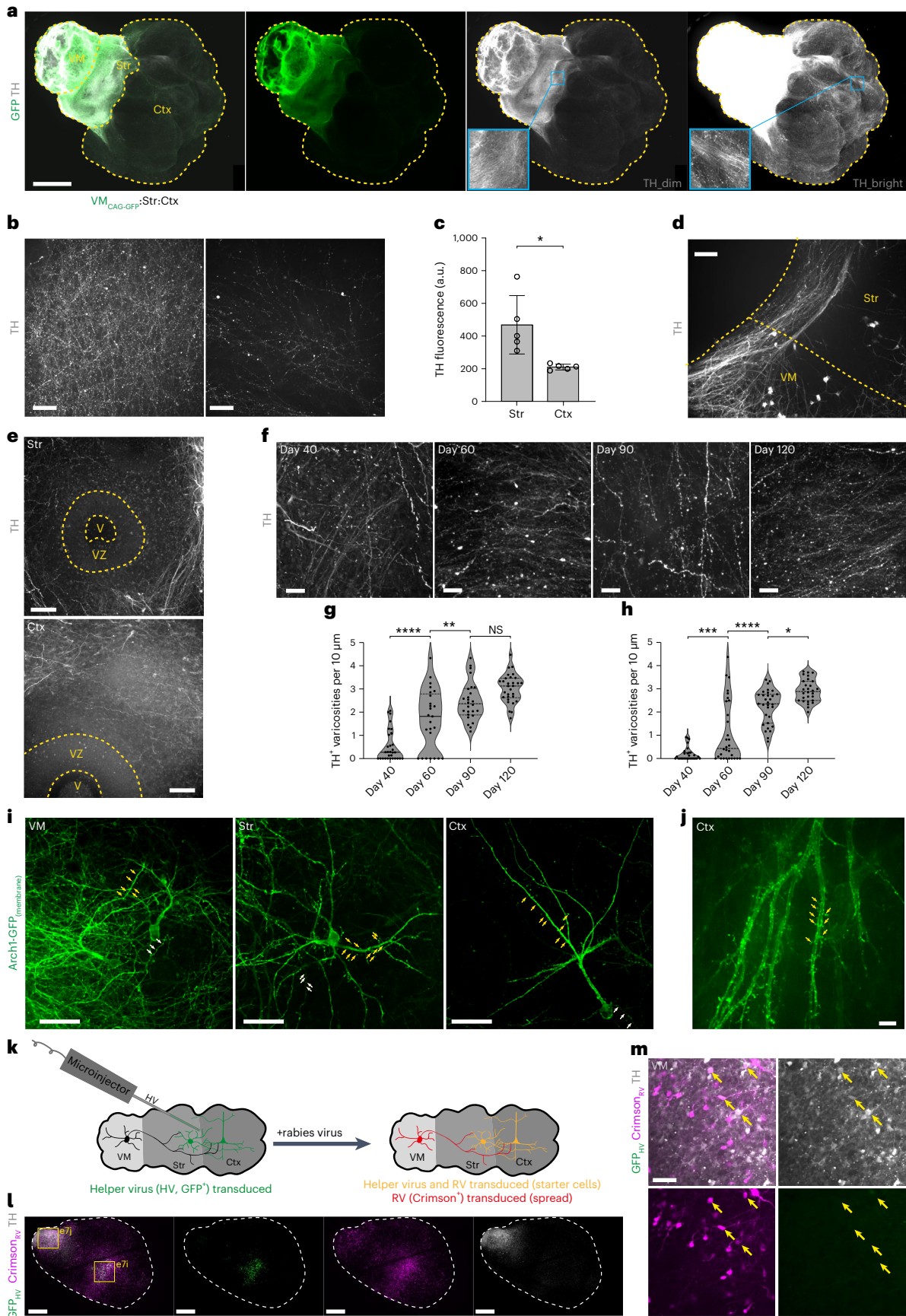

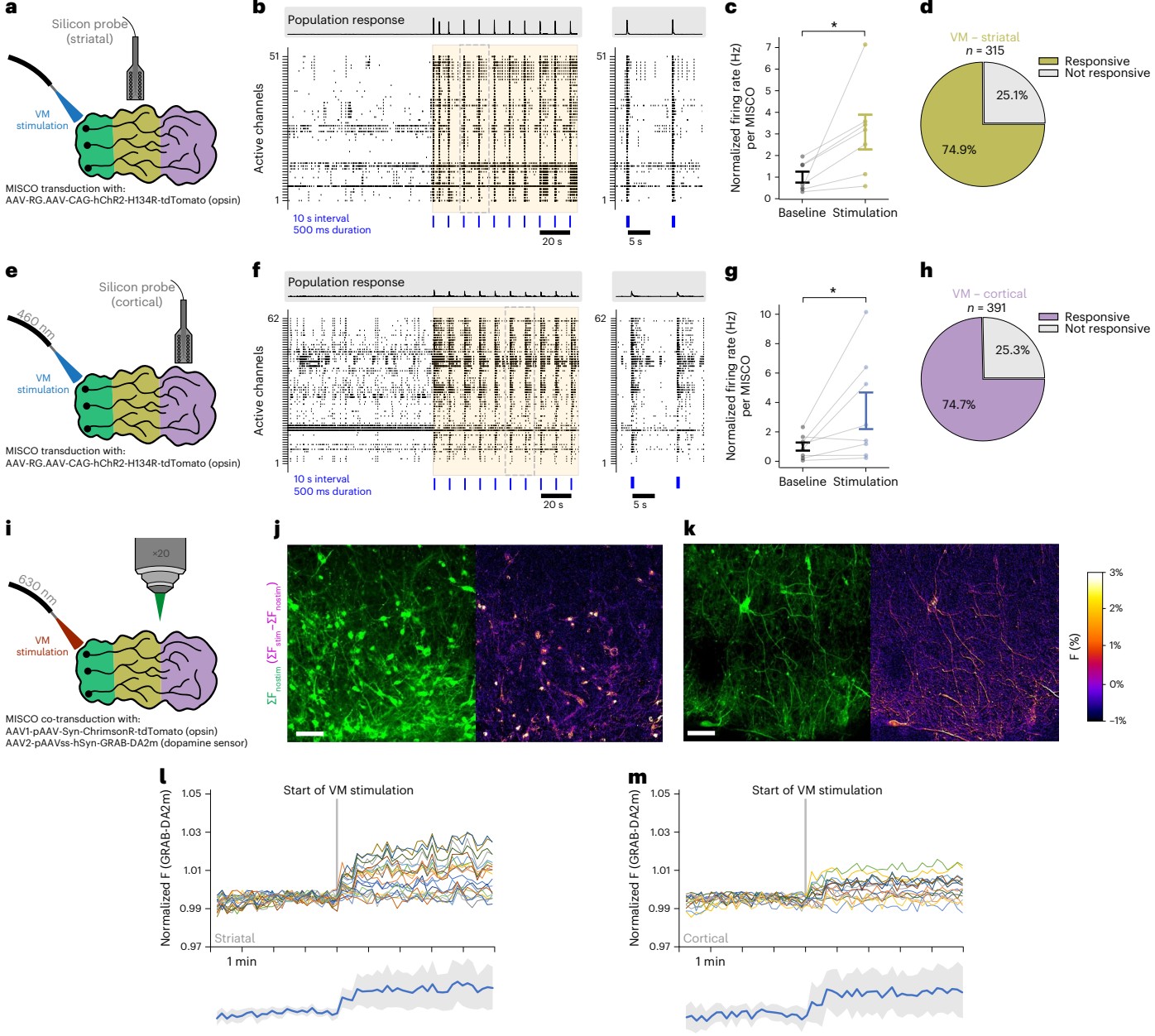

**Fig. 4 | MISCOs show neuronal activity and functional dopaminergic connectivity. a,e,** Schematic diagram of the optogenetic stimulation of VM with simultaneous extracellular recordings in striatal (**a**) and cortical (**e**) tissue. 140–171-day-old MISCOs were transduced with AAV-RG.AAV-CAG-hChR2-H134R-tdTomato and stimulated with 460 nm light focused on VM tissue using an optical fiber. **b,** Representative active channel raster plots in striatal tissue ±100 s from the initiation of optogenetic stimulation (orange box) with a 20 s recording interval (right). The 460 nm LED light pulses were 500 ms in duration and occurred every 10 s (blue lines). **c,** Normalized firing rate (Hz) changes in striatal tissue across 10 min baseline and 10 min optogenetic stimulation periods, calculated per MISCO across active channels (*n* = 7 independent experiments across two organoid batches, a total of 315 active channels, two-sided Wilcoxon signed-rank test, \**P* = 0.016). **d,** Percentage of active electrodes from extracellular recordings in striatal tissue responsive to VM optogenetic stimulation (235 of 315 active responding channels from 7 independent experiments across two organoid batches). **f,** Representative active channel raster plots in cortical tissue ±100 s from the initiation of optogenetic stimulation (orange box). **g,** Normalized

firing rate (Hz) changes in cortical tissue across 10 min baseline and 10 min optogenetic stimulation periods, calculated per organoid from active channels (8 independent experiments across three organoid batches, a total of 391 active channels, two-sided Wilcoxon signed-rank test, \**P* = 0.039). **c,g,** Data given as mean ± s.e.m. **h,** Percentage of active electrodes in cortical tissue responsive to VM optogenetic stimulation (292 of 391 active responding channels from 8 independent experiments across three organoid batches). **i,** Schematic diagram of optogenetic stimulation of VM tissue and simultaneous fluorescent confocal recording of striatal and cortical tissue. MISCOs were transduced with AAVs containing the optogenetic construct ChrimsonR and the fluorescent genetic encoded dopamine sensor GRAB-DA2m. **j,k,** Recordings of striatal (**j**) and cortical (**k**) tissue in 130-day-old MISCOs show increased fluorescence (F) of GRAB-DA4.4 upon VM stimulation. **l,m,** Timelapse recording of striatal (**l**) and cortical (**m**) regions in MISCOs showing an increase of dopamine release upon stimulation of VM tissue. Bottom: Individual neural GRAB-DA4.4 intensity of both striatal and cortical tissues (mean in blue with s.d. in gray; *n* = 3 of 3 recordings of one batch). Scale bars: **j,k,** 50 μm.

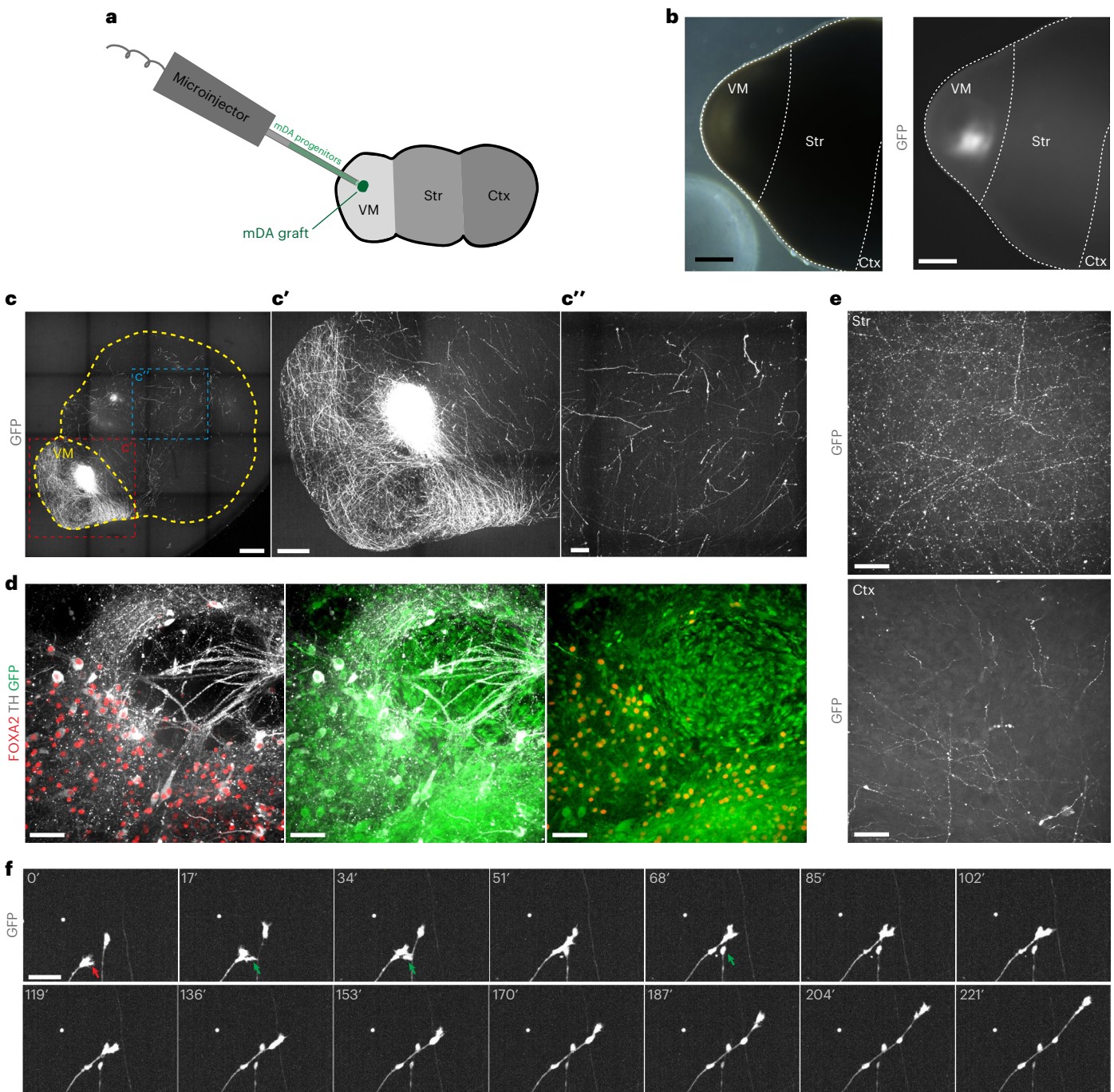

**Fig. 5 | Injection of high purity ventral midbrain progenitors for Parkinson's disease cell therapy enables study of graft innervation. a**, Schematic diagram of injection of dopaminergic progenitors into MISCOs. A total of 40,000 cells in a 200 nl volume were injected into the VM part of MISCOs. **b**, One month after injection, GFP-positive cells were readily visible at the injection site (representative image; similar results in $n$ = 5 of 5 organoids). Left: brightfield of the VM side of a MISCO. Right: same position with GFP fluorescence to visualize injected GFP⁺ cells. **c**, Surface recordings 20 days after injection demonstrate graft innervation from VM (**c′**) into striatal and cortical tissues (**c″**) (representative images; similar results in $n$ = 3 of 3 organoids). **d**, 2Eci recording of a graft in VM immunolabeled for GFP, FOXA2 and TH. **e**, Striatal tissue (top) had denser innervation of grafted cells than cortical tissue (bottom) in 155-day-old organoids. **f**, Live imaging of MISCOs with VM graft with axon outgrowth into the forebrain tissue. Growth cone (red arrow) and axon–axon interactions (green arrows) were readily observable. Scale bars: **b**,**c**, 500 μm; **c′**, 250 μm; **c″**, 100 μm; **e**, 50 μm; **d**,**f**, 20 μm.

enabled us to detect midbrain dopaminergic, striatal (including DRD1- and DRD2-associated sub-clusters) and cortical neuron clusters, as well as ventral midbrain floor plate, LGE and cortical progenitors. We additionally found MGE- and caudal ganglionic eminence (CGE) progenitor and interneuron, midbrain glutamatergic and GABAergic neuron, and glial cell clusters (Fig. 2j and Extended Data Fig. 6d–h). Correlation of the striatal and cortical clusters with the BrainSpan

dataset of the developing human brain at postconception week 20–25 indicated a strong similarity index of cortical neurons with the developing neocortex, while striatal neurons were closest to the developing striatum (Fig. 2k). VoxHunt spatial similarity brain mapping of midbrain dopaminergic, striatal and cortical clusters against the E13.5 Allen Developing Mouse Brain Atlas data showed that all three clusters correlated strongly with their corresponding tissue region

in vivo, further confirming midbrain dopaminergic, striatal and cortical neuron identity (Fig. 2l,m).

Thus, MISCOs are composed of dopaminergic, striatal and cortical neurons in a spatially organized manner, which can be used to study the innervation of midbrain dopaminergic neurons into striatal and cortical tissues.

### Neurons structurally mature and innervate target tissues

To visualize dopaminergic neurons innervating distal regions in MISCOs, we combined whole mount immunohistochemistry with second-generation ethyl cinnamate-based tissue clearing (2Eci)[39]. Immunolabeling of TH enabled us to observe midbrain dopaminergic axons projecting into the striatal and cortical tissue (Fig. 3a). Innervation of striatal tissues was denser than in cortical tissue, resembling the in vivo situation[40] (Fig. 3b,c). Furthermore, midbrain dopaminergic axons formed axon bundles projecting in the direction of the forebrain (Fig. 3d) and did not innervate neurogenic regions (Fig. 3e). Differentiation and circuit formation of midbrain dopaminergic neurons is accompanied by the appearance of axonal active zone-like release sites, which we assayed by measuring varicosities on TH-positive dopaminergic axons[41]. Indeed, TH-positive varicosities in striatal and cortical tissues of 2Eci tissue-cleared MISCOs increased significantly between day 40 and day 120 (Fig. 3f–h).

To investigate reciprocal connectivity, we generated chimeric MISCOs constitutively expressing GFP (CAG-GFP) in the striatum (Str) or cortex (Ctx) tissue[23]. Str$_{CAG-GFP}$ MISCOs highlighted GFP and GABA double-positive reciprocal connections from striatal into ventral midbrain, as well as GFP and GABA double-positive interneurons that migrated into the cortical tissue (Extended Data Fig. 7a). Notably, striatal projections into the ventral midbrain are dis-inhibitory and play a crucial role in decision-making and motor control in vivo[42,43]. Ctx$_{CAG-GFP}$ MISCOs had extensive innervation in both striatum and ventral midbrain tissue, as also found in vivo[44,45]. Intriguingly, innervation from the cortical into the striatal organoid broadly formed axon bundles in the striatum, which were not detected in ventral midbrain, and which we speculate could represent the formation of cortico-striatal fiber tracts[46] (Extended Data Fig. 7b). Finally, ventral midbrain (VM)$_{CAG-tdTomato}$-Str$_{WT}$-Ctx$_{CAG-GFP}$ MISCOs had strong innervation of striatal tissue (DARPP32$^+$, GFP$^-$, tdTomato$^-$) by both ventral midbrain and cortical tissue (Extended Data Fig. 7c). Notably, midbrain dopaminergic neurons in the ventral midbrain often formed dopaminergic clusters (Extended Data Fig. 7d). Sparse TH-positive cells were also found in striatal and cortical tissues. To investigate their identity, we grew MISCOs with Str$_{DLXi5/6-GFP}$[31] and identified GFP and TH double-positive cells, indicating a TH-positive interneuron or striatal TH-positive neuron identity (Extended Data Fig. 7e).

To investigate whether region-specific neuronal morphologies could be recapitulated, we transduced MISCOs with an adeno-associated virus (AAV) containing a membrane-bound GFP (Syn-Arch1-GFP)[47] and found neurons resembling putative cortical pyramidal neuron shapes, multipolar striatal neurons and heterogeneous morphologies in the ventral midbrain (Fig. 3i,j). Neurons also had putative dendritic spines and axonal boutons in 150-day-old organoids, indicating morphological maturation (Extended Data Fig. 7f–h).

To test neural connectivity in MISCOs, we used monosynaptic retrograde rabies viral tracing[48,49]. MISCOs had a stereotypic morphology, enabling the identification of ventral midbrain and forebrain tissue (Extended Data Fig. 9u). Helper virus (Synapsin-TVA-N2cG-EGFP) was specifically injected into the forebrain tissue of 120-day-old MISCOs. MISCOs were transduced with a monosynaptic variant of rabies virus (RV-CVS-N2c-ΔG-Crimson) after 2 weeks, and cultured for 2 more weeks (Fig. 3k). 3D immunolabeling for TH and 2Eci tissue clearing identified rabies virus-positive cells throughout the MISCOs. Most rabies virus-positive cells were found near the injection site, but additional GFP$^-$RV$^+$ clusters were localized in the ventral midbrain (Fig. 3l and Extended Data Fig. 7i–k), where many rabies virus-positive cells could be found, as well as midbrain dopaminergic neurons (Fig. 3m).

Thus, MISCOs contain neurons with ventral midbrain, striatal and cortical neuronal morphologies and recapitulate complex patterns of neuronal innervation and structural maturation of the dopaminergic system, including the formation of reciprocal connectivity.

### MISCOs form integrated, functional neural networks

To investigate functional activity, we generated chimeric MISCOs with a cell line expressing the fluorescent calcium indicator GCAMP6s (Syn-GCAMP) in either ventral midbrain, striatal or cortical tissue. Spontaneous intracellular calcium transients were found in all three regions in 130-day-old MISCOs (Extended Data Fig. 8a–c), demonstrating functionally active neurons. Additionally, GCAMP$^+$ axons from ventral midbrain displayed calcium events in the forebrain (Extended Data Fig. 8d,e), indicating the existence of spontaneously active long-range connections between ventral midbrain and forebrain.

To test whether long-range projections from ventral midbrain can influence network activity in other regions, we transduced MISCOs with an AAV containing the light-sensitive opsin hChR2[50]. We then optogenetically activated only the ventral midbrain by spatially restricted illumination using a glass fiber while simultaneously recording extracellular neural activity from either striatum or cortex tissue using silicon neural probes (Fig. 4a,e and Extended Data Fig. 8f–h). Optogenetic stimulation of ventral midbrain increased multi-unit firing rates in both striatal (Fig. 4a–c) and cortical tissue (Fig. 4e–g)

---

**Fig. 6 | Cocaine treatment of MISCOs enables the study of perturbations of the dopaminergic system in vitro. a**, Schematic diagram of the treatment of MISCOs with cocaine. Organoids were treated with 0.7 µM cocaine hydrochloride for 1 h every 3 days from day 40 until day 130 (chronic condition) and until day 105 (withdrawal condition). Functional, morphological and transcriptional analysis was performed on day 130. **b**, TH- immunolabeled dopaminergic axons in striatal (left) and cortical tissue (right) in control, chronic and withdrawal MISCOs (representative images; similar results in $n$ = 6–8 organoids per condition, 39–49 axons). **c**, Schematic diagram of the varicosity density and varicosity diameter parameters in dopaminergic axons. **d**, Quantification of TH-positive varicosity density in striatal (left) and cortical (right) tissue ($n$ = 6–8 organoids per condition, 39–49 axons). ****$P_{adj}$ < 0.0001, *$P_{adj}$ = 0.0323 (Str), NS, $P_{adj}$ = 0.5420 (Ctx). **e**, Measurement of varicosity diameter (as mean of all boutons per axon) in striatal (left) and cortical (right) tissue ($n$ = 6–8 organoids, 36–47 axons per condition and 1,055, 1,552, 1,347 (striatal) and 923, 1,063, 1,340 (cortical) boutons measured). ****$P_{adj}$ < 0.0001, ***$P_{adj}$ = 0.0002, NS, $P_{adj}$ = 0.2078 (Str) and $P_{adj}$ = 0.7079 (Ctx). **f**, Schematic diagram of the parameters frequency and duration from GCAMP traces after extraction with CaImAn. **g,h**, MISCOs with

GCAMP6S expression in either VM, striatal or cortical tissue were recorded on day 130 in control, chronic and withdrawal conditions (10–21 organoids per condition). **g**, Analysis of calcium event duration showed a significant decrease in striatal (left) and an increase in cortical (right) neuron calcium event duration (striatal: 1,776, 2,144, 4,254 individual calcium events; cortical: 1,437, 6,931, 8,000 individual calcium events). ****$P_{adj}$ < 0.0001. NS, $P_{adj}$ = 0.7303 (Str) and $P_{adj}$ = 0.9653 (Ctx). **h**, Analysis of calcium event frequency for striatal neurons (left) ($n$ = 248, 245, 680 events) and cortical neurons (right) ($n$ = 175, 936, 885 events) in control, chronic and withdrawal conditions. ***$P_{adj}$ = 0.0009, ***$P_{adj}$ = 0.0002, NS, $P_{adj}$ = 0.9434 (Str). ****$P_{adj}$ < 0.0001 *$P_{adj}$ = 0.0300 (Ctx). Data given as mean ± s.d. Samples were analyzed with one-way ANOVA followed by Tukey's multiple comparison test. **i,j**, Volcano plot comparing bulk RNA-seq data of day 130 forebrain organoids for chronic versus control (**i**) and withdrawal versus control (**j**) ($n$ = 6–9 organoids of 2–3 batches). FC, fold change. **k**, GO term overrepresentation analysis of genes downregulated in forebrain (fb) chronic and withdrawal conditions versus forebrain control. Data given as mean ± s.d. Scale bar: **b**, 10 µm.

with at least 74% of active channels responding to 460 nm light pulses (Fig. 4d,h). To exclude antidromic signal propagation through reciprocal axons, we broadly inhibited synaptic transmission with a cocktail of synaptic blockers (D-AP5, CNQX, gabazine, SCH-23390 and sulpiride$^{-/-}$). In the presence of this cocktail, population responses to ventral midbrain stimulation were absent and the normalized firing rate per channel decreased significantly, indicating functional trans-synaptic long-range connectivity (Extended Data Fig. 8i,j).

To directly confirm dopamine release from midbrain dopaminergic axons, MISCOs were transduced with AAVs containing the fluorescent dopamine sensor GRAB-DA2m[51] as well as the light-sensitive opsin ChrimsonR[52] (Fig. 4i and Extended Data Fig. 8k,l). We optogenetically stimulated ventral midbrain tissue (Extended Data Fig. 8h) and

recorded changes in fluorescence from striatal and cortical tissues. Ventral midbrain stimulation increased dopamine levels in most striatal and cortical neurons throughout the period of stimulation (Fig. 4j–m).

To examine whether the formation of long-range connections would lead to changes in activity-regulated genes compared with unfused organoids, we performed RNA-seq of striatal and cortical tissue of MISCOs and compared activity-regulated genes, as well as dopamine signaling response genes in fusions versus individual organoids. Remarkably, fusion resulted in an upregulation of activity-regulated genes such as FOS and BDNF in both striatal and cortical tissues (Extended Data Fig. 8m,n).

In summary, MISCOs develop integrated, functional neural networks across tissues, including midbrain dopaminergic-releasing

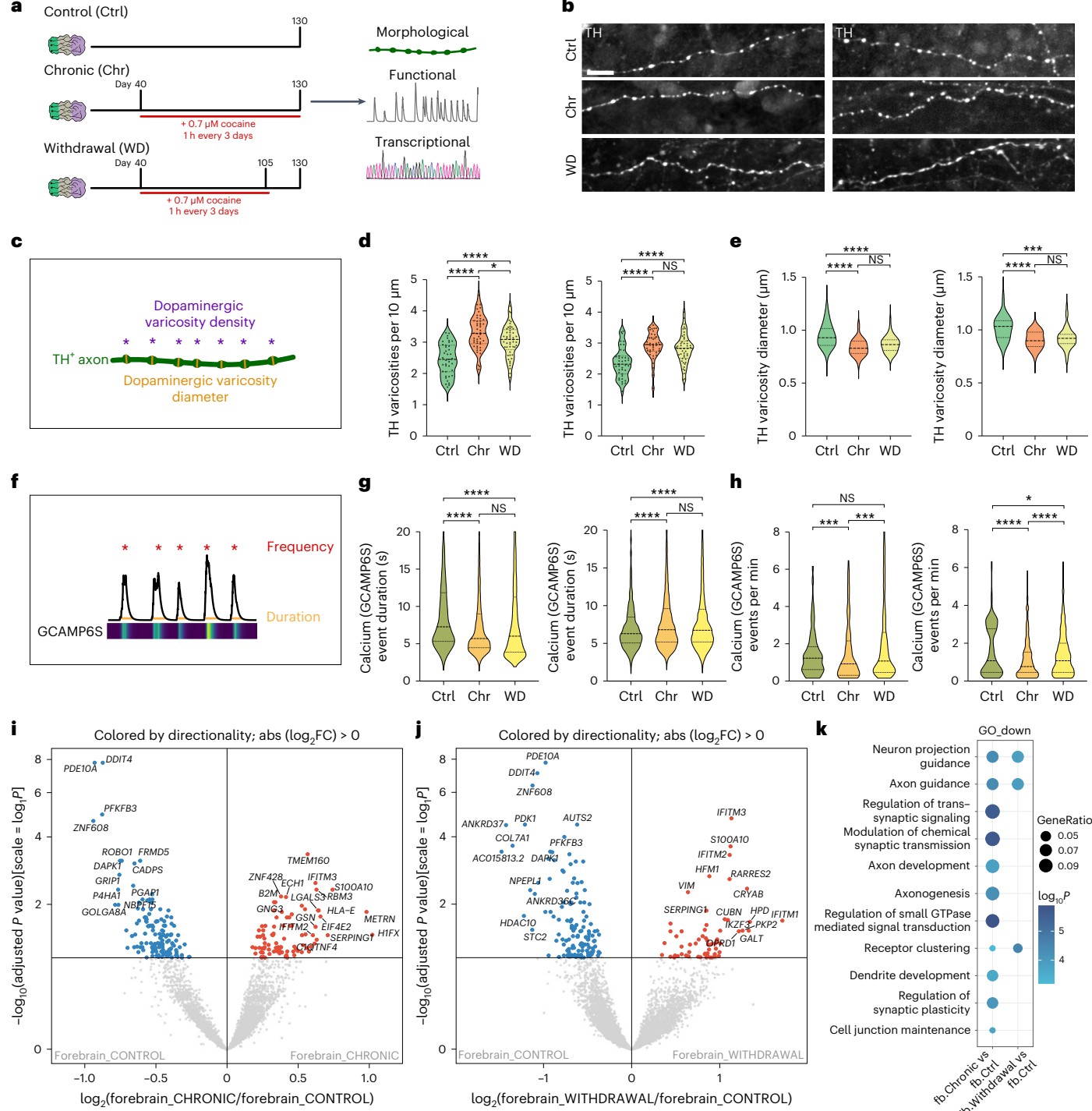

axonal innervation, and recapitulate the influence of ventral midbrain neurons on network activity in striatum and cortex.

## MISCOs as a platform for Parkinson's disease cell replacement therapy

To test whether MISCOs could be used as a human-specific in vitro model to study aspects relevant for Parkinson's disease cell replacement therapy approaches, we injected ventral midbrain progenitors into MISCOs. We used ventral midbrain progenitors with TH-positive cells being genetically labeled with GFP using lentiviral transductions into a *TH*-Cre reporter line as previously described[53], and injected 40,000 progenitors into the ventral midbrain tissue[25] (Fig. 5a and Extended Data Fig. 9a–t,u). After 1 month, GFP-positive cells were readily visible, indicating survival and maturation of injected progenitors (Fig. 5b). Whole mount microscopy recordings showed innervation of striatal and cortical tissues (Fig. 5c). Grafts were positive for FOXA2 and TH, suggesting differentiation of injected cells into midbrain dopaminergic neurons (Fig. 5d). 2Eci of the MISCOs injected with ventral midbrain progenitors showed that already at 3 days after injection, cells sprouted neurites, with a drastic increase in innervation on days 18 and 95, and with the grafted cells innervating striatal tissue more densely than cortical tissues (Fig. 5e and Extended Data Fig. 9v,w). Live imaging showed innervating axons from the graft (Fig. 5f and Supplementary Video 1). Thus, MISCOs can be used as a platform to study the behavior and maturation of ventral midbrain progenitors in an in vitro human-specific, high-throughput model in which the cells are part of their natural circuits.

## Application of MISCOs to study long-term effects of cocaine

Addictive substances increase dopaminergic signaling by directly or indirectly regulating the dopaminergic system[54,55]. Cocaine binds to DAT and inhibits the reuptake of dopamine after release and thus increases the available levels of dopamine to target cells[56]. While the effects of cocaine on the adult brain are relatively well understood, not much is known about the effects of cocaine on the development of the central nervous system, predominantly because of the lack of an effective human model system. We tested whether MISCOs can be used to investigate the effects of cocaine exposure and its potential effect on the establishment of the dopaminergic system both during chronic exposure and after withdrawal. MISCOs were treated with 0.7 µM cocaine hydrochloride (the half-maximal inhibitory concentration (IC50)), a concentration well within the physiological range in humans[57,58] (Fig. 6a). Treatments were performed from day 40 to 130 for 1 h every 3 days to simulate the pharmacokinetics of cocaine in humans. Withdrawal was modeled by depleting cocaine from day 105 onwards.

Cocaine exposure in vivo was shown to affect neuronal plasticity, as characterized by morphological changes and increased density of dopaminergic varicosities[59,60]. We therefore quantified the density of varicosities on TH-positive axons in striatal and cortical tissue and measured the varicosity diameter as a proxy for varicosity size (Fig. 6b,c). Strikingly, we found that the density of varicosities was significantly increased in both tissues, with the withdrawal condition showing only a tendency of recovery (Fig. 6d). Moreover, the diameter of TH-positive varicosities was significantly reduced in both tissues, indicating varicosities of reduced volume, and this effect was not reversed after 25 days of withdrawal (Fig. 6e). Thus, MISCOs recapitulate the effects of cocaine exposure on dopaminergic axons previously described in vivo.

In vivo, cocaine exposure has been associated with changes in neural activity[61,62]. To investigate whether we could recapitulate functional changes, the activity of chimeric MISCOs with region-specific syn-GCAMP6S expression was recorded at day 130 (Extended Data Figs. 8a–c and 10a,b). Using the calcium imaging analysis pipeline CaImAn[63], we extracted a total of 3,662 neurons in the control, chronic treatment and withdrawal conditions of ventral midbrain, striatal and cortical tissues in MISCOs[63]. We found that the calcium event

duration in the striatum became significantly shorter in the chronic treatment condition (Fig. 6f,g). Conversely, cortical patterned neurons showed a significant increase in calcium event duration. No significant recovery was observed in the withdrawal condition. Interestingly, ventral midbrain neurons showed a decrease in calcium event duration without recovery upon cocaine withdrawal (Extended Data Fig. 10c). Furthermore, we found that neurons in the ventral midbrain, striatal and cortical tissue generally had fewer events per minute, which did only partially recover in the withdrawal condition (Fig. 6h and Extended Data Fig. 10d), thereby strengthening our observations that exposure to cocaine causes long-term changes in neuronal activity.

To investigate transcriptional changes upon cocaine exposure, we separated MISCOs into forebrain and ventral midbrain and separately performed RNA-seq. Although little transcriptional change was seen in ventral midbrain (Extended Data Fig. 10e,f), a broad list of genes were differentially expressed in the chronic and withdrawal conditions in forebrain (Fig. 6i,j). Among the top hits of upregulated genes in the chronic and withdrawal conditions, we found genes associated with interferon response (such as IFITM2 and IFITM3). Notably, a correlation between cocaine and interferon response has previously been shown in neural stem cells and astrocytes[64,65]. Conversely, genes associated with neural circuit formation were downregulated, which was also apparent in gene ontology (GO) term analysis[66,67] (Fig. 6k). Although chronic treatment produced the strongest transcriptional changes regarding circuit formation, withdrawal still displayed significant changes. Gene set enrichment analysis identified GO terms associated with neural circuit formation (dendrite development, axon development and synapse organization) to be consistently and significantly downregulated upon cocaine treatment (Extended Data Fig. 10g–i). Downregulation of genes in all three GO terms was consistently observed upon chronic exposure but continued upon withdrawal, further highlighting the long-term effects of cocaine treatment on neural circuit formation.

Thus, chronic exposure to physiological concentrations of cocaine affects neuronal activity and circuit formation at the morphological and transcriptional level. Notably, effects lasted after withdrawal of cocaine, suggesting a long-lasting impact on the development of the dopaminergic system and potentially on other circuits. In the future, analyzing cell-type specific readouts (for example, DRD1/DRD2 medium spiny neurons, A9/A10 midbrain dopaminergic contribution) will enable elucidation of these perturbations in more detail.

## Discussion

In this study we have developed a 3D in vitro model using brain organoids that recapitulates key morphological and functional features of the human dopaminergic system. To date, most studies investigating dopaminergic neuronal circuitry rely on animal models, which are limited by inherent physiological differences to the human brain. While the development of the murine dopaminergic system can be counted in days, the human dopaminergic system develops over months, opening a crucial and vulnerable time window for potential perturbations. Modeling human dopaminergic neurodevelopment in ventral midbrain organoids has been previously demonstrated, but these models did not address the connectivity of midbrain dopaminergic neurons with their target regions. We demonstrate that MISCOs provide an opportunity to study the formation of functional dopaminergic innervation, the release of dopamine into the striatum and cortex, and the formation of reciprocal connections. This opens new avenues for studying the dopaminergic system in vitro, including morphological and functional maturation of both neurons and neural circuits, and might help to further reduce animal experiments.

We describe two different applications in which MISCOs can be used to study the human dopaminergic system. First, they can be used to recapitulate key aspects of circuitry establishment in grafted ventral midbrain progenitors, which is of relevance for developing cell therapies in Parkinson's disease, and demonstrates an application of MISCOs

in an exciting phase of innovation in which pioneering clinical stem cell trials are underway[68,69]. Nevertheless, there is still the need to better control precise dopaminergic subtype identity and enhance the capacity for innervation of the correct target region[70–73]. We believe that the MISCO methodology could be used as an accessible high-throughput platform to speed up the development of new approaches to increase and/or guide dopaminergic innervation.

Second, MISCOs can be used to study the morphological, functional and transcriptional effects of chronic cocaine exposure. Notably, effects were persistent even after withdrawal, indicating long-term changes of neuronal circuits.

This further strengthens the key relevance of brain organoids as models to study neuronal circuit formation and the effects of environmental factors on neuronal circuits and enables the possibility of pharmacological investigation of the dopaminergic system. Although our studies have been limited to investigation of the effects of cocaine and its effects at a population level, our platform provides an ideal system to study the link between dysfunction of the dopaminergic system and associated psychiatric disorders (such as attention-deficit–hyperactivity disorder, depression and schizophrenia) also in a neural subtype-specific manner[74,75]. Our system could also be used for pharmacological and toxicological studies investigating, for example, the effects of pesticides and herbicides to understand how environmental factors could lead to sporadic Parkinson's disease[13,76,77]. We also suggest that this system could be used in overexpression models of, for example, α-synuclein to study familial Parkinson's disease and prion spread in vitro[13,78,79].

MISCOs might also be used to study Huntington's disease, particularly the spread of pathogenic huntingtin (HTT) from the striatum and its effect on different neural populations[80]. The most commonly used antidepressants worldwide are selective serotonin reuptake inhibitors (SSRIs), however, the mechanism of depression and the function of SSRIs in the brain are poorly understood[81,82]. MISCOs could be expanded to include hindbrain organoids to facilitate studies of the serotonergic system and its effects on other brain regions. Furthermore, we propose that our engineered fusion method could be expanded to other axes, for example, the addition of lateral or ventral–dorsal organoids to generate even more sophisticated brain-like interaction networks such as hippocampal organoids[83] for contribution to memory formation, or retinal organoids[84] for re-creation of the visual pathways. Despite their great potential, MISCOs are limited by the same restrictions that apply to other organoid models, such as lack of vascularization and incomplete neural maturation. In addition, although their degree of functionality is surprising, MISCOs do not fully recapitulate the in vivo architecture, and we do not know whether the axonal innervation we see follows the same cellular specificity that is seen in vivo. Further work could address these issues, for example by creating perfused or vascularized tissues, or by artificially speeding up neural maturation and by performing connectome analysis to investigate the specificity of interactions.

Thus, we provide a versatile model for studying the development and impairment of the human dopaminergic system that expands the capabilities of in vitro modeling of brain development to the development and function of neural circuits, and which can be used in basic science as well as in applied research.

## Online content

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

**Daniel Reumann**[1,2], **Christian Krauditsch** [1], **Maria Novatchkova**[1], **Edoardo Sozzi**[3], **Sakurako Nagumo Wong**[1,2], **Michael Zabolocki** [4,5], **Marthe Priouret**[1], **Balint Doleschall**[1,2], **Kaja I. Ritzau-Reid**[6], **Marielle Piber**[1,10], **Ilaria Morassut** [1,11], **Charles Fieseler**[7], **Alessandro Fiorenzano** [3,8], **Molly M. Stevens** [6], **Manuel Zimmer**[7], **Cedric Bardy** [4,5], **Malin Parmar** [3] **& Jürgen A. Knoblich** [1,9] ✉

[1]Institute of Molecular Biotechnology (IMBA) of the Austrian Academy of Sciences, Vienna BioCenter, Vienna, Austria. [2]Vienna BioCenter, Doctoral School of the University of Vienna and Medical University of Vienna, Vienna, Austria. [3]Department of Experimental Medical Science, Developmental and Regenerative Neurobiology, Wallenberg Neuroscience Center, Lund Stem Cell Center, Lund University, Lund, Sweden. [4]Laboratory for Human Neurophysiology and Genetics, South Australian Health and Medical Research Institute (SAHMRI), Adelaide, South Australia, Australia. [5]Flinders Health and Medical Research Institute, Flinders University, Adelaide, South Australia, Australia. [6]Department of Materials, Department of Bioengineering, Institute of Biomedical Engineering, Imperial College London, London, UK. [7]Department of Neuroscience and Developmental Biology, University of Vienna, Vienna, Austria. [8]Stem Cell Fate Laboratory, Institute of Genetics and Biophysics 'Adriano Buzzati Traverso' (IGB), CNR, Naples, Italy. [9]Department of Neurology, Medical University of Vienna, Vienna, Austria. [10]Present address: Zebrafish Neurogenetics Unit, Institut Pasteur, Paris, France. [11]Present address: Department of Basic Neurosciences, Faculty of Medicine, University of Geneva, Geneva, Switzerland. ✉e-mail: Juergen.knoblich@imba.oeaw.ac.at

## Methods

### Cell lines and cell culture

The hES cell lines (H1 (WAe001-A), H7 (WA07), H9 (WA09) (all WiCell) and in-house generated iPS cell lines (176/1, 178/5, 178/6) were cultured feeder-free on growth factor-reduced Matrigel (Corning, cat. no. 354277) in mTESR1 (Stemcell Technologies, cat. no. 85875). Cells were tested for genome integrity, and *Mycoplasma* tests were performed regularly. Cells were split after 3–5 days with avoidance of confluently coated wells and splitting before colonies develop a dense core, in a ratio of 1:6 to 1:10 using $PBS^{-/-}$ containing 0.5 mM EDTA (PBS: Bio Trend PBS-1A, EDTA: Sigma-Aldrich, cat. no. E6758) and incubation until the colonies start to break apart (4–8 min after starting incubation). All cell culture in this work was performed in a 5% $CO_2$ incubator at 37 °C.

### Generation of ventral midbrain, striatal and cortical organoids

The exact formulations for neural induction medium, Improved−A medium, Improved+A medium and Brainphys are given in the Organoid Media section. For brain organoid (BORG) generation, cells were used 2–3 days after splitting. Cells were dissociated using 600 µl Accutase (Sigma-Aldrich, cat. no. A6964) and incubated until all cells detached from the plate (3–5 min). Cells were washed off with 1.4 ml stem cell medium and spun down at 150 ×*g* for 5 min. Cells were resuspended in 1 ml stem cell medium containing 1:100 ROCK (Rho kinase) inhibitor Y27632 (Selleck Chemicals, cat. no. S1049) and counted. A total of 9,000 cells in 150 µl neural induction medium plus patterning factors (ventral midbrain and striatal) or stem cell medium (cortical) containing 1:100 ROCK inhibitor per well were then transferred into an ultra-low attachment U-shaped 96-well plate (Thermo Fisher Scientific, cat. no. 136101 and Corning, cat. no. 7007).

For ventral midbrain patterning, 9,000 cells were transferred into 150 µl neural induction medium containing 200 ng ml$^{-1}$ Noggin (R&D Systems, cat. no. 6057), 10 µM SB431542 (Stemgent, cat. no. 04-0010-10) and 1 µM CHIR99021 and ROCK inhibitor in an ultra-low attachment U-shaped 96-well plate. On day 2, 100 µl of this medium was removed and replaced with 150 µl fresh medium of the same formulation. On day 4, 150 µl medium was exchanged, with the addition of 200 ng ml$^{-1}$ Noggin, 10 µM SB431542, 1 µM CHIR99021, 300 nM SAG (Merck, cat. no. US1566660) and 100 ng ml$^{-1}$ FGF-8 (fibroblast growth factor 8; R&D Systems, cat. no. 5057-FF) to the newly added medium. On day 6, 150 µl medium was exchanged with 150 µl neural induction medium containing 300 µM SAG and 100 ng ml$^{-1}$ FGF-8. On day 8 up to 30 embryoid bodies were transferred into a 10 cm plate coated with anti-adherence rinsing solution (Stemcell Technologies, cat. no. 07010) and fed with 12 ml Improved−A medium containing 2% liquid Matrigel (Corning, cat. no. 356235) (added to fridge-cold medium and used within 1 h) and given a last treatment with 300 µM SAG and 100 ng ml$^{-1}$ FGF-8 for the first feeding. Notably, hPSCs are variable in morphogen expression and sensing, and hence further dose curving might be required in other cell lines.

For striatal patterning, 9,000 cells were transferred into 150 µl neural induction medium containing 10 nM SAG and 2.5 µM IWP-2 (Sigma-Aldrich, cat. no. 10536) as well as 1:100 ROCK inhibitor in an ultra-low attachment U-shaped 96-well plate. The medium was exchanged on day 2 by replacing 100 µl medium with 150 µl fresh neural induction medium containing 10 nM SAG, 2.5 µM IWP-2 and 1:100 ROCK inhibitor. On day 4 the medium was exchanged by replacing 150 µl medium with 150 µl fresh neural induction medium containing 10 nM SAG and 2.5 µM IWP-2. On day 6, 150 µl medium was exchanged with neural induction medium without factors. On day 8 up to 30 embryoid bodies were transferred into a 10 cm plate coated with anti-adherence rinsing solution. Medium was exchanged with 12 ml Improved−A medium containing 2% Matrigel (added to fridge-cold medium and used within 1 h). Notably, hPSCs are variable in morphogen expression and sensing, and further dose curving might be required in other cell lines.

For cortical patterning, organoids have been grown as previously described. In brief, 9,000 cells were transferred into 150 µl stem cell medium with 1:100 ROCK inhibitor per well of an an ultra-low attachment U-shaped 96-well plate. After 3 days, 100 µl medium was removed and replaced with 150 µl fresh stem cell medium. On days 5, 7 and 9, 150 µl medium was exchanged with 150 µl neural induction medium. On day 11 up to 30 embryoid bodies were transferred into a 10 cm plate that was coated with anti-adherence rinsing solution (Stemcell Technologies, cat. no. 07010). The embryoid bodies were transferred into 10 ml Improved−A medium containing 2% liquid Matrigel (Corning, cat. no. 356235). On day 12, 3 µM CHIR99021 (Merck Millipore, cat. no. 361571) was added to the medium. On day 13 the medium was exchanged with fresh medium containing 3 µM CHIR99021 and not changed for 3 days.

For further differentiation of brain organoids, organoid medium was exchanged every 3 days, with a change to Improved+A medium around day 16. Organoids were transferred to orbital shakers (Inforce Celltron HT) with reduced shaking speed (42 r.p.m.) on day 20 and cultured in Improved+A medium until day 60, when a gradual shift to Brainphys medium (25%, 50%, 75%, 100%) is carried out. As soon as organoids were transferred to orbital shakers, medium volume was increased to 30 ml.

### PDMS mold generation

Embedding molds were designed in Tinkercad (www.Tinkercad.com) and were adjusted to organoid size (on the day of fusion) in diameter and length. The .stl CAD files were then sliced using the slicer software XYZprint v1.4.0. The negative was printed using an XYZ Printing da Vinci Color 3D printer using transparent PLA with 100% infill density, 0.1 mm layer height and a 215 °C nozzle temperature. Printed molds were treated with a heat gun (Bosch Hot Air Blower 1800 W) at 550 °C to carefully melt the surface of the mold and create a smooth finish and remove the extrusion printing typical rough surface. Notably, heat treatment was kept as short as possible because heating up the entire PLA negative can lead to warping. The positive embedding mold was then cast using PDMS. In brief, 5 ml curing agent and 45 ml monomer (both Sylgard 184 Elastomer Kit, VWR) were thoroughly mixed for at least 3 min. The mixture was then spun down to remove air bubbles and directly used. The 3D printed negative was placed in a 10 cm plate with the wells facing upwards. The PDMS was then carefully poured onto the middle of the mold and the plate was filled with approximately 45 ml PDMS. After pressing the negative down to remove air bubbles, the molds were cured overnight. To reduce the extent of bubble formation during curing, the molds were first cast at room temperature (24 °C). For faster curing, the mold was alternatively transferred into a 55 °C incubator, however, this can increase the release of gas and thus bubble formation in the PDMS. Alternatively to the room temperature curing step, a vacuum can be pulled to degas PDMS. When curing at room temperature overnight, the cast was transferred to 55 °C for 2–3 h the next day to finalize curing. The mold was then cut out of the plate using a scalpel and washed in 70% ethanol (pure ethanol in Milli-Q water) for 30 min and then dried and stored in a sterile environment. Embedding molds were re-used 3–4 times without an observable decline in performance. Before each use, rinsing with 70% ethanol and a 30 min optional UV treatment were carried out. Notably, PDMS that was not properly cured or which had expired, resulted in poor performance and should not be used.

### Fusion of organoids for the generation of MISCOs

Prior to use the PDMS molds were coated in an anti-adherence solution to increase the non-stick behavior of PDMS. After coating, the molds were washed once in PBS. For linearly fusing organoids, organoids were transferred group by group into the embedding molds, with up to 32 fusions at once. For this, the first group (cortical organoids) was transferred into all molds first, while transferring as little medium as possible with the organoid. Second, striatal organoids were transferred

to the right of the cortical organoids, and last, the procedure was repeated with ventral midbrain organoids. After transfer, residual medium around the organoid was removed entirely, while ensuring that the positioning of the different organoids was maintained. This step enables the attachment of organoids linearly (without medium, organoids become sticky and attach to each other). After approximately 30 s–1 min, a small amount of liquid Matrigel (~15 µl) is added to the fusions and the fusions were transferred into an incubator (37 °C, 5% $CO_2$, humidified) for 20 min. After this incubation step, MISCOs can be easily washed out of the embedding molds. The embedding molds can be directly used again for fusing organoids or washed in PBS and 70% ethanol (cell culture grade) and then dried.

Fused organoids were cultured in 10 cm plates without shaking for the first 2 days, before being transferred to an orbital shaker in 30 ml medium with low shaking speed (42 r.p.m.).

## Organoid medium
**Neural induction medium.** DMEM/F12 (Invitrogen, cat. no. 11330-057), 1% N2 Supplement (Thermo Fisher, cat. no. 17502001), 1% GlutaMAX-I (Thermo Fisher, cat. no. 35050-038), 1% MEM-NEAA (Sigma-Aldrich, M7145), 1:1,000 heparin solution (Sigma-Aldrich, cat. no. H3149-100KU), 1% PenStrep (Sigma-Aldrich, cat. no. P4333).

**Improved−A medium.** 50:50 DMEM/F12 : Neurobasal (Gibco, cat. no. 21103049), 0.5% N2 supplement, 2% B27−A (Thermo Fisher, cat. no. 12587010), 1:4,000 insulin (Sigma-Aldrich, I9278), 1% GlutaMAX, 0.5% MEM-NEAA, 1% Antibiotic-Antimycotic (Thermo Fisher, cat. no. 15240062).

**Improved+A medium.** 50:50 DMEM/F12 : Neurobasal (Gibco, cat. no. 21103049). 0.5% N2 Supplement, 2% B27+A (Thermo Fisher, cat. no. 17504044), 1:4,000 insulin (Sigma-Aldrich, I9278), 1% GlutaMAX, 0.5% MEM-NEAA, 1% Antibiotic-Antimycotic (Thermo Fisher, cat. no. 15240062), 1% vitamin C solution (40 mM stock in DMEM/F12) (Vitamin C: Sigma-Aldrich, cat. no. A4544), 1 g l⁻¹ sodium bicarbonate (Sigma-Aldrich, cat. no. S5761).

**Brainphys.** BrainPhys Neuronal Medium[85] (Stemcell Technologies, cat. no. 05790), 2% B27+A, 1% N2 Supplement, 1 ml CD Lipid Concentrate (Thermo Fisher Scientific, cat. no.11905031), 1% Antibiotic-Antimycotic, 1:147 20% glucose solution, 20 ng ml⁻¹ BDNF (Stemcell Technologies, cat. no. 78005.3), 20 ng ml⁻¹ GDNF (Stemcell Technologies, cat. no. 78057.3), 1 mM db-cAMP (Santa Cruz Biotechnology, cat. no. sc-201567C).

## RNA extraction
RNA was extracted using the RNeasy mini kit (Qiagen). Synthesis of complementary DNA was performed using 1 µg total RNA and the Superscript II (Invitrogen) enzyme, following protocols provided by the manufacturer.

## qPCR (*FOXA2*)
For quantitative polymerase chain reaction (qPCR) of *FOXA2*, 6–8 organoids were collected at indicated time points into 2 ml RNAse-free tubes. RNA was extracted as described in the RNA extraction section. qPCR was performed using Sybr Green master mix (Promega) on a BioRAD 384-well machine (CXF384). The protocol used is as follows: 95 °C for 3 min, 95 °C for 10 s, 62 °C for 10 s, 72 °C for 40 s, repeat steps 2–4 40 times, then 95 °C for 1 min, followed by 50 °C for 10 s. Quantification was performed in excel by calculating $\Delta C_t$ relative to *TBP* (which encodes TATA binding protein). Data are represented as expression level ($2^{-\Delta C_t}$) relative to *TBP*.

## qPCR with reverse transcription for midbrain dopaminergic progenitor characterization
Total RNAs were isolated using the RNeasy Micro Kit (Qiagen cat. no. 74004) and reverse transcription was performed using the Maxima First

Strand cDNA Synthesis kit (Thermo Fisher, K1641). Primers (0.95 µM, Integrated DNA technologies, Supplementary Table 1) were prepared together with cDNA and SYBR Green Master Mix (Roche) using a Bravo pipetting robot (Agilent). qPCR was performed on a LightCycler 480 II instrument (Roche) using a 40× cycle two-step protocol (60 °C 1 min annealing–elongation step and 95 °C 30 s denaturation step). Average $C_t$ values were derived from technical triplicates, analyzed with the $\Delta\Delta C_t$ method, and normalized against two distinct housekeeping genes (*ACTB* and *GAPDH*). The results are given as relative gene expression levels over undifferentiated hPSCs.

## Bulk RNA sequencing analysis
RNA was extracted as described, and sequenced using Smart-seq3. Smart-seq3 read analysis was performed using zUMI v2.9.7, providing the expected barcodes file, using STAR 2.7.7a with additional STAR parameters '−limitOutSJcollapsed 50000000 −limitIObufferSize 1500000000 −limitSjdbInsertNsj 2000000 −clip3pAdapterSeq CTGTCTCTTATACACATCT' and *Homo sapiens* Ensembl GRCh38 release 94 as a reference. Differential gene expression analysis on zUMI inex (combined intron + exon) counts and variance-stabilized transformation of count data for heatmap visualization and PCA were performed using DESeq2 v1.36.0. Gene set overrepresentation analysis of differentially expressed genes was conducted using clusterprofiler v4.4.4, and gene set enrichment analysis was performed with fgsea v1.22.0. Comparison of bulk and single transcriptomes to Allen Brain Atlas in situ hybridization data and BrainSpan micro-dissected brain data was performed with VoxHunt v1.0.1. Enrichment lists used in this work are as follows: the striatum and cortex gene lists are derived from ref. 28; the ventral midbrain gene list is from GO:0071542 plus own markers (accessed July 2022); the E13.top100.pallial gene list is from the VoxHunt E13 mouse dataset top 100 pallial enriched genes (https://doi.org/10.1016/j.stem.2021.02.015, Epub 11 March 2021); the activity-regulated gene list is from ref. 86; and the list for the dopaminergic signaling response is from rgd.mcw.edu/rgdweb/pathway/pathwayRecord.html?acc_id=PW:0000394&species=Human#gviewer (accessed July 2022) with the gene list adapted from ref. 87.

## scRNA-seq
**Preparation.** MISCOs were separated by a scalpel. Ventral midbrain, striatal and cortical tissue were processed separately. Before dissociation, necrotic material from the core of the organoid was washed off in PBS⁻/⁻. Organoids were then transferred into a 1.5 ml Eppendorf tube and all PBS was removed. A total of 1 ml Trypsin–Accutase (1:10) (Trypsin: Thermo Fisher, cat. no. 15090046) plus DNAse (2 µl ml⁻¹, TURBOTM DNase, Thermo AM2238) was added to the organoids, and the tubes were transferred to a tube shaker at 37 °C and 700 r.p.m. Tubes were flipped every 3–4 min until the tissue was dissolved, or for a maximum of 40 min. After dissociation, 500 µl ice-cold PBS⁻/⁻ were added on top. Cloudy material forming on top of the solution can be removed with a pipette. Tubes were then spun down in a precooled (4 °C) table centrifuge at 400 ×*g* for 5 min. As much liquid as possible was then removed without damaging the cell pellet. Pellets are then resuspended carefully in 400 µl PBS⁻/⁻ and filtered through a 40 µm cell strainer. Cells are then counted. The live cell count had to be above 80% and above a concentration of 200,000 cells ml⁻¹, but not more than 2.7 million cells ml⁻¹. Samples were diluted down when needed. MULTI-seq labeling using MULTI-seq Lipid-Modified Oligos was performed directly after counting following the manufacturer's protocol (Sigma-Aldrich, cat. no. LMO001). In brief, 40 µl anchor solution including barcode was mixed with the sample and incubated on ice for 5 min. A total of 40 µl co-anchor was then added and mixed and incubated on ice for 5 min. The reaction was then topped up with 1 ml ice-cold PBS⁻/⁻ containing 2% BSA, and mixed to bind residual anchor and co-anchor. Cells were spun down at 400 ×*g* for 5 min at 4 °C. The supernatant was removed and the cells were resuspended in 200 µl PBS⁻/⁻ with 2% BSA and supplemented

with 0.8 μg ml$^{-1}$ DAPI. Cells then underwent live cell sorting (DAPI$^-$) in a FACS Aria III machine with a 70 μm nozzle (~1 nl per cell) and were transferred into a 1.5 ml Eppendorf tube with 20 μl PBS$^{-/-}$ and 2% BSA.

**Analysis.** Library preparation was performed at the Vienna BioCenter next-generation sequencing facility using a 10x Genomics stranded kit and NovaSeq S1 Asymmetric 10X Illumina sequencing. scRNA-seq reads were processed with Cellranger count v7.0.0 using the prebuild 10X GRCh38 reference, refdata-gex-GRCh38-2020-A, and default settings. Seurat v4.1.1 in R Studio (2022.07.0 Build 548 and newer) was used to perform dimensionality reduction, clustering and visualization for the scRNA-seq data, together with the plot_density function of the Nebulosa package (v1.8.0)[88,89]. Samples were demultiplexed using MULTIseqDemux. Stressed cells were excluded by gruffi[88], while also only keeping cells with more than 1,500 detected genes, an UMI (unique molecular identifier) count greater than 3,000 and less than 10% mitochondrial content. Count data were log-normalized and scaled, dimensionality reduction was performed using PCA on the top 2,000 most variable genes, and the first 20 PCs were selected for the subsequent analysis. Two-dimensional representations were generated using uniform manifold approximation and projection (UMAP) (uwot v0.1.11). For top cluster marker analysis, genes were identified using the Wilcoxon rank sum test implemented in presto, and genes were ranked by $P$ value.

## Extracellular recordings and optogenetic stimulation
MISCOs were immobilized by embedding half of their height in a 4% in 0.1 M phosphate buffer agar block in a bath recording chamber (RC-41 LP, Warner Instruments). The chamber was moved to a 34 °C pre-warmed dish incubator (DH-35iL, Warner Instruments) and perfused through a peristaltic pump (PPS2, Multichannel system) with artificial cerebrospinal fluid (aCSF) containing 125 mM NaCl, 2.5 mM KCL, 1.25 mM NaH$_2$PO$_4$, 1 mM MgCl$_2$, 2 mM CaCl$_2$, 25 mM NaHCO$_3$, 10 mM D-(+)-glucose and 2 mM CaCl$_2$ equilibrated with a 95% O$_2$ and 5% CO$_2$ (vol/vol) gas mixture and adjusted to pH 7.4 with KOH (~315 mOsm). The fused organoid was first allowed to rest for up to 5 min prior to data acquisition. For optogenetic stimulation, a solid-state light source (Sola SMII, Lumencor) was connected to a collimator with insertable emission filters (460 nm, Thorlabs). From there, the filtered light was transported with an optic glass fiber (400 μM, numerical aperture (NA) 0.39, Thorlabs cat. no. M118L03) to the ventral midbrain side of an AAV-RG. AAV-CAG-hChR2-H134R-tdTomato-transduced MISCO. For stimulation, the fiber optic glass was positioned less than 1 mm away from the site of stimulation. Illumination was set at a constant light intensity and triggered every 10 s with a 500 ms duration for 10 min using the open-ephys device, Pulse Pal v2 (Sanworks). Simultaneous extracellular recordings were conducted across four shanks with a separation distance of 250 μm and 16 recording sites spaced every 25 μm (ASSY-77 P-2, Cambridge NeuroTech) in either fused striatal or cortical organoids. Recordings were sampled at 30 kHz per channel and filtered at 0.1 Hz (high-pass) and 5 kHz (anti-aliasing) following insertion ~300 μm deep into the organoid. After insertion, baseline recordings of 10–15 min were made prior to subsequent 10 min optogenetic stimulation. Acquisition was at 16 bit resolution on a 64-amplifier chip (RHD2164, Intan Technologies) through a USB 3.0 interface board (RHS2000, Intan Technologies) with Open Ephys GUI (graphical user interface) (https://github.com/open-ephys/plugin-GUI) under GNU/Linux. For cleaning prior to insertion, the probe was immersed in 1% (mass/vol) Tergazyme (Merk, cat. no. Z273287) in ultrapure water (Milli-Q, IQ 7000, Merck) solution for 30 min at 60 °C, washed with iso-propanol and rinsed with distilled water extensively. Channels exceeding twofold the median standard deviation (s.d.) were automatically removed with SpikeInterface (v0.7.6, https://github.com/SpikeInterface)[90]. Multi-unit spikes were detected by bandpass filtering (300–3,000 Hz) of raw recordings with a fourth order Butterworth filter and referencing to

the median signal (common median reference). Spike timestamps were then detected on the filtered signals as >5 s.d. above the noise floor, for which the standard deviation was calculated using the absolute median signal per channel in accordance with Donoho's rule[91,92]. To account for refractory periods, spike timestamps were included if they were ≥2 ms from another spike. Spikes with amplitudes at least 40 μV from the noise floor were included. For all subsequent analysis, active channels were selected for and defined as electrodes with ≥5 spikes min$^{-1}$ (0.083 Hz) across 10 min optogenetic stimulation periods. Active channels responding to optogenetic stimulation were defined as those with a ≥ 10% firing rate change from baseline recordings. For optogenetic response visualization, population event arrays were generated by summing binarized spike timestamps across all active channels. The resultant vector was summed using a discrete 1 ms sliding window and smoothened with a normalized 100 ms kernel. For synaptic inhibition (Extended Data Fig. 8m), baseline recordings were performed before a combination of 100 μM D-AP5 (NMDA receptor antagonist), 10 μM NBQX (AMPA receptor antagonist), 20 μM gabazine (SR-95531, GABA$_A$ receptor antagonist), 10 μM SCH-23390 (DRD1 antagonist) and 50 μM sulpiride$^{-/-}$ (DRD2 antagonist) synaptic blockers were added to aCSF and perfused into the bath recording chamber for at least 5 min, before 10 min optogenetic stimulations were recorded. For synaptic blocker applications, organoids responsive to optogenetic stimulation (>1 Hz firing rate) were included. All analysis was completed using custom Python pipelines that were made publicly accessible.

## Ventral midbrain progenitor injections
The 16-day-old ventral midbrain progenitors were thawed in a water bath until a small sliver of ice remains. A total of 500 μl room temperature DMEM/F12 with 20% KOSR (Knockout Serum Replacement; Thermo Fisher, cat. no. 10828028) and 1:100 ROCK inhibitor was added drop by drop on top of the cells. Cells were then transferred into 10 ml DMEM/F12 with 20% KOSR and 1:100 ROCK inhibitor and spun down at 500 ×g for 5 min. The supernatant was removed and cells were resuspended in exactly 1 ml medium and counted. Cells were then spun down again and resuspended in the required volume for injecting (20,000 cells per 100 nl medium). Organoids were injected using a Nanoinjector (Nanoject II, Drummond) and pulled glass capillaries (Drummond Capillaries for Nanojet II injectors), which were pulled using a micropipette puller (model P-97, Sutter Instrument). Forebrain and midbrain were distinguished by stereotypic morphology and only organoids with clear morphological features were used (Extended Data Fig. 9u). A total of 207 nl was injected in three pulses (69 nl each) in the fast injection mode. Prior to injection, organoids were transferred onto an empty plate and all medium was removed. After injection, organoids were not touched for 1 min to enable closure of the injection site. Organoids were then transferred back into 10 cm plates with medium and incubated without shaking for 1 day, before being transferred back to an orbital shaker.

## Virus production
Rabies virus was produced as previously described[93]. AAV was generated as described by previous protocols. In brief, 10 × 15 cm$^2$ dishes were transfected with plasmids carrying the capsid, helper and cargo genes according to the previously described method[94]. AAVs from the supernatant and cells were recovered, concentrated and iodixanol purified as previously described[90]. qPCR was used to titer the virus as described previously[95].

## Viral transduction
For individual organoids, a total of 1 × 10$^{10}$ viral genomes (vg) was used for transduction. For MISCOs, 3 × 10$^{10}$ vg were used for transduction. The viral titer may need to be significantly adjusted based on organoid size or transduction efficiency of the virus. Organoids were transferred into 24-well plates and were individually incubated with the

virus in 600 µl medium for 4 h. Subsequently, up to four organoids of the same group were transferred into 6 cm plates together with the medium containing the virus. Medium was then added to achieve a volume of 6 ml (individual organoids) or 7 ml (MISCOs). Organoids were transferred to an orbital shaker overnight. On the next day, organoids were transferred back into 10 cm plates containing 30 ml medium. For injection of helper virus for rabies tracing, the same set-up was used as for midbrain dopaminergic progenitor injections. A total of 20 nl AAVDJ-Synapsin-TVA-N2cG-EGFP-WPRE3-SV40pa ($3.58 \times 10^{12}$ vg ml$^{-1}$) were injected into the forebrain part of an organoid for region-specific transduction of neurons. A total of $1.5 \times 10^6$ IU rabies virus was used per organoid.

Forebrain and midbrain tissue were distinguished by stereotypic morphology (ventral midbrain tissue is smaller and has a more transparent appearance than the forebrain part of MISCOs; Extended Data Fig. 9u), and injection specificity was confirmed by counterstains for TH as a ventral midbrain marker. The following viruses have been used in this study: pAAV-Syn-Archon1-KGC-GFP-ER2 (gift from E. Boyden; Addgene viral prep 115892-AAV8; n2t.net/addgene:115892; RRID:Addgene_115892); pAAV-Syn-ChrimsonR-tdT (gift from E. Boyden; Addgene viral prep 59171-AAV1; n2t.net/addgene:59171; RRID:Addgene_59171); AAV-CAG-hChR2-H134R-tdTomato (gift from K. Svoboda; Addgene viral prep 115892-AAV8; n2t.net/addgene:28017; RRID:Addgene_28017); AAV2 (pAAVss_hsyn-GRAB-DA4.4): $3.61 \times 10^{13}$ vg ml$^{-1}$ (gift from B. Hengerer, Department of CNS Diseases Research, Boehringer Ingelheim); AAVDJ-Synapsin-TVA-N2cG-EGFP-WPRE3-SV40pa (generated in-house); and rabies CVS-N2c Strain ΔG Crimson (RV-CVS-N2c-ΔG-Crimson; generated in-house). The in-house rabies and AAVs have been produced according to previously published protocols (see the Virus Production section).

## Pharmacology

For pharmacological treatment, cocaine hydrochloride (Serobac) was added to organoid medium to achieve a concentration of 0.7 µM (IC50). Organoid medium was exchanged every 3 days, and cocaine hydrochloride was added 1 h before the medium exchange. Single uptake of 50 mg cocaine hydrochloride might result in concentrations of ~2.1–2.5 µM for a 70 kg person[57,96]. However, cocaine is highly unstable in vivo and is metabolized by plasma esterases and liver cholinesterases, which are components not present in brain organoids. Thus, the IC50 concentration and the treatment for 1 h were chosen to approximate cocaine exposure of an individual dose.

## Cryosectioning and immunostaining

Organoid tissue was rinsed once in PBS and fixed in 4% formaldehyde solution in 1× PBS for 4 h at room temperature. The tissue was then incubated for 1 day in 30% sucrose at 4 °C on a roller mixer. Tissue was subsequently transferred in a bed of OCT (optimal cutting temperature compound; Scigen, cat. no. 4586) and OCT was carefully swirled around the organoid. After 5 min incubation, organoids were transferred into cryomolds and transferred on a metal block on dry ice. Once a small ring of solidified (that is, frozen) OCT appears, the cryomolds were filled with OCT and were completely frozen. The resulting cryoblocks were then wrapped in aluminum foil and were transferred to an ultra-freezer with a temperature below −70 °C until cryosectioning. Frozen organoid tissue was sliced into 20 µm (regular organoids) or 30 µm (for better representation of axons) using a cryostat and collected on cryoslides. Sections were dried for at least 3 h before processing for immunostaining, or alternatively dried overnight and then stored at −20 °C.

## 2D immunohistochemistry

For immunohistochemical fluorescent labeling, slides were thawed and dried for 1 h at room temperature. Residual OCT was then washed off by washing for 5 min in PBS on an orbital shaker. Permeabilization and blocking were performed by gently dropping PBS containing 5% BSA and 0.3% TX100 (Sigma-Aldrich, cat. no. 93420) on the slides and incubating them at room temperature in a humidification chamber for 30 min. Permeabilization and blocking solution was sterile filtered and then frozen, and fresh aliquots were used for every staining round. Primary antibodies were added at desired dilutions in staining solution (5% BSA, 0.1% TX100 in PBS, sterile filtered and frozen until needed) and incubated at 4 °C overnight in a humidification chamber. On the second day, the slides were rinsed three times with PBS and then washed three times for 10 min each in PBS-T (PBS + 0.01% TX100) at room temperature on an orbital shaker. Secondary antibodies were added at a 1:500 dilution and slides were incubated for 2 h at room temperature in a humidification chamber. DAPI (2 µg ml$^{-1}$ in PBS) was added for 7 min, then slides were rinsed three times in PBS and then washed twice in PBS-T. The third and final washing step was then performed using only PBS without TX100. Coverslips were mounted using fluorescent mounting medium (DAKO, cat. no. S3023). After coverslip mounting, slides were stored at room temperature for at least 4 h, before using for microscopy or storing them at 4 °C. Slides were kept at 4 °C for long-term storage. A list of primary and secondary antibodies used, including dilution, is given in Supplementary Tables 2 and 3.

## Tissue clearing and 3D immunohistochemistry

The 2Eci tissue clearing was performed as previously described, with minor modifications[39]. In brief, organoids were fixed with 4% formaldehyde in PBS for 4 h at room temperature. For 3D immunohistochemistry, PBS-TxDB was prepared as follows: 10× PBS, 5% BSA and 2% TX100 were pre-mixed and topped up with distilled water to approximately 70% of the desired volume (for example, for 1 l PBS-TxDB, 100 ml 10× PBS, 50 g BSA and 20 ml TX100 were used and topped up to 700 ml with distilled water. Then 20% dimethylsulfoxide (DMSO) was added slowly under heavy stirring (if DMSO is added too fast it may result in irreversible BSA aggregation). The solution was filled up to 1 l with distilled water and sterile filtered. PBS-TxDB is stable at room temperature for at least several weeks, or at 4 °C for up to 1 year. Organoids were blotted and permeabilized on a rotor or shaking plate for 1–2 days in 10 ml PBS-TxDB at room temperature. Organoids (dependent on the size, between 3 and 6) were then transferred into 2 ml Eppendorf tubes containing primary antibody solution (for concentration, see antibody list) in PBS-TxDB (between 500 µl and 1 ml) and transferred to a rotor or shaking plate for 5 days. Organoids were then washed in 10 ml PBS-TxDB for 2 days (1× rinse, 3× PBS-TxDB exchange) and the same steps were repeated for the secondary antibodies. After the residual secondary antibodies were washed off, the organoids were washed in PBS for 1 h and then fixed in 4% formaldehyde solution for 30 min–2 h. Organoids were then ready to be dehydrated in a 1-propanol gradient (30%, 50%, 70%, 100%, 100%, 4–8 h per step). Given that there was no endogenous fluorescence that had to be maintained, pH adjustment of the 1-propanol was skipped and Milli-Q water was used instead of PBS as the water component. After dehydration, organoids were transferred into ethyl cinnamate and were ready for imaging as soon as they were completely transparent (1–3 h or overnight at room temperature). Cleared organoids should be stored at room temperature and should not be transferred to 4 °C, given that ethyl cinnamate crystallizes out at lower temperatures.

## Microscopy

Cell culture microscopy was performed using a Zeiss Axio Vert.A1 widefield microscope with the Axiocam ERc 5s camera (Zeiss). The 2D and 3D tissue clearing recordings were done using an Olympus Spinning Disk system based on the Olympus IX3 Series (IX83) inverted microscope. The system is equipped with a dual-camera Yokogawa W1 spinning disk using 405 nm, 488 nm, 561 nm and 640 nm lasers, and a Hamamatsu Orca Flash 4.0 camera for recording. The objectives used were ×10/NA 0.3 (air) with a working distance (WD) of 10 mm, ×10/NA 0.4 (air) WD

3.1 mm, ×20/NA 0.75 (air) WD 0.6 mm and ×40/NA 0.75 (air) WD 0.5 mm. For live imaging, the attached incubator set-up was used (37 °C, 5% $CO_2$). For high-magnification live imaging with air objectives, the additional ×3.2 magnification of the Orca Flash 4.0 camera was used.

For dopamine sensor live imaging, a Visiscope Spinning Disc Confocal (Visitron Systems) was used. This system is based on a Nikon Eclipse Ti E inverted microscope, and is equipped with a Yokogawa W1 spinning disc and an incubator set-up (used at 37 °C, 5% $CO_2$). Components are controlled by the Visiview software. The laser (488 nm, 200 mW) was used with the Andor Ixon Ultra 888 EMCCD camera (13 µm pixel, 1,024 × 1,024 pixels) or the PCO Edge 4.2 m sCMOS camera (6.5 µm pixel, 2,024 × 2,024 pixels). For stimulation, the same stimulation set-up was used as for extracellular recordings, but with a 630 nm filter.

### Image processing
Images were processed using the FIJI distribution of the open-source image processing application ImageJ (v1.53q).

### GCAMP recordings
Organoids were grown until day 130. GCAMP was recorded using an Olympus Spinning Disk system (see the Microscopy section). In brief, ×20 magnification and an exposure time of 65 ms per frame (continuous recording) were used for up to 6.5 min of continuous recording.

For trace extraction, recordings were scaled down from 2,048 × 2,048 pixels to 512 × 512 pixels. Trace extraction was performed using the open-source software package CaImAn (Flatiron Institute)[63]. Spike detection and event duration measurement were carried out using a custom code in Python. Event duration was measured as the width of peaks at 5% height.

### Statistical analysis
Statistical analysis was performed using GraphPad Prism 9.4.1 using one-way ANOVA and unpaired t-test or Welch's t-test (assuming a Gaussian distribution), or the Mann–Whitney test for multiple comparison groups and Wilcoxon signed-rank test (all two-sided) for paired statistics in normalized firing plots. The P value thresholds used are *$P \le 0.05$, **$P \le 0.01$, ***$P \le 0.001$ and ****$P \le 0.0001$. $P_{adj}$ represents the adjusted P value. All details on sample size, number of replicates, statistical tests and P values for each experiment are provided in the relevant figure legends. The sample size of organoids per experiment was estimated based on previous experience.

### Reproducibility
Given that human stem cell-derived tissues can be heterogeneous, a focus has been placed on the number of cell lines and organoids in key experiments. Images of organoids for which n numbers are provided are representative examples of non-displayed organoids that had similar phenotypes. Per default, one section per organoid was used for analysis. Unless otherwise specified, the default cell line used for experiments was H9 (WA09, WiCell).

### Reporting summary
Further information on research design is available in the Nature Portfolio Reporting Summary linked to this article.

### Data availability
Bulk and single-cell RNA-seq data generated in this study have been made available at NCBI Gene Expression Omnibus (GEO) under the accession number GSE219247. Data and other unique reagents or biological materials generated in this study are available from the lead contact, J.A.K. (juergen.knoblich@imba.oeaw.ac.at) in compliance with Material Transfer Agreements (MTA). An STL CAD file for the embedding mold negative is provided in Supplementary Data 1. Source data are provided with this paper.

### Code availability
Code used for analysis of data in this study has been deposited in a Github repository and is publicly available (https://github.com/bardylab/miscos_org_ephys).

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

## Acknowledgements
The authors thank all current and past members of the Knoblich laboratory and Vienna BioCenter (VBC) for technical expertise and manuscript feedback, especially J. Sidhaye, C. Li, S. Emtenani, F. Bonnay, E. Chatzidaki, C. Schmidt and I. Keisuke. The authors also thank the IMBA/IMP BioOptics facility for flow cytometry, microscopy services and discussions, particularly P. Pasierbek and A. Moreno Cencerrado; the IMBA IPSC Biobank and IMBA Stem Cell Core Facility (https://www.oeaw.ac.at/imba/scientific-facilities/stem-cell-core-facility) for generation of iPS cell lines, the Ethics and Biosafety Department for coordinating ethics approvals, the VBC Core Facilities (VBCF), particularly the HistoPathology facility and the VBCF NGS Unit for sequencing (www.viennabiocenter.org), and the Molecular Biology Service Department (cores.imp.ac.at/molecular-biology-service) for consultation and discussions; collaborators B. Hengerer and S. Zach at Boehringer Ingelheim for providing GRAB-DA2m plasmid and viral preps and for exciting discussions; S. J. Guzman for the establishment of the silicon probe set-up on the campus; and the VBC Campus Core Facilities who keep everything running. Work in the laboratory of J.A.K. is supported by the Austrian Federal Ministry of Education, Science and Research, the Austrian Academy of Sciences, the City of Vienna, a European Research Council (ERC) Advanced Grant (no. 695642) under the European Union's Horizon 2020 programme, the Austrian Academy of Sciences, the Austrian Science Fund (FWF), (Special Research Programme F7804-B and Stand Alone grants P 35680

and P 35369), and funding by the Austrian Lotteries. Work by D.R. is supported by the ERC Advanced Grant (no. 695642), the FWF P 35369 grant and the Austrian Lotteries. Work in the laboratory of M. Parmar was supported by funding from the New York Stem Cell Foundation, ERC Grant Agreement 771427, European Union-funded project NSC-Reconstruct (European Union, H2020, GA no. 874758, 563 2020-23) and the Swedish Research Council (2021-00661 and 3R: 2022-01-01–2024-12-31). K.I.R-R. and M.M.S. acknowledge funding through the EPSRC Centre for Doctoral Training in Neurotechnology (EP/L016737/1) and Rosetrees Trust. M.M.S. acknowledges funding from the UK Regenerative Medicine Platform Hub 'Acellular/Smart Materials–3D Architecture' (MR/R015651/1). Work in the laboratory of C.B. was supported by the Michael J Fox Foundation, the Hospital Research Foundation, the Shake it Up Foundation, the Neurosurgical Research Foundation and the Australian Research Council (FT230100138). A.F. acknowledges funding through the Swedish Research Council (2022-01432). C.F. acknowledges funding from the European Union's Framework Programme for Research and Innovation Horizon 2020 (2014-2020) under the Marie Curie Skłodowska grant agreement no. 847548.

## Author contributions

D.R. and J.A.K. conceived and planned the project and wrote the manuscript with support from all of the authors. D.R. performed experiments, and collected and analyzed data. C.K. performed experiments and analyzed data under the supervision of D.R. and J.A.K. RNA-seq analysis was performed by D.R. and M.N. Characterized cells for grafting experiments were provided by E.S. and A.F. with the help of M. Parmar. Electrophysiology experiments and analysis were performed by D.R., S.N.W. and M.Z. with the help of C.B. Calcium imaging and analysis were performed by D.R., M. Piber, M. Priouret and C.F. with the help of M.Z. Rabies virus tracing and analysis were done by B.D. The development of the PDMS molds was supported by K.I.R-R. with the help of M.M.S. Preparation and execution of scRNA-seq experiments were supported by I.M. Funding was acquired by J.A.K. and D.R.

## Competing interests

J.A.K. is an inventor on a patent describing cerebral organoid technology (European patent application no. EP2743345A1), and is a co-founder and member of the scientific advisory board of a:head bio AG. J.A.K. and D.R. are inventors on a patent application describing brain organoid fusion technology (patent application no. EP22177191.8). M.M.S, K.I.R.-R., D.R. and J.A.K. are inventors on a patent describing organoid technology (patent submission ID: GB2206768.0). M. Parmar is the owner of Parmar Cells that holds related intellectual property (US patent 15/093,927, 570 PCT/EP17181588), performs paid consultancy to Novo Nordisk AS and serves on the scientific advisory board for Arbor Biotechnologies. C.B. is an inventor on a patent about cell culture media for neuronal cell culture (BrainPhys) (international publication number: WO2014/172580A1). All other authors have no competing interests.

## Additional information

**Extended data** are available for this paper at https://doi.org/10.1038/s41592-023-02080-x.

**Correspondence and requests for materials** should be addressed to Jürgen A. Knoblich.

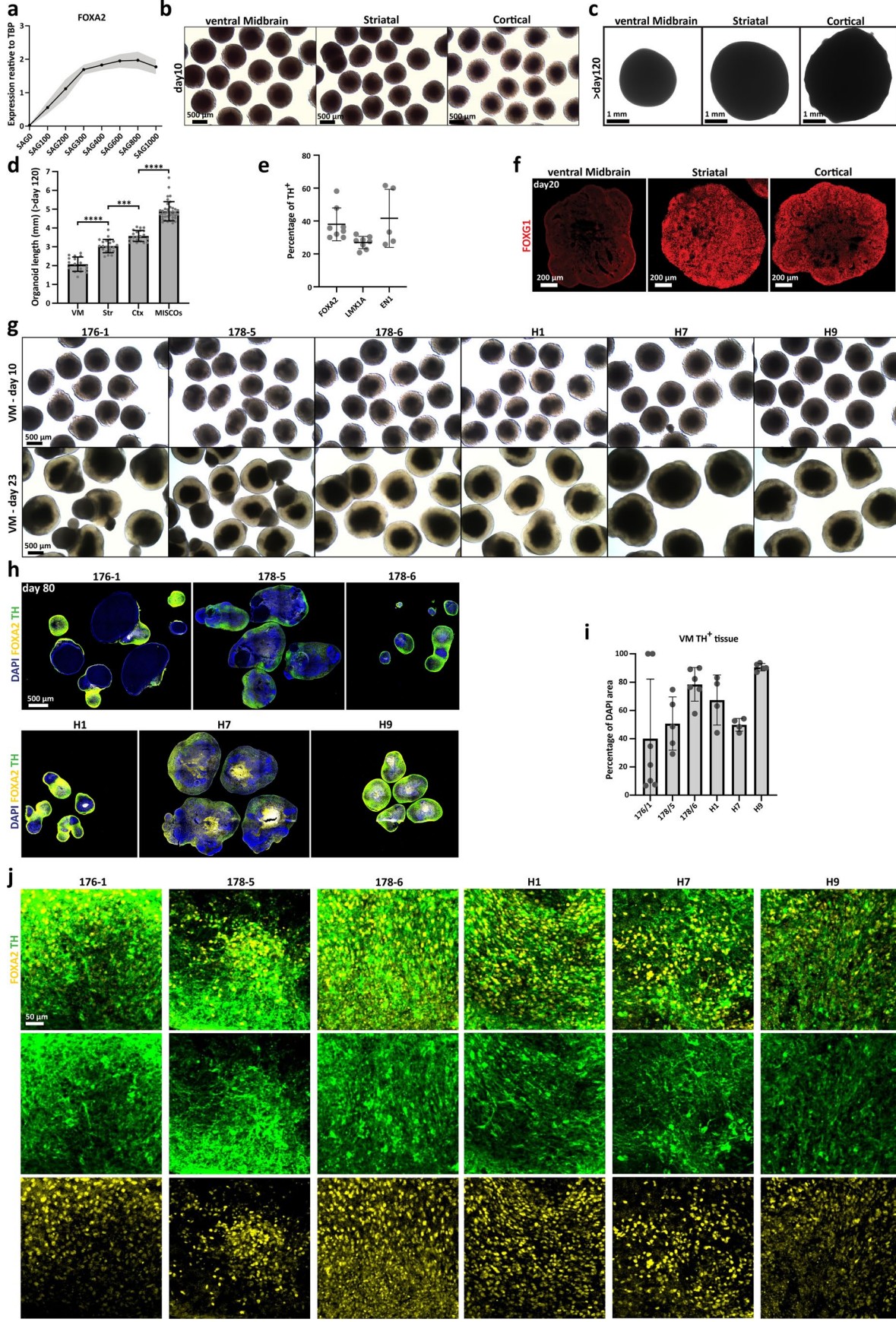

**Extended Data Fig. 1 | See next page for caption.**

**Extended Data Fig. 1 | VM patterning robustly induces dopaminergic neurogenesis in multiple cell lines. a**, qPCR analysis of organoids treated with a dose curve of the small molecule SAG as a replacement for SHH indicates appropriate floor plate induction at 300 nM on day 20 of organoid differentiation. 5 independent batches with 8-10 organoids per batch. Gray area indicates ±SEM. **b**, Representative images of 10 day-old VM, striatal and cortical patterned organoids. **c**, Representative images of VM, striatal and cortical organoids of >120 days. **d**, Quantification of VM, striatal, cortical and MISCO organoid length. n = 17|22|21 organoids of 2 batches per condition of >120 days (individual organoids) and 39 organoids of 6 batches (MISCOs). Statistical analysis was performed with a one-way ANOVA followed by Tukey's multiple comparisons test. $P_{adj}$ = <0.0001|0.0003|< 0.0001. Data show mean ± SD.

**e**, Quantification of the fraction of FOXA2/LMX1a/EN1 positive cells expressing TH. n = 8|8|5 positions of individual organoids at day 80. Data shown as mean ± SD. **f**, Day 20 cortical and striatal organoids express the forebrain marker FOXG1, which is absent in VM patterned organoids (representative images, similar results in n = 3-4 organoids). **g**, VM patterned organoids from 6 different cell lines (3 embryonic and 3 iPSCs) on day 10 and day 23. **h**, Co-expression of the mDA neuronal markers FOXA2 and TH indicate VM dopaminergic neurogenesis in all 6 cell lines on day 80 of differentiation. **i**, Quantification of TH$^+$ tissue as percentage of DAPI positive area of 6 different cell lines on day 80. n = 7|5|6|4|4|5 organoids of one batch per cell line. Data show mean ± SD. **j**, Magnified regions of TH and FOXA2-positive regions from 6 cell lines (n = 6 cell lines of one batch).

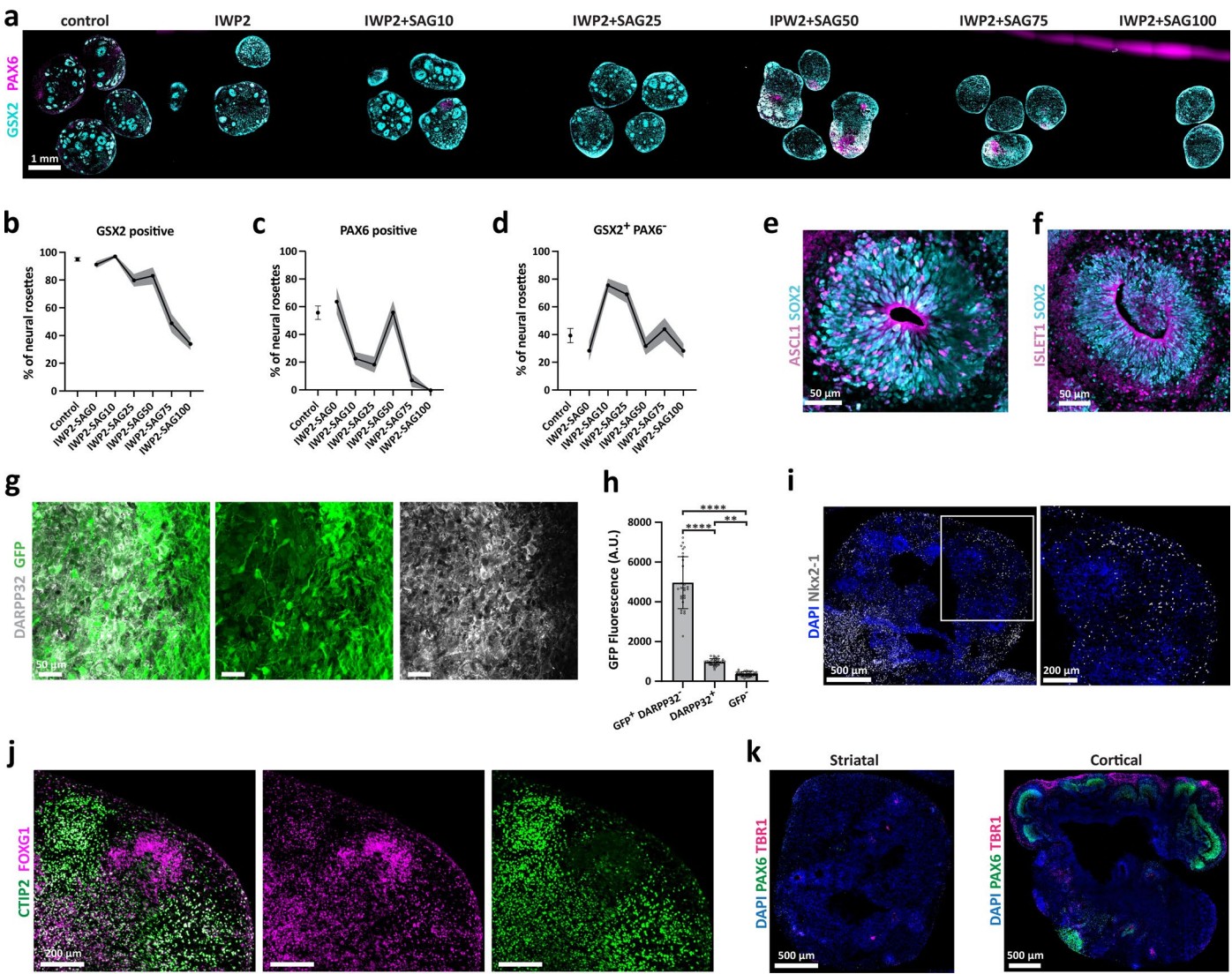

**Extended Data Fig. 2 | Low levels of SHH activation and Wnt inhibition are sufficient to induce LGE neuroepithelium and striatal neurogenesis. a**, Dose–response curve of SAG (0-100 nM) to enrich for GSX2+ LGE neuroepithelium in day 27 old organoids (representative batch, similar results in n = 10-15 organoids per condition of 3 independent batches). **b-d**, Quantification of GSX2+ (b) and PAX6+ (c), as well as GSX2+ & PAX6− neural rosettes after SAG dose curve in 27-day-old organoids. n = 10-15 organoids per condition of 3 independent batches. Gray areas visualize ±SEM. **e**, IWP-2−SAG10nM treated organoids display ASCL1 positive neural progenitors (SOX2+) (representative image, similar results in n = 4/4 organoids). **f**, IWP-2-SAG10nM treated organoids produce ISLET1 positive neurons (representative image, similar results in n = 4/4 organoids). **g,h**, DARPP32+ clusters express the subpallial marker DLX5 in a DLXi5/6-GFP

reporter line, but at lower levels than interneurons (DLXi5/6-GFP+, DARPP32−) on day 80 (representative image, similar results in n = 6/6 organoids). Statistical analysis was done with a one-way ANOVA followed by Tukey's multiple comparisons test. $P_{adj}$ = <0.0001|0.0017|< 0.0001. Data show mean ± SD. **i**, Striatal patterned organoids on day 60 display NKX2-1+ interneurons scattered through the organoid (representative image, similar results in n = 5/5 organoids). **j**, Striatal organoids are positive for FOXG1 and CTIP2 (representative image, similar results in n = 5/5 organoids). **k**, Striatal organoids do not have PAX6+ regions and are negative for the cortical neuron marker TBR1, unlike cortical organoids (representative image, similar results in 39 of 41 organoids of 6 different cell lines).

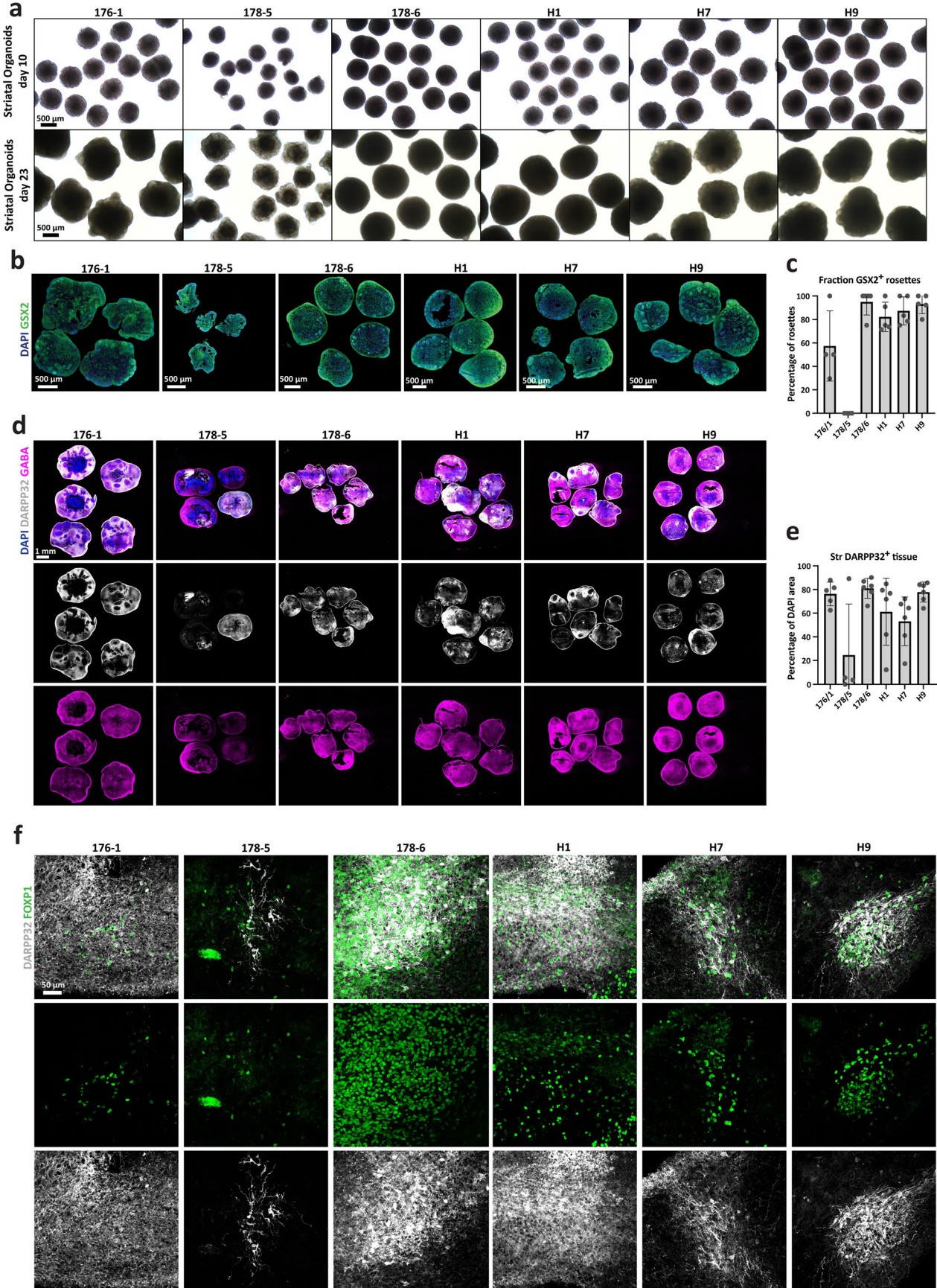

**Extended Data Fig. 3 | See next page for caption.**

**Extended Data Fig. 3 | Striatal patterning robustly induces striatal neurogenesis in multiple cell lines. a**, Organoids from 6 different cell lines (3 embryonic and 3 induced PSCs) treated with striatal patterning on day 10 and day 23 (one batch per cell line). **b**, Immunohistochemistry of day 33 striatal organoids for the LGE marker GSX2. **c**, Quantification of GSX2⁺ rosettes as percentage of all neural rosettes of individual organoids of 6 different cell lines on day 33. n = 4|4|5|5|5|5 organoids of one batch per cell line. **d**, 5 out of 6 cell lines robustly produce DARPP32⁺ neurons which co-localize with immunolabeling of GABA in day 80 organoids (cell line 178-5 being broadly negative). **e**, Quantification of DARPP32⁺ tissue as percentage of DAPI positive area of 6 different cell lines on day 80. n = 5|4|6|6|6 organoids of one batch per cell line. **f**, Clusters of DARPP32-positive neurons co-express the marker FOXP1 in all cell lines in day 80 organoids (n = 6 cell lines of one batch). Data shown as mean ± SD.

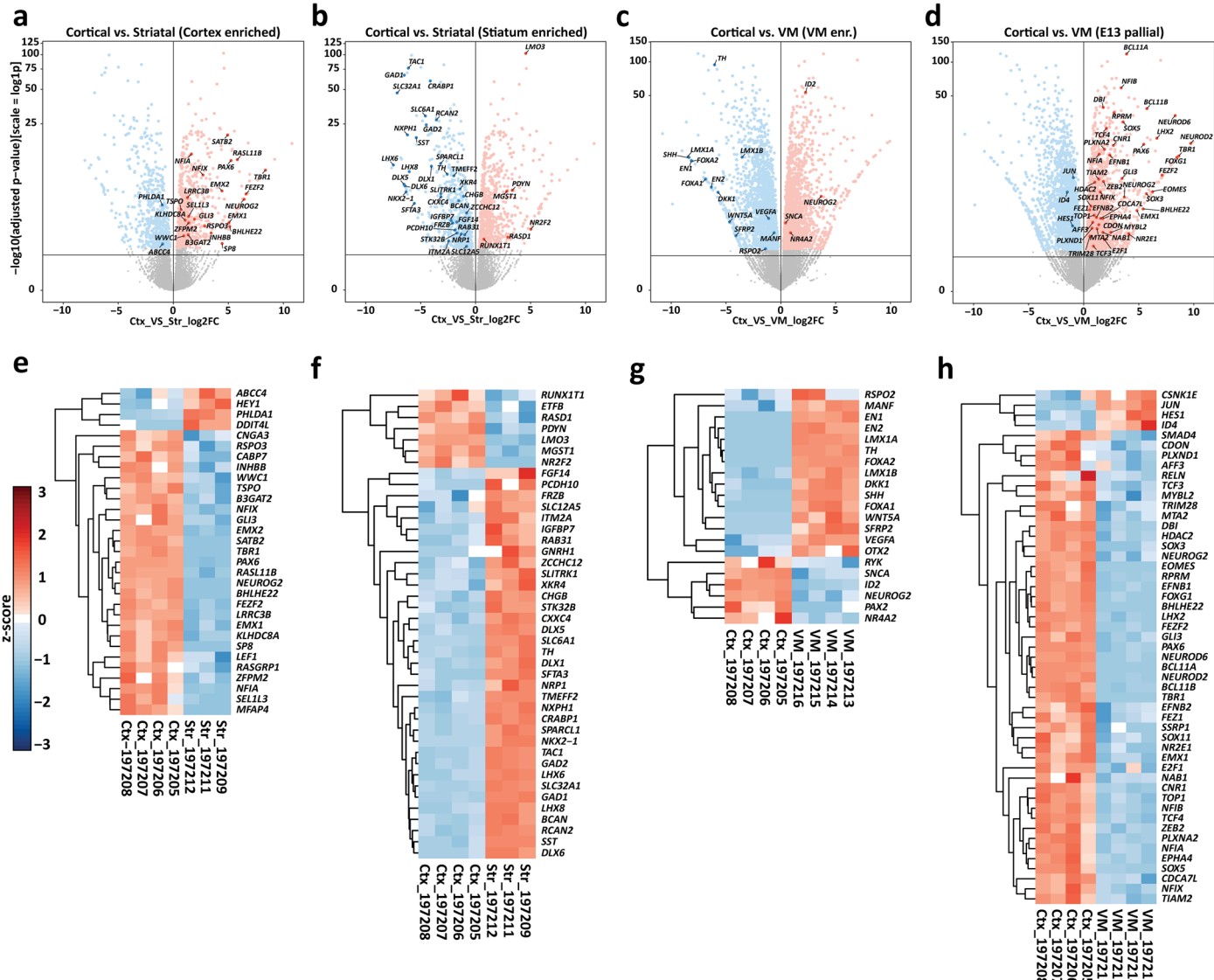

**Extended Data Fig. 4 | VM, striatal and cortical patterned organoids are enriched in corresponding brain region markers. a–h,** Volcano plots and corresponding relative gene expression heatmaps showing VM, striatal and cortical marker genes in differential expression analysis of cortical versus striatal and cortical versus VM organoids. Cortical organoids were enriched for cortical markers in comparison with striatal (**a, e**) and VM (**d,h**) tissues. **b, f,** Striatal organoids were enriched for striatal markers. **c, g,** VM organoids were enriched for mDA markers. p value was calculated using a Benjamin-Hochberg corrected Wald test using DESeq2.

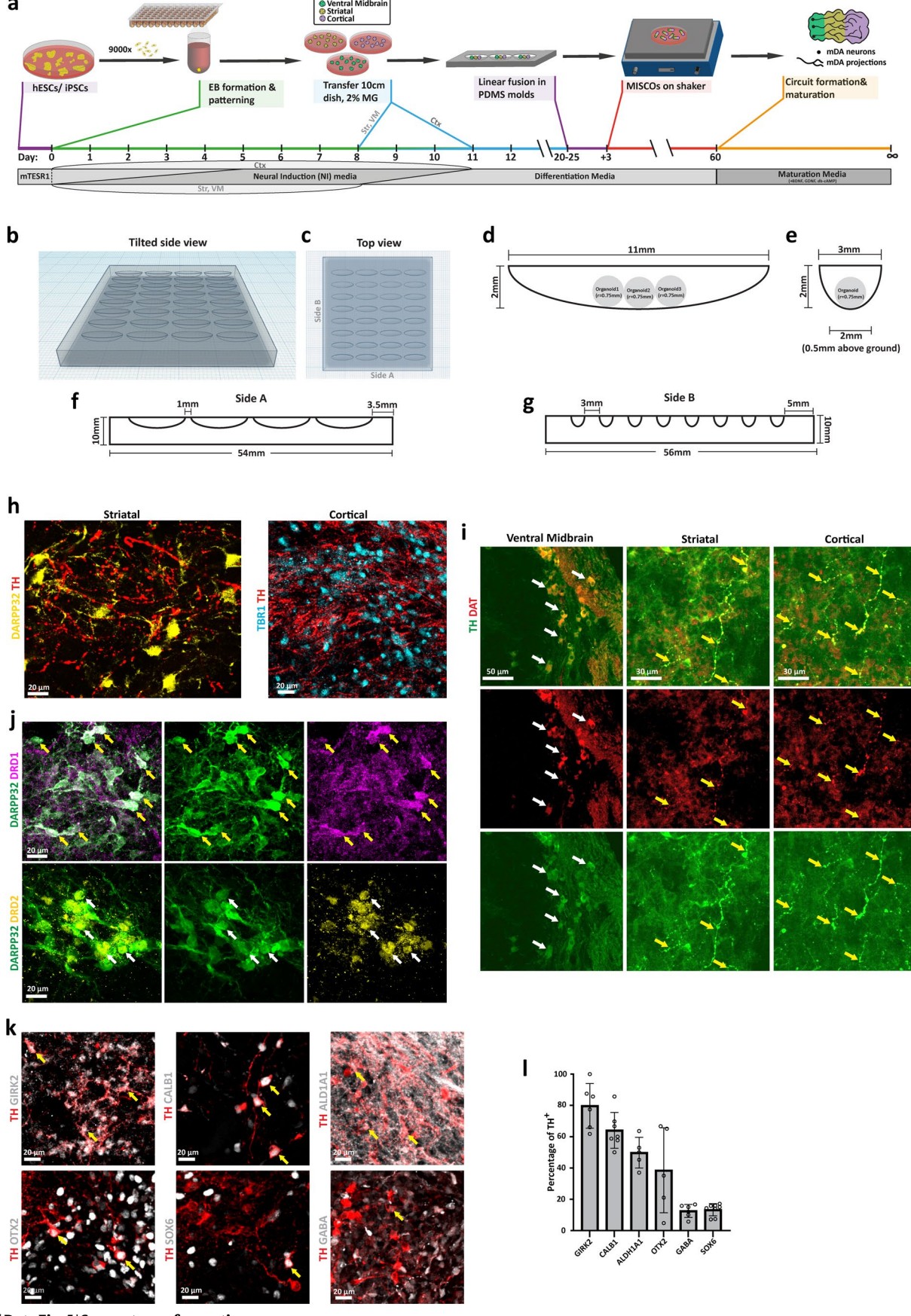

**Extended Data Fig. 5 | See next page for caption.**

**Extended Data Fig. 5 | PDMS molds allow fusion of organoids and the observation of dopaminergic innervation. a**, Schematic for the generation of MISCOs. **b-c**, Tilted side view and top view of PDMS embedding molds for linear fusions. **d-e**, size considerations for the fusion of three organoids. **f-g**, Side view of the PDMS embedding mold for MISCO generation. **h**, Cortical (TBR1⁺) and striatal (DARPP32⁺) tissue were innervated by dopaminergic axons (representative images, similar results in n = 8 MISCOs of 2 batches). **i**, Dopaminergic (TH⁺) neurons in the VM (white arrow) as well as their axons in striatum and cortex (yellow arrows) of day 90 MISCOs express the mature mDA marker Dopamine Transporter (DAT) on day 90 (representative images, similar results in n = 9/9 MISCOs). **j**, DARPP32⁺ neurons in 120-day-old MISCOs in the striatum expressed the striatal subtype markers DRD1 (yellow arrows) and DRD2 (white arrows) (representative images, similar results in n = 6-8 MISCOs of 2-3 batches). **k-l**, Dopaminergic neurons in MISCOs expressed the dopaminergic subtype markers GIRK2, CALB1, ALDH1A1, OTX2, GABA and SOX6 (n = 6|7|5|5|6|8 MISCOs of 3-4 batches). Yellow arrows: representative double-positive neurons. Data shown as mean ± SD.

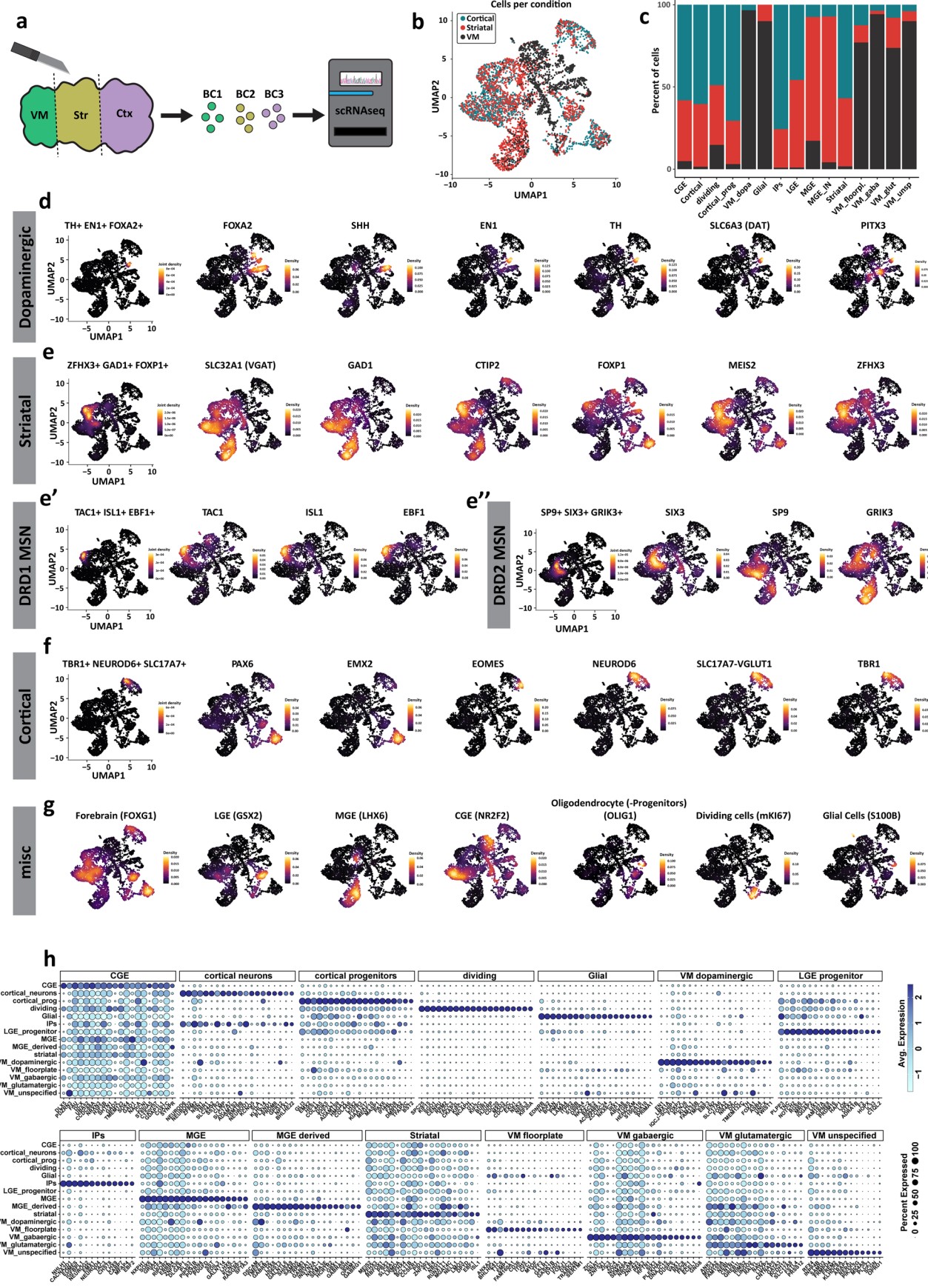

**Extended Data Fig. 6 | See next page for caption.**

**Extended Data Fig. 6 | scRNA-seq confirms neuronal populations associated with the dopaminergic system in MISCOs. a**, Schematic for barcoding of VM, striatum and cortex tissue for droplet based scRNA-seq using MULTI-seq. **b**, Plotting of UMAP with indication of tissue origin. **c**, Stacked bar plot showing barcode origin of clusters in percent. **d**, Density plots of floor plate (*FOXA2*, *SHH*) and dopaminergic neuron (*EN1*, *TH*, *DAT*, *PITX3*) as well as a *TH*, *EN1* and *FOXA2* joint density plot indicate mDA neuronal cluster identity. **e**, The striatal neuron cluster is GABAergic (*VGAT*, *GAD1*) and is positive for the striatal markers *CTIP2*, *FOXP1*, *MEIS2* and *ZFHX3*. The joint density plot of *ZFHX3*, *GAD1* and *FOX1* indicates striatal neuron cluster identity. **e'-e"**, The striatal cluster expresses markers of DRD1 medium spiny neurons (**e'**, *TAC1*, *ISL1*, *EBF1*) as well as DRD2 medium spiny neurons (**e"**, *SIX3*, *SP9*, *GRIK3*). **f**, Density plots for cortical progenitor markers (*PAX6*, *EMX2*), the intermediate progenitor marker *EOMES* and the cortical neuronal markers *NEUROD6*, *VGLUT1* and *TBR1*. **g**, Forebrain (cortical and GE patterned) clusters express the forebrain marker *FOXG1*. Clusters for LGE progenitors (*GSX2*) as well as MGE (*LHX6*) and CGE-derived (*NR2F2*) cells were observable. Additionally, clusters with the identities of oligodendrocyte (-progenitors) (*OLIG1*), dividing progenitors (*mKI67*) and glial cells (*S100B*) clusters were present. **h**, Dot plot of top marker genes expressed in at least 20% of cells in individual clusters ranked by p value, calculated by presto implementation of the Wilcoxon rank sum test and auROC analysis.

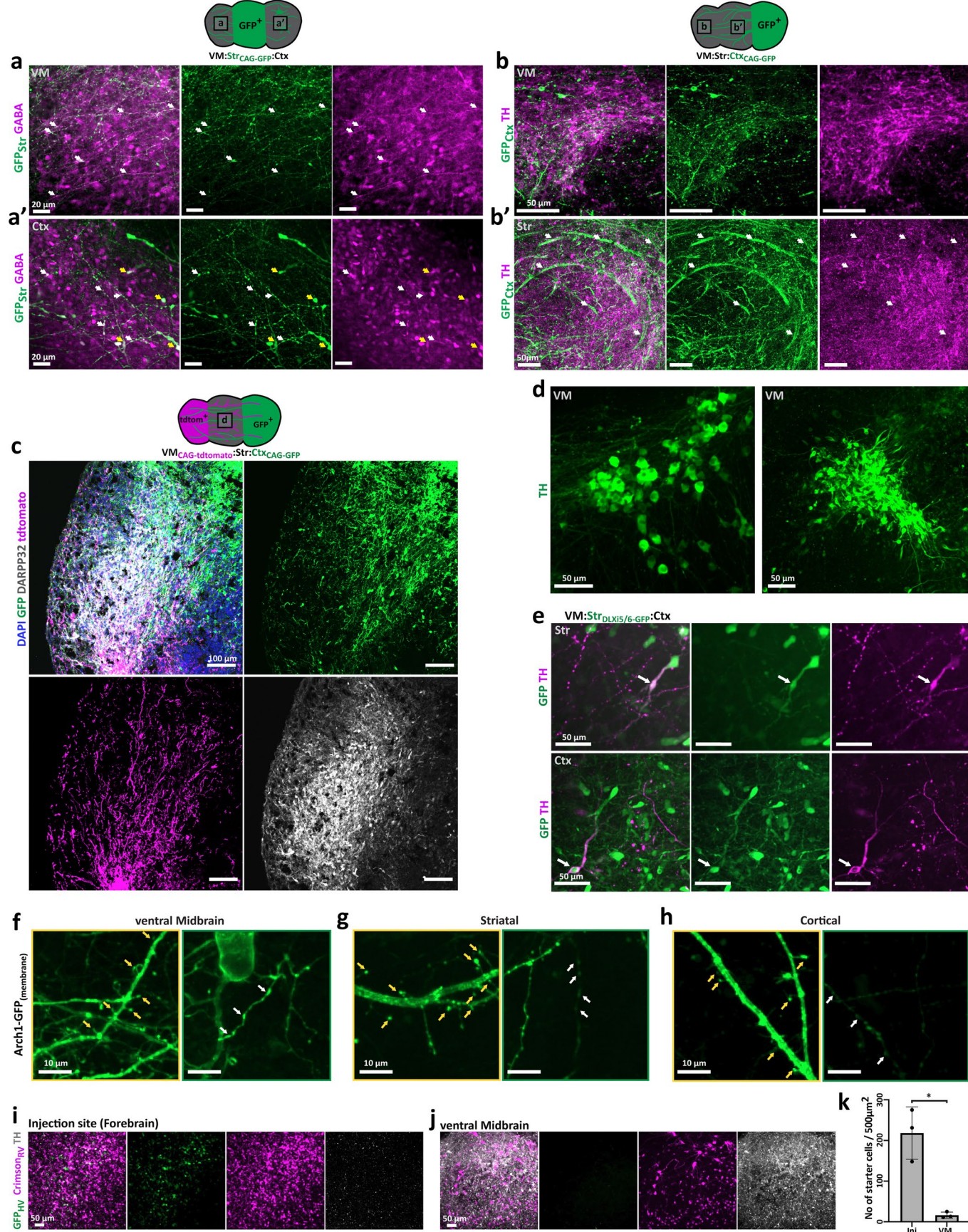

**Extended Data Fig. 7 | See next page for caption.**

**Extended Data Fig. 7 | MISCOs allow the study of reciprocal connections from cortical and striatal tissue. a**, Day 60 VM$_{WT}$-Str$_{CAG-GFP}$-Ctx$_{WT}$ MISCOs allow to study striatal innervation and show reciprocal GABA$^+$ innervation from striatal into VM (**a**) and cortical (**a'**) tissue (white arrows), indicating reciprocal connectivity between VM and striatal tissue. Additionally, migrated interneurons into cortical tissues could be observed (yellow arrows). (representative images, similar results in n = 8/8 60-day-old MISCOs). **b**, Day 60 VM$_{WT}$-Str$_{WT}$-Ctx$_{CAG-GFP}$ MISCOs display innervation from cortical tissue into both VM (**b**) as well as striatal tissue (**b'**), displaying cortical projection axon bundles (white arrows in **b'**) in the striatum as well as axonal innervation of mDA clusters in VM tissues (representative images, similar results in n = 6/6 60-day-old MISCOs). **c**, Striatal DARPP32$^+$ clusters were readily innervated from cortical and VM tissues in day 40 VM$_{CAG-tdTomato}$-Str$_{WT}$-Ctx$_{CAG-GFP}$ MISCOs (representative image, similar results in 6/6 organoids). **d**, Dopaminergic (TH$^+$) neurons often cluster together and display heterogeneous morphologies (representative images, similar results in n = 13/13 organoids of 3 batches). **e**, Co-labeling of *TH* and GFP in VM$_{WT}$-Str$_{DLXi5/6-GFP}$-Ctx$_{WT}$ MISCOs in striatal and cortical tissues allowed the observation of TH expressing interneurons (representative neurons of 2 recordings of 2 organoids). **f–h**, Putative dendritic spines (yellow arrows) and axonal boutons (white arrows) in Arch1-GFP transduced organoids in VM, striatal and cortical tissue. **i**, The injection site displayed HV (GFP$^+$) and RV (Crimson$^+$) double-transduced starter cells as well as monosynaptic, retrogradely transduced Crimson$^+$ GFP$^-$ cells (representative image, similar results in n = 3/3 organoids of one batch). **j**, In the VM, neurons were RV positive, but helper virus negative, indicating retrograde spreading (representative image, similar results in n = 3/3 organoids of one batch). **k**, Number of starter cells of 500 µm$^2$ of whole organoid z projections of both injection site vs. VM. n = 3 MISCOs. Paired t-test, *P* = 0.0399. Data shown as mean ± SD.

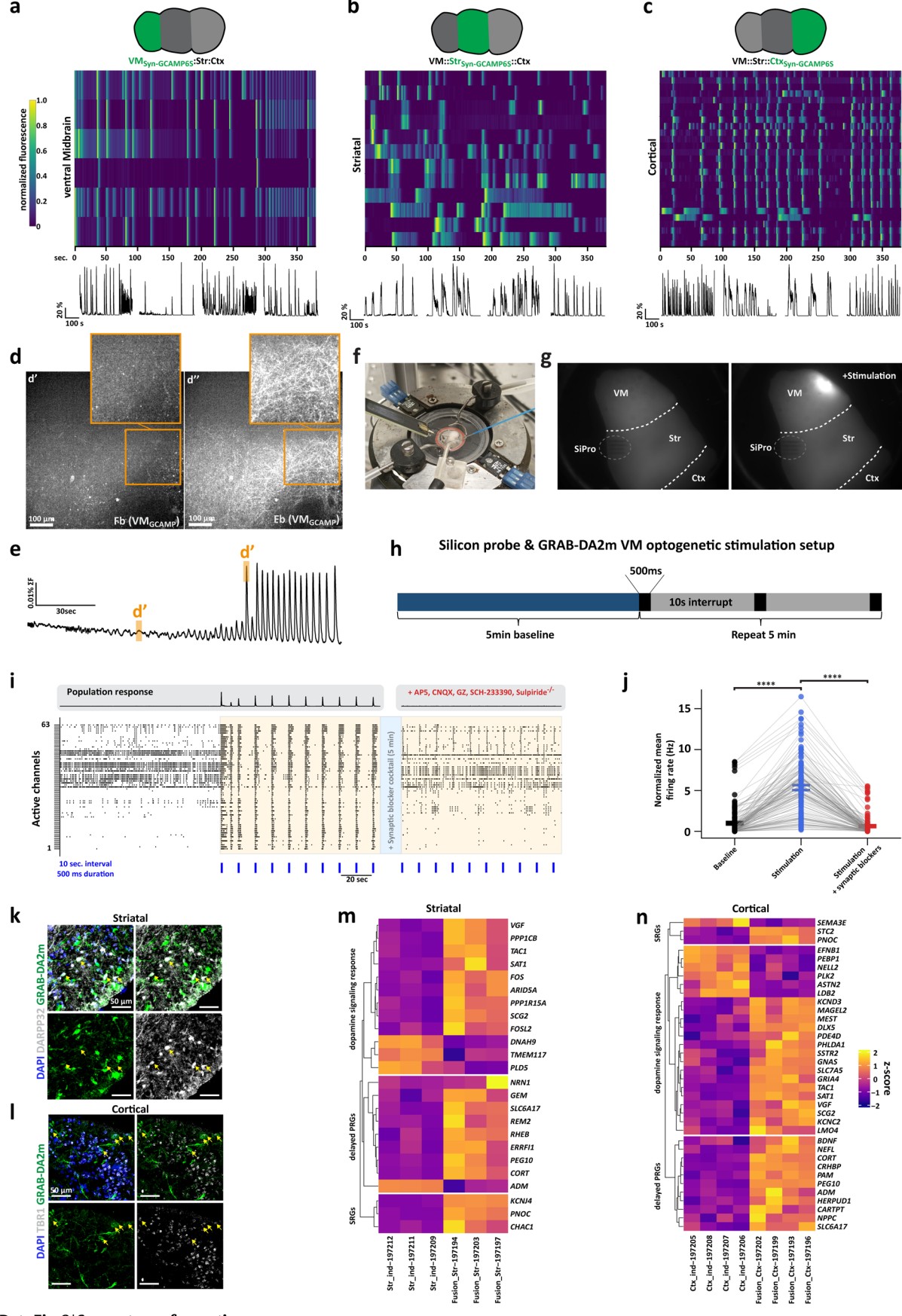

**Extended Data Fig. 8 | See next page for caption.**

**Extended Data Fig. 8 | MISCOs display functional dopaminergic innervation.**
**a-c**, Representative GCAMP recordings of 130-day-old MISCOs with Syn-GCAMP expression either in the VM, or striatal, or cortical tissue as well as selected traces of individual neurons (bottom) (similar results in n = 7/7 organoids per region) **d**, Ctx-Str-VM$_{Syn-GCAMP}$ MISCOs displayed calcium network events in VM derived projections in forebrain (Fb) tissue (**d'**: no synchronous event, **d"**: calcium network event) (similar results in n = 4/4 organoids of 2 batches). **e**, Cumulative fluorescence intensity over time of the recording from (**d**) displaying VM derived network events. Inserts indicate selected time points for panel **d'** and **d"**. **f**, Photography of a silicon neural probe set-up with optogenetic stimulation with mounted fused organoid. Organoids in this experiment were transduced with the optogenetic vector AAV-RG.AAV-CAG-hChR2-H134R-tdTomato. **g**, Widefield images before (left) and during stimulation (right) of the VM tissue while recording from striatal tissue. SiPro…Silicon Probe. **h**, Stimulation set-up. After baseline recording, the VM side of a MISCO was stimulated for 500 ms every 10 sec. **i**, Representative active channel raster

plots in forebrain tissue with baseline (left), VM stimulation (middle) and VM stimulation after application of synaptic blockers (right). 460-nm LED light pulses set at an interval of 10 seconds with a 500-millisecond duration (blue). After a 5 min incubation with a cocktail of synaptic blockers (D-AP5, CNQX, Gabazine, SCH-23390, Sulpiride$^{-/-}$), population responses in forebrain were absent during stimulation of VM. **j**, Normalized mean firing rate in MISCOs before and after light application, and repeated with synaptic blocker application (n = 127 active channels from 2 organoids). $P_{adj}$ = <0.0001|< 0.0001. Samples were analyzed with one-way ANOVA followed by Tukey's multiple comparisons test. **k-l**, AAV2$^-$pAAVss-hSyn-GRAB-DA2m transduces striatal (DARPP32$^+$ in striatal tissue, yellow arrows) as well as cortical (TBR1$^+$ in cortical tissue) neurons (yellow arrows: double-positive cells). **m-n**, Heatmap of activity regulated response genes and dopamine signaling response genes. The majority of activity-regulated genes were upregulated in RNA-seq of striatal and cortical tissues of separated MISCOs on day 60 in comparison to day 60 individual organoids. PRGs… primary response genes. SRGs… secondary response genes.

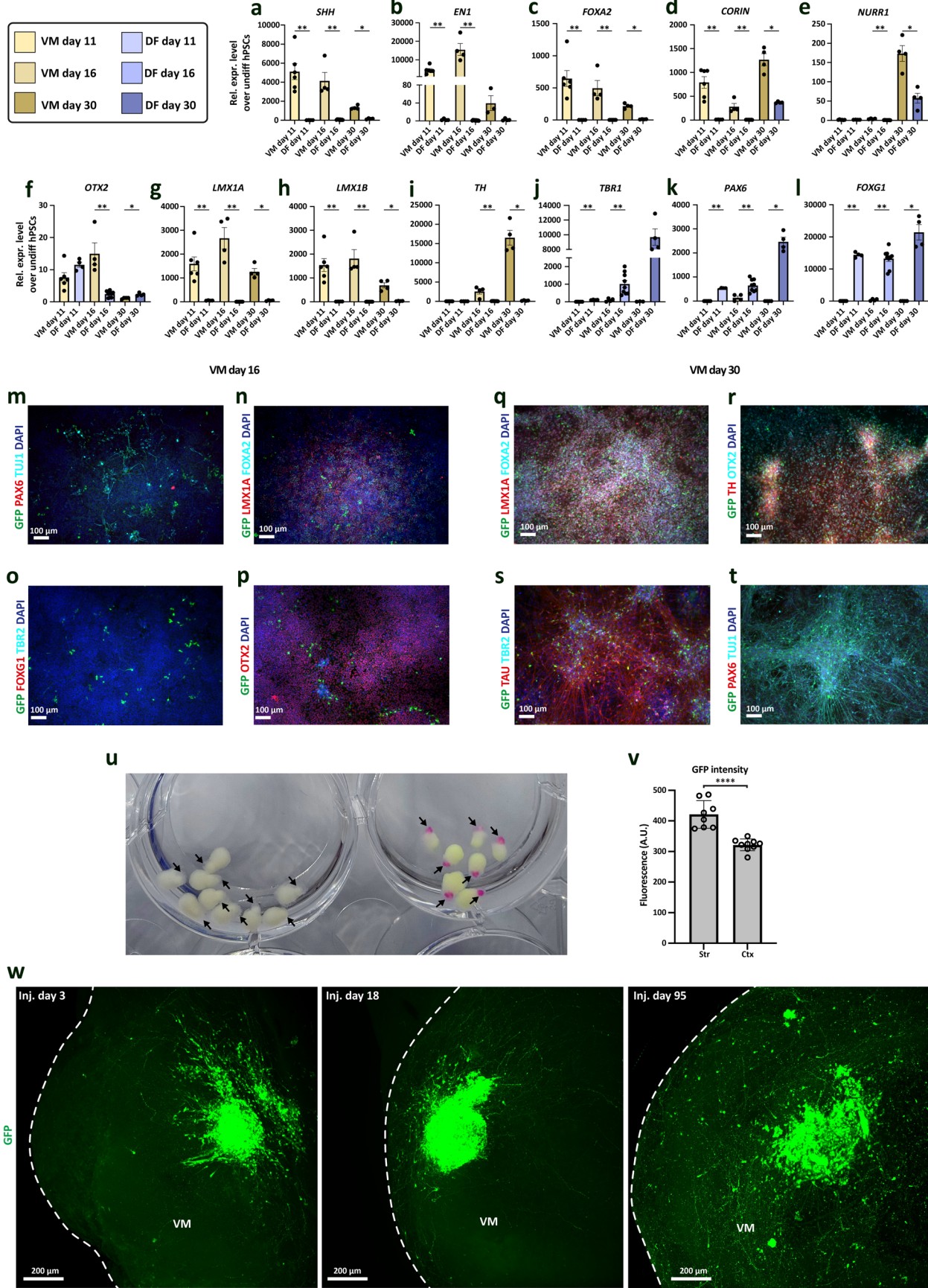

**Extended Data Fig. 9 | See next page for caption.**

**Extended Data Fig. 9 | Quality control for VM progenitors for grafting.**
**a-e**, Quality control for mDA neuronal differentiation as for dopaminergic cell therapy. mDA differentiation expressed the mDA progenitor markers *SHH*, *EN1*, *FOXA2*, *CORIN* and *NURR1*. mDA differentiation in brown, dorsal forebrain (DF) comparison in blue. **f-i**, the mDA markers *OTX2*, *LMX1A*, *LMX1B* and *TH* were highly expressed in VM in comparison to DF. **j-l**, The dorsal forebrain markers *TBR1*, *PAX6* and *FOXG1* were absent in the mDA differentiation. **a-l**, Significance was calculated using a two-sided Mann–Whitney test (n = 4-9 individual wells of one differentiation). Data shown as mean ± SEM. p values: **a**, 0.0095|0.0028|0.0286, **b**, 0.0095|0.0028, **c**, 0.0095|0.0028|0.0286, **d**, 0.0095|0.002|0.0286, **e**, 0.0028|0.0286, f, 0.0028|0.0286, **g**, 0.0095|0.0028|0.286, **h**, 0.0095|0.0028|0.0286, **i**, 0.0028|0.286, **j**, 0.0095|0.0028, **k**, 0.0095|0.0028|0.0286, **l**, 0.0095|0.0028|0.0286. **m-p**, Immunofluorescent staining of mDA differentiation on day 16 with mDA markers (LMX1A, FOXA2, OTX2), forebrain markers (PAX6, FOXG1 and TBR2) and

the neuronal marker TUJ1. **q-t**, Immunofluorescent labeling of mDA differentiation on day 30 with mDA neuronal markers (LMX1A, FOXA2, OTX2, TH), forebrain markers (PAX6 and TBR2) and the neuronal marker TUJ1 and TAU. **u**, Image of day 120 MISCOs in 6 well plates. Stereotypic morphologies in WT MISCOs (left) allow the identification of the VM tissue (black arrows), which was additionally tested by MISCO fusions containing CAG-tdTomato expression in the VM organoid derived tissue (black arrows). **v**, Quantification of GFP fluorescence in striatal and cortical tissue 95 days after injection. Statistical significance was calculated using an unpaired two-sided t-test. $P < 0.0001$. n = 8|9 regions of one differentiation/injection batch. Data shown as mean ± SD. **w**, Between day 3 and day 95 after injection in 60-day-old MISCOs, mDA grafts mature and innervation increases (representative images of 3D IHC and 2Eci tissue-cleared MISCOs, similar results in n = 4-5 organoids per timepoint).

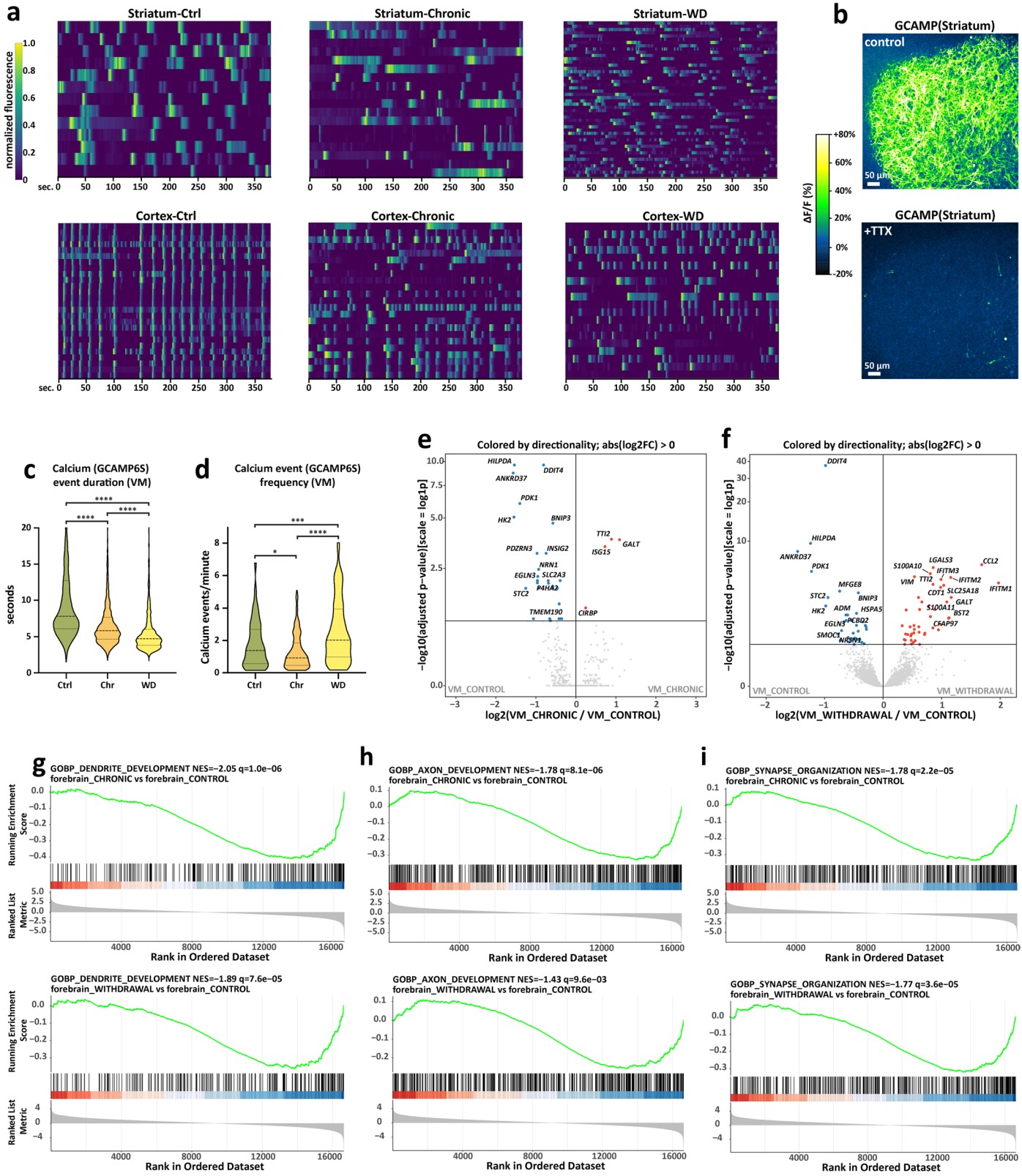

**Extended Data Fig. 10 | Cocaine treatment of MISCOs. a**, Representative GCAMP recordings of 130-day-old MISCOs with Syn-GCAMP expression either in the cortical or striatal tissue in control (left), chronic (middle) and withdrawal (right) condition. **b**, Representative GCAMP-recording of a MISCO with Syn-GCAMP in striatal tissue before (control) and after TTX application (0.5 μM), visualized as cumulative neuronal activity in a 6.5 min recording and displayed as ΔF/F. **c**, Calcium event duration of VM neurons in control, chronic and withdrawal conditions. 876|2058|1068 individual calcium events of 7-17 organoids. All $P_{adj}$ < 0.0001. **d**, Analysis of calcium event frequency of

VM (n = 100|278|115) neurons in control, chronic and withdrawal conditions. $P_{adj}$ = 0.0341|0.002|< 0.0001. Data shown as mean ± SD. Samples were analyzed by one-way ANOVA followed by Tukey's multiple comparisons test (**c,d**). **e-f**, Volcano plots of differentially expressed genes in VM tissue in chronic versus control (**e**) and withdrawal versus control (**f**). *P*-value was calculated using a Benjamin-Hochberg corrected Wald test using DESeq2. **g-i**, GSEA analysis of three GO terms associated with neural circuit formation (dendrite development, axon development and synapse organization) in forebrain chronic vs control (top) and forebrain withdrawal versus control (bottom).

# Reporting Summary

## Statistics

For all statistical analyses, confirm that the following items are present in the figure legend, table legend, main text, or Methods section.

| n/a | Confirmed | |
|---|---|---|
| ☐ | ☒ | The exact sample size (*n*) for each experimental group/condition, given as a discrete number and unit of measurement |
| ☐ | ☒ | A statement on whether measurements were taken from distinct samples or whether the same sample was measured repeatedly |
| ☐ | ☒ | The statistical test(s) used AND whether they are one- or two-sided *Only common tests should be described solely by name; describe more complex techniques in the Methods section.* |
| ☐ | ☒ | A description of all covariates tested |
| ☐ | ☒ | A description of any assumptions or corrections, such as tests of normality and adjustment for multiple comparisons |
| ☐ | ☒ | A full description of the statistical parameters including central tendency (e.g. means) or other basic estimates (e.g. regression coefficient) AND variation (e.g. standard deviation) or associated estimates of uncertainty (e.g. confidence intervals) |
| ☐ | ☒ | For null hypothesis testing, the test statistic (e.g. *F*, *t*, *r*) with confidence intervals, effect sizes, degrees of freedom and *P* value noted *Give P values as exact values whenever suitable.* |
| ☒ | ☐ | For Bayesian analysis, information on the choice of priors and Markov chain Monte Carlo settings |
| ☒ | ☐ | For hierarchical and complex designs, identification of the appropriate level for tests and full reporting of outcomes |
| ☒ | ☐ | Estimates of effect sizes (e.g. Cohen's *d*, Pearson's *r*), indicating how they were calculated |

*Our web collection on statistics for biologists contains articles on many of the points above.*

## Software and code

Policy information about availability of computer code

| Data collection | Commercial imaging software associated with Zeiss (Zen Blue), Olympus IX3 series and Visiscope spinning disk system was used. For Silicon Probe recordings, the software Open Ephys GUI.<br><br>For Bulk RNAseq, STAR 2.7.7a and DESeq2 v1.36.0 as well as Voxhunt v1.0.1 were used.<br><br>Exact usage of data collection can be found in the Methods section.<br><br>PDMS molds were designed in Tinkercad (www.Tinkercad.com) and sliced using the slicer software XYZ print 1.4.0. |
|---|---|
| Data analysis | Images were processed using the FIJI distribution of the open-source image processing application ImageJ (version 1.53q).<br><br>GCAMP trace extraction was performed using the open-source software package CaImAn (Flatiron Institute). Spike detection and event duration were performed using a custom code in Python 3.9.<br><br>Statistical analysis was performed using GraphPad Prism 9.4.1.<br><br>(sc)RNAseq analysis was performed using R Studio (2022.07.0 Build 548 and newer) and Cellranger count v7.0.0 and Seurat v4.1.1 were used.<br><br>Details of exact usage as well as data availability is provided in the Methods section. |

Code used for analysis of data in this study have been deposited in a Github repository and is publicly available (https://github.com/bardylab/miscos_org_ephys).

For manuscripts utilizing custom algorithms or software that are central to the research but not yet described in published literature, software must be made available to editors and reviewers. We strongly encourage code deposition in a community repository (e.g. GitHub). See the Nature Portfolio guidelines for submitting code & software for further information.

## Data

Policy information about availability of data

All manuscripts must include a data availability statement. This statement should provide the following information, where applicable:
- Accession codes, unique identifiers, or web links for publicly available datasets
- A description of any restrictions on data availability
- For clinical datasets or third party data, please ensure that the statement adheres to our policy

Bulk and single cell RNA-seq data generated in this study have been made available at NCBI Gene Expression Omnibus (GEO) under the accession number GSE219247. Data and other unique reagents or biological materials generated in this study are available from the lead contact Dr. Juergen Knoblich (juergen.knoblich@imba.oeaw.ac.at) upon reasonable request and in compliance with Material Transfer Agreements (MTA). A STL CAD file for the embedding mold negative will be provided in the supplements.

## Human research participants

Policy information about studies involving human research participants and Sex and Gender in Research.

| | |
|---|---|
| Reporting on sex and gender | n/a |
| Population characteristics | n/a |
| Recruitment | n/a |
| Ethics oversight | n/a |

Note that full information on the approval of the study protocol must also be provided in the manuscript.

# Field-specific reporting

Please select the one below that is the best fit for your research. If you are not sure, read the appropriate sections before making your selection.

☒ Life sciences    ☐ Behavioural & social sciences    ☐ Ecological, evolutionary & environmental sciences

For a reference copy of the document with all sections, see nature.com/documents/nr-reporting-summary-flat.pdf

# Life sciences study design

All studies must disclose on these points even when the disclosure is negative.

| | |
|---|---|
| Sample size | No predetermined sample size calculations were performed and sample size was determined by common practice in the field and based on previous studies in the field (Lancaster et al., Nature 2013, Lancaster et al., Nature Biotechnology 2017, Bagley et al., Nature Methods 2017, Esk&Lindenhofer et al., Science 2020, Eichmüller et al., Science 2022) |
| Data exclusions | No data which met the quality criteria of the designed experiments (e.g. read count in RNAseq) was excluded in this study. |
| Replication | All key experiments were successfully replicated at least twice and number of replicates are mentioned in figure legends. |
| Randomization | Organoids were randomly selected from organoid batches for each type of experiment. |
| Blinding | This study involved unbiased analysis and quantification for immunostaining, functional activity and gene expression data sets. Analysis was, where possible, performed on raw data by inherently unbiased pipelines. There was no expected outcome prior to any analysis, thus blinding was not performed. |

# Reporting for specific materials, systems and methods

We require information from authors about some types of materials, experimental systems and methods used in many studies. Here, indicate whether each material, system or method listed is relevant to your study. If you are not sure if a list item applies to your research, read the appropriate section before selecting a response.

## Materials & experimental systems

| n/a | Involved in the study |
|---|---|
| ☐ | ☒ Antibodies |
| ☐ | ☒ Eukaryotic cell lines |
| ☒ | ☐ Palaeontology and archaeology |
| ☒ | ☐ Animals and other organisms |
| ☒ | ☐ Clinical data |
| ☒ | ☐ Dual use research of concern |

## Methods

| n/a | Involved in the study |
|---|---|
| ☒ | ☐ ChIP-seq |
| ☒ | ☐ Flow cytometry |
| ☒ | ☐ MRI-based neuroimaging |

## Antibodies

Antibodies used

Primary Antibodies:
Species Antigen Producer Cat# Dilution used in 2D*
Mouse ALDH1A1 (Clone B-5) Santa Cruz Biotech sc-374149 1:50
Rabbit Calbindin Aves lab/Swant CD38a D-28K 1:1000
Rat CTIP2/ BCL11b Abcam ab18465 1:300
Rabbit DARPP32 (Clone EP720Y) Abcam ab40801 1:100
Goat DARPP32 R&D systems AF6259 1:100
Rat DAT (Clone DAT-NT) Millipore MAB369 1:1000
Goat DLX2 Santa Cruz Biotech sc-18140 1:100
Sheep DLX5 R&D Systems AF6710 1:100
Rabbit DRD1 Abcam ab20066 1:400
Rabbit DRD2 Millipore AB5084P 1:100
Rabbit DsRed/tdtomato/Crimson Clontech 632496 1:250
Rabbit EN2 LSBio LS-B9057-200 1:100
Goat FOXA2 R&D Systems AF2400 1:300
Rabbit FoxG1 Abcam ab18259 1:200
Rabbit FOXP1 Abcam AB16645 1:200
Rabbit GABA Sigma-Aldrich a2052 1:200
Mouse GAD67 (Clone 1G10.2) Millipore MAB5406 1:200
Chicken GFP Aves Labs GFP-1020 1:500
Rabbit GIRK2 (Kir3.2) Alomone Labs APC-006 1:400
Rabbit GSX2/GSH Millipore ABN162 1:100
Rabbit ISL-1 Abcam AB20670 1:100
Rabbit LMX1a Sigma-Aldrich AB10533 1:100
Rabbit MASH1/ASCL1 Abcam AB74065 1:100
Rabbit Nkx2.1/Thyroid (TTF1)
 (Clone EPR5955(2)) Epitomics 6594-1 1:1000
Goat OTX2 R&D Systems AF1979 1:100
Mouse PAX6 (Clone AD2.38) Abcam ab78545 1:100
Sheep PAX6 R&D Systems AF8150 1:200 of 100 μl reconstitute
Rabbit RFP/tdtomato/Crimson Abcam ab62341 1:100
Rabbit SOX2 Abcam ab97959 1:600
Goat SOX2 R&D Systems AF2018 1:100
Rabbit SOX6 Abcam ab30455 1:500
Rabbit TBR1 Abcam ab31940 1:300
Rabbit TH Abcam ab112 1:300
Sheep TH Abcam ab113 1:400

Secondary Antibodies:
Species Anti- Fluorophore Producer Cat#
Donkey Mouse AF488 Invitrogen A21202
Donkey Mouse AF568 Invitrogen A10037
Donkey Mouse AF647 Invitrogen A31571
Donkey Rabbit AF488 Invitrogen A21206
Donkey Rabbit AF568 Invitrogen A10042
Donkey Rabbit AF647 Invitrogen A31573
Donkey Rat AF488 Invitrogen A21208
Donkey Rat AF647 Jackson ImmunoResearch 712-605-150
Donkey Goat AF488 Invitrogen A11055
Donkey Goat AF568 Invitrogen A11057
Donkey Goat AF647 Invitrogen A21447
Donkey Sheep AF488 Invitrogen A11015
Donkey Sheep AF568 Invitrogen A21099
Donkey Sheep AF647 Jackson ImmunoResearch 713-605-147

Donkey Chicken AF488 Jackson ImmunoResearch 703-545-155
Donkey Chicken AF647 Jackson ImmunoResearch 703-605-155

| Validation | Mouse ALDH1A1 (Clone B-5) Santa Cruz Biotech sc-374149: Validated by the company and used in 26 scientific literatures.<br>Rabbit Calbindin Aves lab/Swant CD38a D-28K: Validated by the company and used in one scientific literature.<br>Rat CTIP2/ BCL11b Abcam ab18465: Validated by the company and used in 718 scientific literatures.<br>Rabbit DARPP32 (Clone EP720Y) Abcam ab40801: Validated by the company and used in 96 scientific literatures.<br>Goat DARPP32 R&D systems AF6259: Validated by the company and used in one scientific literature.<br>Rat DAT (Clone DAT-NT) Millipore MAB369: Validated by the company.<br>Goat DLX2 Santa Cruz Biotech sc-18140: Validated by the company and used in 6 scientific literatures.<br>Sheep DLX5  R&D Systems AF6710: Validated by the company.<br>Rabbit DRD1 Abcam ab20066: Validated by the company and used in 47 scientific literatures.<br>Rabbit DRD2 Millipore AB5084P: Validated by the company.<br>Rabbit DsRed/tdtomato/Crimson Clontech 632496: Validated by the company and used in 2004 scientific literatures.<br>Rabbit EN2 LSBio LS-B9057-200: Validated by the company and used in one scientific literature.<br>Goat FOXA2 R&D Systems AF2400: Validated by the company and used in 66 scientific literatures.<br>Rabbit FoxG1 Abcam ab18259: Validated by the company and used in 126 scientific literatures.<br>Rabbit FOXP1 Abcam AB16645: Validated by the company and used in 97 scientific literatures.<br>Rabbit GABA Sigma-Aldrich a2052: Validated by the company.<br>Mouse GAD67 (Clone 1G10.2) Millipore MAB5406: Validated by the company.<br>Chicken GFP Aves Labs GFP-1020: Validated by the company and used in 747 scientific literatures.<br>Rabbit GIRK2 (Kir3.2) Alomone Labs APC-006: Validated by the company and used in 187 scientific literatures.<br>Rabbit GSX2/GSH Millipore ABN162: Validated by the company.<br>Rabbit ISL-1  Abcam AB20670: Validated by the company and used in 83 scientific literatures.<br>Rabbit LMX1a Sigma-Aldrich AB10533: Validated by the company.<br>Rabbit MASH1/ASCL1 Abcam AB74065: Validated by the company and used in 36 scientific literatures.<br>Rabbit Nkx2.1/Thyroid (TTF1) (Clone EPR5955(2)) Epitomics 6594-1: Validated by the company and used in 2 scientific literatures.<br>Goat OTX2 R&D Systems AF1979: Validated by the company and used in 81 scientific literatures.<br>Mouse PAX6 (Clone AD2.38) Abcam ab78545: Validated by the company and used in 44 scientific literatures.<br>Sheep PAX6 R&D Systems AF8150: Validated by the company and used in 10 scientific literatures.<br>Rabbit RFP/tdtomato/Crimson Abcam ab62341: Validated by the company and used in 290 scientific literatures.<br>Rabbit SOX2 Abcam ab97959: Validated by the company and used in 656 scientific literatures.<br>Goat SOX2 R&D Systems AF2018: Validated by the company and used in 196 scientific literatures.<br>Rabbit SOX6 Abcam ab30455: Validated by the company and used in 56 scientific literatures.<br>Rabbit TBR1 Abcam ab31940: Validated by the company and used in 425 scientific literatures.<br>Rabbit TH Abcam ab112: Validated by the company and used in 332 scientific literatures.<br>Sheep TH Abcam ab113: Validated by the company and used in 63 scientific literatures. |

# Eukaryotic cell lines

Policy information about cell lines and Sex and Gender in Research

| Cell line source(s) | The hESC lines (H1 (WAe001-A), H7 (WA07), H9 (WA09) (all WiCell) and in-House (IMBA Stem Cell Core Facility)  generated human iPSC lines (176/1, 178/5, 178/6) were used in this study. |
|---|---|
| Authentication | Cell lines were authenticated by WiCell or IMBA Stem Cell Core Facility and checked for pluripotency and genomic integrity. Cell lines were checked to be morphologically consistent and show no signs of differentiation before every passage.<br><br>The protocol and informed consent form regarding iPSC derivation for the cell lines SCCF-176J clone#1, SCCF-178 clone#5 and #6  was reviewed and approved by the properly constituted Institutional Review Board/Independent Ethics Committee at the Medical University of Vienna (EK No. 1596/2017 ).<br><br>Cell lines SCCF-176J clone#1 and SCCF-178 clone#5 have been whole-genome sequenced and sequences are available from the EGA database (Study ID EGAS00001006262; Dataset: https://ega-archive.org/datasets/EGAD00001008769). Cell lines SCCF-176J clone#1 (https://hpscreg.eu/cell-line/IMBAi001-A), SCCF-178 clone#5 (https://hpscreg.eu/cell-line/IMBAi003-A) and SCCF-178 clone#6 (https://hpscreg.eu/cell-line/IMBAi003-B) have been registered at the Human Pluripotent Stem Cell Registry (https://hpscreg.eu/). |
| Mycoplasma contamination | All cell lines were routinely tested to be negative for mycoplasma contamination. |
| Commonly misidentified lines<br>(See ICLAC register) | No common misidentified lines have been used in this study. |

