## [Peer Review File · Nature Methods]

Peer Review Information

Manuscript Title: In vitro Modeling of the Human Dopaminergic System using Spatially Arranged ventral Midbrain-Striatum-Cortex Assembloids

Corresponding author name(s): Juergen Knoblich

Editorial Notes: n/a

Reviewer Comments & Decisions:

Decision Letter, initial version:

Dear Jürgen,

I hope you're doing well. Your Article, "In vitro modeling of the human dopaminergic system using spatially arranged ventral midbrain-striatum-cortex assembloids", has now been seen by 3 reviewers. As you will see from their comments below, although the reviewers find your work of considerable potential interest, they have raised a number of concerns. We are interested in the possibility of publishing your paper in Nature Methods, but would like to consider your response to these concerns before we reach a final decision on publication.

We therefore invite you to revise your manuscript to address these concerns particularly those about claims without experimental evidence and some missing methodological details in the paper. Please also make sure to highlight the advance represented by the MISCOS.

* include a point-by-point response to the reviewers and to any editorial suggestions

* please underline/highlight any additions to the text or areas with other significant changes to facilitate review of the revised manuscript

* address the points listed described below to conform to our open science requirements

* ensure it complies with our general format requirements as set out in our guide to authors at www.nature.com/naturemethods

* resubmit all the necessary files electronically by using the link below to access your home page

[Redacted] This URL links to your confidential home page and associated information about manuscripts you may have submitted, or that you are reviewing for us. If you wish to forward this email to co-authors, please delete the link to your homepage.

We hope to receive your revised paper within 10 weeks. If you cannot send it within this time, please let us know. In this event, we will still be happy to reconsider your paper at a later date so long as nothing similar has been accepted for publication at Nature Methods or published elsewhere.

OPEN SCIENCE REQUIREMENTS

REPORTING SUMMARY AND EDITORIAL POLICY CHECKLISTS

Please note that these forms are dynamic 'smart pdfs' and must therefore be downloaded and completed in Adobe Reader. We will then flatten them for ease of use by the reviewers. If you would

like to reference the guidance text as you complete the template, please access these flattened versions at <http://www.nature.com/authors/policies/availability.html>.

DATA AVAILABILITY

We strongly encourage you to deposit all new data associated with the paper in a persistent repository where they can be freely and enduringly accessed. We recommend submitting the data to discipline-specific and community-recognized repositories; a list of repositories is provided here:

<http://www.nature.com/sdata/policies/repositories>

All novel DNA and RNA sequencing data, protein sequences, genetic polymorphisms, linked genotype and phenotype data, gene expression data, macromolecular structures, and proteomics data must be deposited in a publicly accessible database, and accession codes and associated hyperlinks must be provided in the “Data Availability” section.

Please include a “Data availability” subsection in the Online Methods. This section should inform readers about the availability of the data used to support the conclusions of your study, including accession codes to public repositories, references to source data that may be published alongside the paper, unique identifiers such as URLs to data repository entries, or data set DOIs, and any other statement about data availability. At a minimum, you should include the following statement: “The data that support the findings of this study are available from the corresponding author upon request”, describing which data is available upon request and mentioning any restrictions on availability. If DOIs are provided, please include these in the Reference list (authors, title, publisher (repository name), identifier, year). For more guidance on how to write this section please see: <http://www.nature.com/authors/policies/data/data-availability-statements-data-citations.pdf>

MATERIALS AVAILABILITY

SUPPLEMENTARY PROTOCOL

To help facilitate reproducibility and uptake of your method, we ask you to prepare a step-by-step Supplementary Protocol for the method described in this paper. We [encourage authors to share their step-by-step experimental protocols](https://www.nature.com/nature-research/editorial-policies/reporting-standards#protocols) on a protocol sharing platform of their choice and report the protocol DOI in the reference list. Nature Portfolio's Protocol Exchange is a free-to-use and open resource for protocols; protocols deposited in Protocol Exchange are citable and can be linked from the published article. More details can found at www.nature.com/protocolexchange/about.

ORCID

Please do not hesitate to contact me if you have any questions or would like to discuss these revisions further by email or video chat. We look forward to seeing the revised manuscript and thank you for the opportunity to consider your work.

Sincerely,
Madhura

Madhura Mukhopadhyay, PhD
Senior Editor
Nature Methods

Reviewers' Comments:

Reviewer #1:

Remarks to the Author:

Comments to “In vitro modeling of the human dopaminergic system using spatially arranged ventral midbrain-striatum-cortex assembloids”

In the present study, Reumann et al. reported modified methods for generating VM, striatal and cortical organoids from human pluripotent stem cells. Using immunofluorescence staining, RT-qPCR, and transcriptomics, they demonstrated these methods leading to the fate of targeted brain regions. Using custom embedding molds, authors further generate ventral Midbrain-Striatum-Cortical Organoid assembloids by positioning the organoids linearly in their anterior: posterior direction, named MISCOs. In this study, immunofluorescent staining and rabies virus-based retrograde tracing were used to verify the anatomical connectivity of MISCO assembloids; while optogenetics, calcium imaging, and fluorescent dopamine sensors were used to verify functional connectivity. Finally, as a test of the application of this system, authors demonstrated the innervation and maturation properties of injected dopaminergic progenitors and the Cocaine-induced morphological, functional, and transcriptional changes in MISCO assembloids. The variety of cell lines used and systems tested is impressive, which adds to the MISCO's robustness. The testing of this system, including injected dopaminergic progenitors and neural/neuronal circuit-related changes by cocaine, provides a glimpse of potential applications in the future. As far as I am aware, there has not been a report before for testing addictive drugs in artificial complex neural circuits in vitro, which adds another layer to their system and observed overstimulation of dopaminergic signaling.

I have several general issues with the manuscript in its current form, which will have to be addressed in the revision.

1. Generally, using organoid assembloids to explore the functional and structural wiring between different human brain regions is interesting. From a specific observing angle, the wiring between two brain regions is linear anyway, which can easily be recapitulated by two fused organoids. If the fusion reached 3 or more, the fusion becomes more complicated because of the varied distance between brain

regions in vivo, leading the multiple-dimensional connections. To what extent does the recently fused organoid reflect/mimic the neuronal wiring in the human brain? A recent publication reported a microfluidics-based strategy for human brain organoid assembly in a controlled, which allowed sequential assemboid covering 1D sequences or 2D arrays(Zhu, Zhang, et al. 2023). How should the wiring principle be in three or more assembloids? Which is more advantageous for interrogating human neural circuits, linear wiring or arrayed?

2. For an intact neural circuit in vivo, the guidance of A to B is as important as the feedback from B to A, while the indirect interaction of A to B is as important as the direct interaction. Compared with the complex inter-regional wiring in the brain and the inter-organ crosstalk in the body, current assembloid models still need to be improved in their complexity. Therefore, this manuscript's MISCO system provided the chance to access more complex neural circuits. In the present study, authors focus on the interactions between the two brain regions, as in the two organoid assembloid. Decoding advanced crosstalk preliminarily in MISCOs will be interesting and of importance to this field. It is also a challenge, as we know.

Please see a list of major and minor points to address:

Major points:

1. In Extended Data Fig. 3 a-d:

(1) The days of samples are missing in Fig 3a. Several publications indicated that PAX6 is the earliest neuroectodermal marker expressed in the developing human brain, as well as a neocortex-enriched marker(Zhang, Huang, et al. 2010, Onorati, Castiglioni, et al. 2014). Thus, when GSX2 and PAX6 antibodies immunostaining are used to identify ventral and dorsal forebrain, the culturing days of the sample should be displayed.

(2) It is exceptional to observe a large number of GSX2+ cells in unpatterned organoids. The study from the same group showed that the 'dorsalUnt' organoids were nearly all dorsal tissue (96% TBR1 and 76% PAX6) with only small amounts of ventral tissue (0% NKX2-1, 5% DLX2, and 6% GSX2)(Bagley, Reumann et al. 2017). The induction strategy that they used was: 'dorsalUnt', with no drugs, distinguished from Ventral organoids with 2.5 μ M IWP2 and 100 nM SAG(Bagley, Reumann et al. 2017). Therefore, it is necessary to explain why GSX2 is highly expressed in unpatterned organoids in this study. Please specify and provide more details of the quantifying methods used in Extended Data Fig. 3a-d.

(3) An unexpected "Acta" appeared in the X axis of Extended Data Fig. 3 d.

2. The data of MISCOs scRNA-seq in Fig2 and Extended Data Fig. 6:

(1) For scRNAseq preparation, MISCOs were separated by a scalpel into the cortex/striatum/midbrain. What is this method's advantage when compared to digesting whole MISCOs for supervised clustering?

(2) The results showed the volume of VM/striatum/cortex organoids in MISCOs was different after the long-term culturing (Fig.2e, 2f, 3a, and 5c), and the cortical region acquired a larger volume. These data are consistent with the observation in our previous work(Chen, Saiyin et al. 2022). The ratio of the cortical progenitor/excitatory neurons is relatively low in scRNA-seq data, and striatal parts are relatively high. Please specify the steps used to annotate cell types and provide the possibility that causes the low cortical progenitor/excitatory neurons and high striatal cells in scRNA.

(3) The authors used Gruffi methods for selection and filtering in scRNAseq analysis. In our opinion, Gruffi are better than the basic filtering widely used, such as genes, mitochondrial- and ribosomal reads. Thus, we are wondering if the Gruffi method is the cause of the question mentioned above. For example, cortical regions in MISCOs have more "stressed cells" because of the larger volume. If the filtering caused the low cortical progenitor/excitatory neurons in this dataset, the reviewer feels it is necessary to re-adjust the filtering method.

3. Regional judgments in MISCOs, including Fig. 2, 3, and 5, Extended Data Fig 7 and 8.

The question below originated from our experience in the fused organoid culture. The distribution pattern of different regions in fused organoids did not have a clear margin in the fused organoids with long-term culture. A large region with quicker growth sometimes circles a smaller region. The structure in Fig. 2f supported the existence of this phenomenon. In such conditions, the fusion will blur the margins of two non-GFP organoids in three organoids fusion, which might cause a challenge to identify different regions during staining or analysis.

(1) Please clearly annotate how to identify the GFP+ region in long-term cultured fused organoids. If the identification is based on the experience in the author's lab, we suggested that it is necessary to show clearly regional criteria in the long-term cultured MISCO. These figures will clarify the correctness of their experience. We suggest using a combined strategy of viruses (GFP/tdtomato) and specific markers (FOXA2/DARPP32).

(2) Please specify in the methods how the different regions in the MISCOs are distinguished during microinjection, including retrograde viral tracing and VM progenitors injecting.

4. The quantification needs to be included in Fig.5d and 5e.

The missing quantification did not fully support the statements (Line 334-337):

"Similar to our previous observations, we found different innervation densities of the grafts in striatal and cortical tissues (Fig. 5e). Grafts were positive for the markers FOXA2 and TH, suggesting differentiation of injected cells into mDA neurons (Fig. 5d)."

It came to our attention that the progenitor of mDA used for transplantation did not uniformly express GFP in Extended Data Fig. 9 m-t. Please specify the criteria for selecting 40, 000 GFP+ in transplantation. In addition, the ratios of FOXA2 and TH in GFP+ cells in the transplanting cells are critical to deciding the efficiency of transplantation. Thus, counting the ratios of FOXA2 and TH+ in GFP+ cells is necessary.

Minor points

1.Line 96: please replace "stratal" with "striatal."

2.A small insert to Fig. 1d is necessary to show the specificity of FOXA2 in the organoid.

3.Line 114: (such as sonic hedgehog (SHH)) should be changed to (such as sonic hedgehog [SHH])

4.Lines 116 and 117: SHH is a protein here. It should be SHH but not SHH.

5.In Fig.1 d-g, please count TH+ cells in FOXA2/EN1/LMX1A+ cells.

6.The reviewer suggests showing the DAPI channel with DARPP32 and GAD1 antibodies staining. DAPI staining not only shows the nucleus but also displays the necrosis.

7. In Fig. 1S and Fig. 2K-L, the author used Voxhunt to compare a single time point in the mouse brain in BulkRNA-seq analysis, whereas the author compared both human and mouse brain data in the scRNA-seq analysis. Please explain the rationality of this analysis.
8. The nucleus staining should be shown in Extended Data Fig. 7d.
9. Fig. 4 J and 4K: It is necessary to display calibration bar of Pseudo-color here.
10. Line 255: (Extended Fig. 6b). It should be (Extended Fig. 7b).
11. Line 269: (Fig. i,j, Extended Data Fig. 7f-h). It should be (Fig. 3 i,j, Extended Data Fig. 7f-h).
12. LMX1A is a transcription factor. Thus, the LMX1A staining signal should be in the nucleus. In Extended Data Fig. 9s. LMX1A signal in this image resembles the cytoskeleton. Please check the original images in Extended Data Fig. 9s.
13. Line 368-370: the author state that:
Moreover, the diameter of TH+ varicosities was significantly reduced in both striatal and cortical tissue indicating varicosities of reduced volume. At the same time, 25-day long withdrawal also failed to rescue the neuromorphological effects of cocaine.

But in citation 61(Wildenberg, Sorokina et al. 2021):

The second obvious feature of DA axons exposed to cocaine was the occurrence of large axonal swellings or bulbs (Figure 7). The swellings were large (mean \pm SEM diameter: $2.2 \pm 0.3 \mu\text{m}$, $n = 23$), significantly larger than varicosities in control animals (mean \pm SEM diameter: $0.4 \pm 0.02 \mu\text{m}$, $n = 118$ varicosities) and at times reaching the size of neuronal soma (Figure 7A). These 'bulbs' were common in axons (~56%, 17/30 axons) in two cocaine-exposed animals, and we did not see a single example in DA axons from two control animals (0/29 axons), suggesting that Apex2 expression alone does not cause these swellings (Figure 7B; mean \pm SEM swellings/ μm length of axon: +saline, 0.00 ± 0.0 , $n = 29$ axons, two mice; +cocaine, 0.04 ± 0.02 , $n = 30$ axons, two mice. $p = 1.7e-5$).

Regarding this point, the exposure of MISCO to cocaine did not recapitulate the morphological /structural changes of TH+ varicosities in vivo. If this finding has support from other publications? Or does the author feel that MISCO is a hitherto model that brings a new perspective?

Citation

- Bagley, J. A., D. Reumann, S. Bian, J. Lévi-Strauss and J. A. Knoblich (2017). "Fused cerebral organoids model interactions between brain regions." *Nature Methods* 14(7): 743-751.
- Chen, X., H. Saiyin, Y. Liu, Y. Wang, X. Li, R. Ji and L. Ma (2022). "Human striatal organoids derived from pluripotent stem cells recapitulate striatal development and compartments." *PLoS Biology* 20(11): e3001868.
- Onorati, M., V. Castiglioni, D. Biasci, E. Cesana, R. Menon, R. Vuono, F. Talpo, R. Laguna Goya, P. A. Lyons, G. P. Bulfamante, L. Muzio, G. Martino, M. Toselli, C. Farina, R. A. Barker, G. Biella and E. Cattaneo

(2014). "Molecular and functional definition of the developing human striatum." *Nature Neuroscience* 17(12): 1804-1815.

Wildenberg, G., A. Sorokina, J. Koranda, A. Monical, C. Heer, M. Sheffield, X. Zhuang, D. McGehee and B. Kasthuri (2021). "Partial connectomes of labeled dopaminergic circuits reveal non-synaptic communication and axonal remodeling after exposure to cocaine." *ELife* 10.

Zhang, X., C. T. Huang, J. Chen, M. T. Pankratz, J. Xi, J. Li, Y. Yang, T. M. Lavaute, X.-J. Li, M. Ayala, G. I. Bondarenko, Z.-W. Du, Y. Jin, T. G. Golos and S.-C. Zhang (2010). "Pax6 is a human neuroectoderm cell fate determinant." *Cell Stem Cell* 7(1).

Zhu, Y., X. Zhang, L. Sun, Y. Wang and Y. Zhao (2023). "Engineering Human Brain Assembloids by Microfluidics." *Advanced Materials* (Deerfield Beach, Fla.): e2210083.

Reviewer #2:

Remarks to the Author:

The manuscript by Reumann et al uses hiPSC-derived region-specific organoids to generate a model of the human dopaminergic system. By fusing separately-generated midbrain, striatum and cortical organoids they generate an assembloid that mimics some aspects of this system in vitro, and can be used to ask questions about transplantation and the effects of drugs. Overall, although the generation of the individual region-specific organoids is not novel, the assembly of these 3 regions together has not been done before and has important implications for research on human-specific aspects of the dopaminergic system. There are, however, a number of specific points that should be addressed:

1- Quantification, analysis and stats

1.1- The authors include the number of organoids used for experiments in the figure legends as a way of quantifying their results. However, in most cases it is unclear what they are quantifying. There is no section in the methods that specifies how the quantifications are done either.

For example, in Figure 1e authors say "44-day old organoids display clusters of TH+ and FOX2A+ mDA neurons (n=8/8 organoids of 2 batches and 2 cell lines)". The figure only shows one immunostaining of one organoid stained with TH and FOX2A. What is the quantification for? Are authors saying that 8 out of 8 organoids they looked at had TH+/FOX2A+ clusters? What do they consider a cluster for this quantification? A cluster could be anything from 3 cells to thousands of cells. And what percentage of the organoids do these clusters correspond to? This should be noted, otherwise the numbers shown have little meaning.

This is one example, but there are many instances of this throughout the manuscript. These include: Fig 1d, 1e, 1f, 1g, 1i, 1j, 1k, 1l, 2c, 2e, 2f, 2g, 2h, 3a, 3l, 3m, 5b, 5c, 5e and Ext Data Fig 1c, 3i, 3j, 3k, 5g, 5h, 5i, 7a, 7b, 7d, 9u.

Please specify in each case what the quantification denoted in the figure legends for the panels above correspond to. This could be either in the figure legends or as a separate methods section.

1.2- In general, however, I am confused as to why these quantifications are not part of the figures. If these quantifications were performed, wouldn't they strengthen the manuscript if included in the form of graphs as part of the figures?

1.3- Quantification is missing for some experiments. For example:

- Fig 3e says "TH+ axons generally avoid neurogenic regions" - there is no quantification for this.
- Fig 5e says "Striatal tissue displayed denser innervation of grafted cells in comparison with cortical tissues..." - there is not quantification for this
- In the text for Ext Data Fig 7e it says "...found that most forebrain TH+ cells expressed GFP..." - there is not quantification for this

Either quantification should be performed or statements toned down

1.4- The authors perform ANOVA analysis in a few instances throughout the manuscript followed by t-tests. This is not standard practice. What is the reason for this? Usually ANOVA tests are followed by tests that correct for multiple comparisons (eg Bonferroni, Dunnett)

1.5- P-values and/or details of statistical tests performed are missing in a few instances (see for example Fig 3c, 3g, 3h, 6d, 6e, 6g, 6h and Ext data Fig 3h, 9a-l)

1.6- For each of the graphs in the manuscript, please state:

- what each of the points corresponds to. For example, in Fig 1n, is each dot an organoid? The legend says "n=8/8 organoids of 2 batches", yet there are only 4 points in the graph that I can see.
- how many sections per organoid are counted in each case (either in legend or in methods).
- how many lines are used per experiment (including RNAseq experiments)

1.7- While I appreciate that authors characterized their organoid recipes in different lines, I believe quantification (IHC or qPCR) would be needed to claim "We confirmed our findings in organoids from three different hiPSC and three different hESC lines, highlighting the robustness of the protocol." The data supporting this claim is limited to one immunostaining per line.

1.8- There are some experiments that appear anecdotal as presented. For example:

- the authors mention they can recapitulate neuronal morphologies within the three brain regions, but I think that this would be stronger if it was paired with immunostainings for relevant markers of those cells.
- in Fig 5g the authors mention “putative axonal transport” with very little evidence (and no quantification). Some other indication that there is transport would be important

2- Functional connectivity

2.1- The authors perform a rabies tracing experiment to show neuronal connectivity in MISCOs. However, the experiment as performed only allows them to show the presence of projections and not neuronal connectivity or functional projections (specifically they say “RV monosynaptic tracing indicated potentially functional mDA long-range projections”). The authors show the presence of mDA neuronal projections from the VM organoid to the striatal and cortical organoids. Injection of the helper virus into the cortical side will transduce mDA projections as well as cortical cells allowing the rabies virus to infect mDA cells directly.

In order to claim spread of the rabies virus authors should perform the transduction before assembly.

2.2- Similarly, the authors use an AAV to transduce Chr2 into MISCOs but they do so after assembly, therefore all cells express it. Even if the authors use a targeted approach to stimulate only VM tissue, Ext Data Fig 7 shows there are reciprocal projections (ie projections from the striatum and cortex to VM). Therefore, light stimulation in VM would also be activating striatal and cortical cells that received the AAV. Based on this experiment the authors cannot say “Together, this data confirms the ability of MISCOs to develop functional neural networks across all regions”.

If the authors want to claim connectivity between the different organoids they would need to either transduce organoids before assembly or use a soma-restricted Chr2.

2.3- Again, authors transduce all regions in MISCOs to show striatal and cortical cells receive dopamine. Even if authors are imaging the presumed striatal and cortical regions, it would be important to at least show some immunostainings to show that the cells that receive the AAV-GRAB-DA2 in the cortex and striatum are the cells they believe.

2.4- It would be useful to see some examples of activity-dependent and dopamine-related genes upregulated after assembly in the main text (in paragraph starting on line 314)

2.5- Authors say that Ext Data Fig 8d, e “allows them to study calcium events spreading through VM axons into forebrain tissues”. However, I am struggling to see this from the figure provided. It is not clear what the images show and what fluorescence is quantified. Please provide more details. There is also no quantification in this panel.

3- Effects of cocaine on dopaminergic system

3.1- The authors mention “not much is known about the effects of cocaine on the development of the CNS – predominantly because of the lack of an effective model system”. What do they mean by effective? There are mouse and rat models used to study in utero exposure to cocaine

3.2- The authors say “we used 0.7uM...a concentration well within the physiological range in humans” – what is the physiological range in humans? Please state

3.3- I am confused about the duration of calcium events in Fig 6g. If I understand correctly, each calcium event lasts ~5 seconds? This is not standard for neuronal calcium events using Gcamp. If the authors are using a constitutive Gcamp line to perform these experiments then they could be recoding events in progenitors. Treatment with TTX could help establish if events are neuronal (which is important based on the claims made)

3.4- How are varicosities quantified/detected? Is it just based on morphology? Are they specific to TH cells? How do we know they are not just unhealthy beading axons? Because of this, it would be important to show varicosities in another way other than morphology (eg. DAT colocalization)

Minor comments

- There is a typo on line 80 (on are on their way)
- There is a typo on line 96 (stratal)
- Genes in lines 162 and 164 should be in italics
- The term “surface recordings” on line 331 is confusing. Do they mean imaging? The term recording is usually used to mean electrophysiological recordings (see also figure legend for that figure)

Reviewer #3:

Remarks to the Author:

The manuscript entitled, “In vitro modeling of the human dopaminergic system using spatially arranged ventral midbrain-striatum-cortex assembloids” describes the establishment of a protocol to generate a complex human neural tissue (here termed MISCO) that recreates aspects of the human dopaminergic system through fusing 3 distinct regional tissues (ventral midbrain, VM; striatum, and cortex). The authors present data supporting protocol development, and then apply the MISCO assembloid to test cell engraftment related to parkinson’s disease therapy and study the response to cocaine stimulation.

Altogether, this is a very interesting paper that uses diverse methods to characterize the system, and study the functional activities within the assembloid under different conditions. I have several questions/concerns that are related to the specificity and robustness of the Method.

As the first step to generate the three-region fusion, the authors describe modulation of previous protocols to generate the three regions prior to fusion. There have been previous papers that describe the generation of midbrain, striatum, and cortex from human pluripotent stem cells. The authors should provide more details and explanation about what is distinct between previous protocols and the protocol described here. Also, the authors need to ensure that all previous protocols are referenced in the manuscript.

Concerning the VM protocol, the authors state that “we found that a concentration of 300nM SAG from day 4-11, together with dual SMAD inhibition and Wnt activation, was sufficient to introduce maximal FOXA2 expression levels by day 20 (Fig. 1b, d Extended Data Fig. 1a).” Did the authors look beyond day 20 at organoids treated with the different concentrations of SAG? Also, did the authors test treatment for other days before or beyond day 4-11? Finally, did the authors test different initiating sizes of the organoids prior to SAG treatment (e.g. using fewer or more cells to start the culture)? Again, how is this protocol different from previously published ventral midbrain organoid protocols?

The authors present bulk RNA-seq data on each of the three tissues, and then compare this data to primary reference datasets showing general similarity to the regions of interest. The authors then use single-cell RNA-seq to characterize cell heterogeneity in the day 60 assembloid by first separating the major regions prior to sequencing. The data is presented in Figure 2j-l, and ED Fig. 6. How many assembloids were used to generate the data? Did the authors generate scRNA-seq data from multiple batches? In addition to the selected feature plots in the ED, the authors should include a heatmap showing marker genes (e.g. 20) for each cluster, as well as a supporting table (apologies if this is included already). It would be helpful if the authors could show the UMAP plot in ED 6b with each condition in separate plots, as well as a stacked bar plot showing the proportion of cells per per cluster broken down by condition. It is difficult to determine if each cluster has contributions from all conditions.

The authors present a device that can be used for assisting in linear organoid fusion. The authors state, “In consecutive batches, we achieved 96% ($\pm 2.4\%$ SD) fusion efficiency (Fig. 2d).” This quantification appears to be based on data from 3 days after fusion. What is the percentage at a later time point (e.g. 21, 33, 109 as in the images)? It would be helpful if the authors could have an schematic in Figure 2 that shows the time point of fusion and the media composition in the assembloid culture, similar to what is shown in Fig. 1b, as the particular details are likely relevant for the reproducibility of the Method.

Figure 3 shows an assembloid that has very different sized VM, striatum, and cortex regions. It would be very helpful to describe the variation in assembloid overall size, as well as the size of the individual regions. This information will be very helpful to other researchers that may try to reproduce the method.

Author Rebuttal to Initial comments

General response to all reviewers

We would like to thank all the reviewers for their thoughtful and constructive comments and suggestions. We have followed their recommendations and performed extensive revision experiments to address all the comments and concerns, which have further substantiated the significance of our findings.

Together with the suggested experiments, we created and updated more than 50 figure panels, which resulted in changes in all figures of the manuscript.

To provide a better overview of the changes, we include here a summary list of data figure changes presented in the revision (Table 1), a table with additional significant changes to the manuscript (Table 2), as well as a detailed point-by-point response to all the issues raised by the reviewers.

Table 1: Summary list of changes to the figures.

Figures	Changes/Additions
Figure 1d	Made a small insert to show specificity of FOXA2
Figure 2e, f	Changed label color in panel e' and e'' as well as f' and f'' to orange (to comply with colorblind regulations)
Figure 3c	Added p value and statistic test used to figure legend (unpaired t-test).
Figure 3e	Added n numbers to the figure legend
Figure 3g,h	Updated statistical test used for significance
Figure 4b-d and f-h	Updated 200 second rasterplots (instead of 10 minutes, see reviewer figure 1 d, f) and spike detection parameters (reviewer figure 1 g) to improve spike detection and visualization. Firing rate and active electrode percentages were updated (see comment 2 in table 2).
Figure 4j,k	Pseudo-color calibration bar was added
Figure 4l-m	Updated panel (inconsistent trimming of first frames of recording which caused bleaching artifacts, updated mean)
Figure 5g	Removed axonal transport experiment from the figure
Figure 6b	Added another "TH" label co cortex examples
Figure 6 d, e, g, h	Updated statistical method used for testing significance
Extended Data Figure 1a	Increased n numbers of SAG dosecurve and updated the panel
Extended Data Figure 1c	Added quantification of FOXA2, LMX1A and EN1+ cells being positive for TH.

Extended Data Figure 1e	Added brightfield images of VM, striatal and cortical organoids
Extended Data Figure 1f	Added size quantifications of VM, Str, Ctx and MISCOS
Extended Data Figure 2c	Added quantification of TH+ tissue as % of total tissue area (DAPI+)
Extended Data Fig. 3b-d	Increased batch numbers and visualization of GSX2 and PAX6 quantifications
Extended Data Fig. 3b-d	Changed "unpatterned" in panel b-d with "ctrl" (to keep nomenclature consistent).
Extended Data Fig. 3h	Updated statistical method used for testing significance
Extended Data Fig. 4c, e	Added quantification of GSX2+ rosettes (c) and DARPP32+ tissue as % of total tissue area (DAPI+).
Extended Data Fig. 5a	Added a schematic for the generation of MISCOS
Extended Data Fig. 6c	Added a stacked bar plot showing barcode origin of clusters in percent
Extended Data Fig. 6h	Added a dotplot of top marker genes expressed in individual clusters.
Extended Data Fig. 8d, e	Added inserts as well as positions of recording timepoints to illustrate GCAMP network events from VM in forebrain tissue.
Extended Data Fig. 8i, j	Added stainings to show GRAB-DA2m expression in striatal (DARPP32) and cortical (TBR1) neurons.
Extended Data Fig 8m	Added an experiment to inhibit synaptic activity to confirm existence of synaptic connectivity (and exclude that what we describe is antidromic action potentials through reciprocal axons in the VM).
Extended Data Fig. 9s	Exchanged mislabeled panel s ("LMX1a) with correct labeling (TAU)
Extended Data Fig. 9u	Added an image of WT MISCOS and MISCOS with CAG-tdtomato expression in VM tissue to visualize that MISCOS display a stereotypical morphology, which allows for tissue-specific injections.

Extended Data Fig. 9v	Added a quantification of innervation density of Str vs Ctx tissue (related to Figure 5e).
Extended Data Fig. 10b	Added a TTX experiment to demonstrate the (relative) specificity of GCAMP expression/ activity to neurons.
Extended Data Fig. 10c, d	Updated statistical method used for testing significance

Table 2: Additional significant changes

Position	Changes/Additions
Reproducibility	Added a reproducibility section to the methods, stating the meaning of n numbers and cell lines used for experiments.
Repetition of silicon probe recordings	Since the initial submission, a technical error with the Open EPhys GUI acquisition software (v0.4.4.1) was found and clarified by the manufacturers. This software bug corrupted extracellular signal recordings and entered empty data points into the binary files of two cortical organoids. These have since been removed and replaced with new replicate recordings (Figures 4 c and d) under instructions by the manufacturer. To ensure no channels in previous replicates were impacted, all channels were checked for potential artefact using the Spike Interface channel removal module. An additional spike detection step was amended to account for action potential refractory periods. Together, these changes lead to improved spike detections in extracellular recordings (see reviewer figure 1g) without changes in initial statistical significances (Figures 4c and g).
Update of Interpretation of Silicon probe recordings	We slightly changed the interpretation in the text regarding connectivity between VM and Str, as the updated analysis and replicates showed similar activated channels between striatum and cortex.
Axonal transport statement (previous Figure 5f)	Removed experiment

Point-to-point response to reviewer's comments

Reviewer #1:

Remarks to the Author:

Comments to "In vitro modeling of the human dopaminergic system using spatially arranged ventral midbrain-striatum-cortex assembloids"

In the present study, Reumann et al. reported modified methods for generating VM, striatal and cortical organoids from human pluripotent stem cells. Using immunofluorescence staining, RT-qPCR, and transcriptomics, they demonstrated these methods leading to the fate of targeted brain regions. Using custom embedding molds, authors further generate ventral Midbrain-Striatum-Cortical Organoid assembloids by positioning the organoids linearly in their anterior: posterior direction, named MISCOs. In this study, immunofluorescent staining and rabies virus-based retrograde tracing were used to verify the anatomical connectivity of MISCO assembloids; while optogenetics, calcium imaging, and fluorescent dopamine sensors were used to verify functional connectivity. Finally, as a test of the application of this system, authors demonstrated the innervation and maturation properties of injected dopaminergic progenitors and the Cocaine-induced morphological, functional, and transcriptional changes in MISCO assembloids. The variety of cell lines used and systems tested is impressive, which adds to the MISCO's robustness. The testing of this system, including injected dopaminergic progenitors and neural/neuronal circuit-related changes by cocaine, provides a glimpse of potential applications in the future. As far as I am aware, there has not been a report before for testing addictive drugs in artificial complex neural circuits in vitro, which adds another layer to their system and observed overstimulation of dopaminergic signaling.

I have several general issues with the manuscript in its current form, which will have to be addressed in the revision.

1. Generally, using organoid assembloids to explore the functional and structural wiring between different human brain regions is interesting. From a specific observing angle, the wiring between two brain regions is linear anyway, which can easily be recapitulated by two fused organoids. If the fusion reached 3 or more, the fusion becomes more complicated because of the varied distance between brain regions *in vivo*, leading the multiple-dimensional connections. To what extent does the recently fused organoid reflect/mimic the neuronal wiring in the human brain? A recent publication reported a microfluidics-based strategy for human brain organoid assembly in a controlled, which allowed sequential assembloid covering 1D sequences or 2D arrays (Zhu, Zhang, et al. 2023). How should the wiring principle be in three or more assembloids? Which is more advantageous for interrogating human neural circuits, linear wiring or arrayed?

We first want to thank the reviewer for the effort to review our manuscript and for the interesting questions! We agree with the reviewer that the connections of 2 brain regions will in a certain context always be linear. However reciprocal connectivity cannot be expected in all context. For example, *in vivo* the cortex highly innervates the striatum, but the striatum projects predominantly into other brain regions and not into the cortex. Dopaminergic neurons on the other hand do innervate both cortex and striatum- but innervate the striatum much more than the cortex. For both of these findings we also have correlating findings in our fusions- both on dopaminergic innervation (Figure 3b, c) as well as striatal innervation (Extended Fig. 7a-b). Thus, besides the general mode of "region a interacts with region b" paradigm, the amount (and identity) of connectivity (and the specific type of neurons that form the connections) also need to be considered. We also agree with the reviewer that an increase in the number of regions will make this issue more complex with every region.

We found the work of Zhu, Zhang, et al. 2023 highly interesting and believe that there is significant potential in microfluidic (and also acoustic) manipulation. To date, however, we have not seen methodologies which would allow such spatial arrangements using larger (1mm and more) tissues. We believe that both the size and precise tissue architecture (neuroepithelial

rosettes and neuronal morphologies) are relevant factors in modeling neurodevelopment. Furthermore, we think that a relatively late fusion time point (at which organoids will usually reach about >1 mm in size) is advantageous: the earlier the fusion, the more likely it will be that patterning cross-talk will occur (e.g. from SHH-expressing floor plate cells of the ventral midbrain which might further ventralize forebrain tissue).

Regarding the positioning of organoids for interrogating human neural circuits: we believe that this will very much depend on the circuit investigated. The mesocorticolimbic and nigrostriatal circuits project with posterior to anterior directionality, which (at least speaking about brain regions) is relatively linear: The cortex in the anterior/rostral direction, the ventral midbrain more posterior/caudal, and the striatum in between. We were indeed speculating how to expand our fusion system to more than one axis, e.g. by the introduction of lateral brain regions (as discussed in the discussion of our manuscript), which could be future work to study even more complex brain region interactions.

2. For an intact neural circuit *in vivo*, the guidance of A to B is as important as the feedback from B to A, while the indirect interaction of A to B is as important as the direct interaction. Compared with the complex inter-regional wiring in the brain and the inter-organ crosstalk in the body, current assembloid models still need to be improved in their complexity. Therefore, this manuscript's MISCO system provided the chance to access more complex neural circuits. In the present study, authors focus on the interactions between the two brain regions, as in the two organoid assembloid. Decoding advanced crosstalk preliminarily in MISCOs will be interesting and of importance to this field. It is also a challenge, as we know.

We thank the reviewer for this comment. We also believe that assembloid methodology, as well as studying inter-organ crosstalk *in vitro*, is still in its early steps of development and we hope that MISCOs and similar methodologies and systems will contribute to decipher human specific aspects of development and pathologies. While we demonstrate the existence of reciprocal connectivity on morphological level (Extended Data Fig. 7a, b) we believe that studying the complex regulatory circuits which emerge within- and between- these regions to be out of the

scope of our study. However, we provide several ideas (such as the study of Huntington's Disease) in which these circuits will be of high relevance, and we are keen to follow up these investigations of complex regulatory circuit formation in future work. We also believe that improvements in MEA technology (such as neuropixels, high resolution MEAs and 3D/folding MEAs) are necessary to study the formation of such complex circuits in the future.

Please see a list of major and minor points to address:

Major points:

1. In Extended Data Fig. 3 a-d:

(1) The days of samples are missing in Fig 3a. Several publications indicated that PAX6 is the earliest neuroectodermal marker expressed in the developing human brain, as well as a neocortex-enriched marker (Zhang, Huang, et al. 2010, Onorati, Castiglioni, et al. 2014). Thus, when GSX2 and PAX6 antibodies immunostaining are used to identify ventral and dorsal forebrain, the culturing days of the sample should be displayed.

We apologize for overseeing to add the days of organoids to this panel. We added the days of the samples (day 27) to the description, which is a time point where PAX6 expression is restricted to cortical neuroepithelium.

(2) It is exceptional to observe a large number of GSX2+ cells in unpatterned organoids. The study from the same group showed that the 'dorsalUnt' organoids were nearly all dorsal tissue (96% TBR1 and 76% PAX6) with only small amounts of ventral tissue (0% NKX2-1, 5% DLX2, and 6% GSX2) (Bagley, Reumann et al. 2017). The induction strategy that they used was: 'dorsalUnt', with no drugs, distinguished from Ventral organoids with 2.5 μ M IWP2 and 100 nM SAG (Bagley, Reumann et al. 2017). Therefore, it is necessary to explain why GSX2 is highly expressed in

unpatterned organoids in this study. Please specify and provide more details of the quantifying methods used in Extended Data Fig. 3a-d.

For the generation of both striatal and ventral midbrain organoids, we used an entirely different media composition for EB formation from day 0. While cortical organoids in Bagley et al. 2017 were generated from EBs using homemade stem cell media (which amongst others contains the growth factor FGF-2), we used a neural induction media for striatum and ventral midbrain. This neural induction media could be considered a basal media and is lacking morphogens, thus not having an effect on anterioposterior or dorsoventral patterning. Our interpretation of the PAX6/GSX2 staining of unpatterned organoids is, that through this lack of patterning factors, they are neither dorsally nor ventrally patterned, thus having the identity of roughly the middle of the neural tube- which separates into cortical as well as subcortical/LGE identities. As human PSCs can have a baseline expression of morphogens such as wnt species and shh (which will vary in parts due to the non-naïve, primed state of hESCs), we think that the ratio of PAX6 and GSX2 expressing cells will vary dependent on the expression of such factors.

To clarify this issue in the paper, we added a description to the striatal paragraph: “We performed a SAG dose-response curve in the presence of the Wnt inhibitor IWP2 in an otherwise growth-factor free neural induction media.” We also performed additional quantifications for Extended Data Fig. 3b-d and updated the figure legend with more details.

(3) An unexpected “Acta” appeared in the X axis of Extended Data Fig. 3 d.

We removed the ActA and apologize for the mistake.

2. The data of MISCOS scRNA-seq in Fig2 and Extended Data Fig. 6:

(1) For scRNAseq preparation, MISCOS were separated by a scalpel into the cortex/striatum/midbrain. What is this method's advantage when compared to digesting whole MISCOS for supervised clustering?

As the size of differently patterned organoids was quite different (we provide additional size quantifications in extended data figure 1d and e) we aimed for a reduction of the tissue volume bias per region. To demonstrate this issue further: with an average radius of 1mm, ventral midbrain organoids would have a calculated volume of $4,1\text{mm}^3$. Cortical organoids on the other hand, with an assumed radius of 1.75mm, would have a volume of $22,45\text{mm}^3$ – and thus have approx. 5.5x the volume of an average ventral midbrain organoid (and a similar ratio of cells between cortical and ventral midbrain tissue). On top of this, we found that ventral midbrain tissue was harder to dissociate than cortical or striatal organoids on day 60, which would have created an additional bias. As we wanted to demonstrate the cellular diversity of different tissues in MISCOs, we believe that this was the better approach to an unbiased approach.

(2) The results showed the volume of VM/striatum/cortex organoids in MISCOs was different after the long-term culturing (Fig.2e, 2f, 3a, and 5c), and the cortical region acquired a larger volume. These data are consistent with the observation in our previous work (Chen, Saiyin et al. 2022). The ratio of the cortical progenitor/excitatory neurons is relatively low in scRNA-seq data, and striatal parts are relatively high. Please specify the steps used to annotate cell types and provide the possibility that causes the low cortical progenitor/excitatory neurons and high striatal cells in scRNA.

We thank the reviewer for this comment. We performed extensive supervised analysis by investigating established brain region (and neural subtype) specific marker expression and believe that the annotations in the single cell dataset are correct. We agree with the reviewer that the amount of cortical excitatory neurons in this experiment was somewhat lower than inhibitory neurons. We have several speculations about why this could be.

First, in the past we found that cortical neurons were harder to dissociate in comparison to interneurons which creates a bias towards interneurons. Secondly, while LGE and cortical progenitors clustered separately, CGE and MGE derived interneurons and their progenitors clustered closer together, thus resulting in larger clusters.

The third speculation we have is that on day 60, the neurogenic tissue of the ventral forebrain has already generated many interneurons and striatal neurons, whereas neurogenesis in the cortex starts later- this could create an additional reduction in cortical neurons.

Last but not least, we often see a significant population of CGE derived interneurons in cortical organoids, which is also a significant fraction of cells in our dataset. While we believe that the presence of MGE and CGE interneurons is something good for our system (after all, interneurons are needed for proper circuit formation in the cortex), this could have additionally reduced the amount of cortical neurons further.

(3) The authors used Gruffi methods for selection and filtering in scRNAseq analysis. In our opinion, Gruffi are better than the basic filtering widely used, such as genes, mitochondrial- and ribosomal reads. Thus, we are wondering if the Gruffi method is the cause of the question mentioned above. For example, cortical regions in MISCOS have more "stressed cells" because of the larger volume. If the filtering caused the low cortical progenitor/excitatory neurons in this dataset, the reviewer feels it is necessary to re-adjust the filtering method.

In our initial analysis of our scRNAseq data, we investigated different filtering methods for good/bad quality cells, but preferred Gruffi for its performance. We did not notice a particular bias towards cortical tissues and found that Gruffi was generally performing very well in removing poorly annotated cells from the dataset without a strong bias towards particular tissues. In total, 26,3% cortical, 25,7% striatal and 34,5% VM cells were removed using stress filtering.

3. Regional judgments in MISCOS, including Fig. 2, 3, and 5, Extended Data Fig 7 and 8. The question below originated from our experience in the fused organoid culture. The distribution pattern of different regions in fused organoids did not have a clear margin in the fused organoids with long-term culture. A large region with quicker growth sometimes circles a smaller region. The structure in Fig. 2f supported the existence of this phenomenon. In such

conditions, the fusion will blur the margins of two non-GFP organoids in three organoids fusion, which might cause a challenge to identify different regions during staining or analysis.

The phenomenon the reviewer describes is something that we also observed before- particularly in fusions of very large and very small tissues, where the large tissue still undergoes significant proliferation. However, in the situation of MISCOS, we did not observe this phenomenon. We speculate that one of the reasons is, that we fuse quite late- which results in less growth of the tissue relative to the other tissue. On the other hand, while e.g. VM tissue was smaller than cortical tissue, the size difference was not too much and the tissue boundaries (e.g. in fused organoids with GFP or tdtomato in one region, such as Figure 3a, extended figure 7d) were always quite sharp and easy to spot for us. Figure 2f was recorded in a cell culture microscope with a limited dynamic range, which we think caused the phenomenon of lower GFP intensity on the outside of the VM core (the center of an organoid will always be the brightest). While we show this specificity with GFP in figure 3a for the VM tissue, we do not show that this is the case for the striatum. We thus wanted to provide additional evidence in reviewer figure 1a, where we demonstrate that the striatal tissue is also not engulfed by the cortical tissue. In the context of this question, but also the next question raised by the reviewer, we also now provide an image of MISCOS in a 6well plate (both WT MISCOS as well as MISCOS with tdtomato expression in VM tissue) which demonstrates that the orientation of MISCOS can be observed relatively straightforward (Extended Data Fig. 9u).

(1) Please clearly annotate how to identify the GFP+ region in long-term cultured fused organoids. If the identification is based on the experience in the author's lab, we suggested that it is necessary to **show clearly regional criteria in the long-term cultured MISCO**. These figures will clarify the correctness of their experience. We suggest using a combined strategy of viruses (GFP/tdtomato) and specific markers (FOXA2/DARPP32).

We thank the reviewer for his suggestion. The identification of a GFP+ region in long-term cultured fused organoids was never an issue in our experiments, as the cell line which we use for constitutive GFP expression (H9 CAG-GFP in the AAVS1 safe harbor locus) keeps GFP expression,

making it easy to spot these regions also in long-term cultures (see also Figure 3a, reviewer figure 1a). Furthermore, it was straightforward to assume the orientation of MISCOS based on their morphology. To clarify this issue better, we added a panel into extended Data Fig. 9 which demonstrates the morphological criteria of MISCOS: the ventral midbrain position can be easily spotted as it is the smaller side of the elongated MISCO. To demonstrate this, we display both Wildtype fusions of MISCOS, as well as chimeras containing a CAG-tdtomato expressing cell line for VM organoid tissue which can be readily seen by eye due to the high expression levels of tdtomato. This demonstrates that we can use regional criteria to spot tissue orientation.

We also want to give the example of the RV injection setup in figure 3l, which shows how helper virus injection into the forebrain allows tracing into the ventral midbrain (the only region with TH⁺ cells), thus the TH staining itself demonstrated that we targeted the forebrain part.

To further elaborate on our controls: For all injection experiments, a counterstain for TH has been performed in our tissue clearing recordings, which in the example of mDA injections also informed us about the appropriate injection site (the VM part of MISCOS). For silicon probe recordings, where an imaging approach would not be suitable, we recorded close to the VM tissue for striatal recordings (see also Extended Data Fig. 8g), and opposite to the VM for cortical recordings. Based on the data we provide and our experience, we consider this approach to be sufficient for appropriate tissue targeting.

(2) Please specify in the methods how the different regions in the MISCOS are distinguished during microinjection, including retrograde viral tracing and VM progenitors injecting.

We thank the reviewer for his suggestion and we added a description to the viral transduction protocol to clarify this issue.

4. The quantification needs to be included in Fig.5d and 5e.

The missing quantification did not fully support the statements (Line 334-337): “Similar to our previous observations, we found different innervation densities of the grafts in striatal and cortical tissues (Fig. 5e). Grafts were positive for the markers FOXA2 and TH, suggesting differentiation of injected cells into mDA neurons (Fig. 5d).” It came to our attention that the progenitor of mDA used for transplantation did not uniformly express GFP in Extended Data Fig. 9 m-t. Please specify the criteria for selecting 40,000 GFP+ in transplantation. In addition, the ratios of FOXA2 and TH in GFP+ cells in the transplanting cells are critical to deciding the efficiency of transplantation. Thus, counting the ratios of FOXA2 and TH+ in GFP+ cells is necessary.

We apologize for the unclear formulation in our manuscript. Indeed, not all cells express GFP, but only a fraction. We used a cell line which had endogenous labeling of TH with Cre and which was transduced with a Lentivirus containing a floxed GFP, which were kindly generated and characterized by the Parmar lab according to their previous publication (Fiorenzano, Birtele, Wahlestedt and Parmar, 2021).

Thus, we selected 40,000 VM progenitors where TH expressing, lentiviral labeled cells were expressing GFP. A full characterization of this cell line, including quantifications for FOXA2 and TH in GFP+ cells, can be found in the publication (Fiorenzano, Birtele, Wahlestedt and Parmar, 2021). We updated our text to clarify this in our text.

We also added a quantification for innervation density in striatal and cortical tissue (Extended Data Fig. 9v).

Minor points

1.Line 96: please replace “stratal” with “striatal.”

We exchanged stratal with striatal.

2. A small insert to Fig. 1d is necessary to show the specificity of FOXA2 in the organoid.

We added a small insert into figure 1d to show the specificity of FOXA2.

3. Line 114: (such as sonic hedgehog (SHH)) should be changed to (such as sonic hedgehog [SHH])

We edited as suggested.

4. Lines 116 and 117: SHH is a protein here. It should be SHH but not SHH.

We edited as suggested.

5. In Fig. 1 d-g, please count TH+ cells in FOXA2/EN1/LMX1A+ cells.

We added the quantifications (extended Figure 1 panel c).

6. The reviewer suggests showing the DAPI channel with DARPP32 and GAD1 antibodies staining. DAPI staining not only shows the nucleus but also displays the necrosis.

We added the DAPI channel.

7. In Fig. 1S and Fig. 2K-L, the author used Voxhunt to compare a single time point in the mouse brain in BulkRNA-seq analysis, whereas the author compared both human and mouse brain data in the scRNA-seq analysis. Please explain the rationality of this analysis.

The rationality of this analysis was to compare our sequencing dataset with already existing spatial transcriptomics data, for which at the timepoint of analysis only mouse data was existent. While there are differences between mouse and human brain, in correlative analysis brain

regions of different species will still correlate with each other, which is why we selected this comparison (such as in Miura et al., 2022). We did not perform Brainspan on the bulk RNAseq analysis, as one of the two brain regions which we are establishing in figure 1 (ventral midbrain) is a region which does not exist in Brainspan. However, to clarify this issue we provide a Brainspan analysis of Bulk RNAseq for cortical and striatal organoids in the reviewer figure (panel c).

8.The nucleus staining should be shown in Extended Data Fig. 7d.

The nuclear staining has been added in Extended Data Fig. 7d.

9.Fig.4 J and 4K: It is necessary to display calibration bar of Pseudo-color here.

We added the display calibration bar.

10.Line 255: (Extended Fig. 6b). It should be (Extended Fig. 7b).

We corrected this typo.

11.Line 269: (Fig. i,j, Extended Data Fig. 7f-h). It should be (Fig. 3 i,j, Extended Data Fig. 7f-h).

We corrected this typo.

12.LMX1A is a transcription factor. Thus, the LMX1A staining signal should be in the nucleus. In Extended Data Fig. 9s. LMX1A signal in this image resembles the cytoskeleton. Please check the original images in Extended Data Fig. 9s.

We apologize for this mistake and thank the reviewer for spotting it- the staining in extended Data Fig. 9s is TAU and not LMX1A (LMX1a is already shown in panel q). We corrected this mistake.

13.Line 368-370: the author state that:

Moreover, the diameter of TH+ varicosities was significantly reduced in both striatal and cortical tissue indicating varicosities of reduced volume. At the same time, 25-day long withdrawal also failed to rescue the neuromorphological effects of cocaine. But in citation 61(Wildenberg, Sorokina et al. 2021):

The second obvious feature of DA axons exposed to cocaine was the occurrence of large axonal swellings or bulbs (Figure 7). The swellings were large (mean \pm SEM diameter: $2.2 \pm 0.3 \mu\text{m}$, $n = 23$), significantly larger than varicosities in control animals (mean \pm SEM diameter: $0.4 \pm 0.02 \mu\text{m}$, $n = 118$ varicosities) and at times reaching the size of neuronal soma (Figure 7A). These 'bulbs' were common in axons (~56%, 17/30 axons) in two cocaine-exposed animals, and we did not see a single example in DA axons from two control animals (0/29 axons), suggesting that Apex2 expression alone does not cause these swellings (Figure 7B; mean \pm SEM swellings/ μm length of axon: +saline, 0.00 ± 0.0 , $n = 29$ axons, two mice; +cocaine, 0.04 ± 0.02 , $n = 30$ axons, two mice. $p = 1.7\text{e-}5$).

Regarding this point, the exposure of MISCO to cocaine did not recapitulate the morphological /structural changes of TH+ varicosities in vivo. If this finding has support from other publications? Or does the author feel that MISCO is a hitherto model that brings a new perspective?

We thank the reviewer for this interesting question. Indeed, the morphological effects of cocaine are relatively poorly understood- and we understand both citations we provide (Wildenberg, Sorokina et al, 2021 as well as dos Santos et al, 2018) more as an indicator that morphological changes seem to be a consequence of cocaine exposure in humans as well. There are several important differences between the publication of Wildenberg, Sorokina et al. 2021 and our study- the use of another model system (mouse), the age difference (human development vs. mature mouse) as well as *in vitro* vs *in vivo*. Moreover, dos Santos and Caboche report an increase in the volume of dopaminergic varicosities in the NAc shell and core in cocaine treated mice, which correlates with our findings- but axonal swellings are not reported in this work.

Summarizing, we think that there is just not enough knowledge about the effects on cocaine on the morphology of dopaminergic neurons to make general conclusions, and there are inherent differences between the models which have been used and our *in vitro* model. We believe, however, that our finding might be interesting for translation into e.g. post mortem studies for further validation.

Citation

Bagley, J. A., D. Reumann, S. Bian, J. Lévi-Strauss and J. A. Knoblich (2017). "Fused cerebral organoids model interactions between brain regions." *Nature Methods* 14(7): 743-751.

Chen, X., H. Saiyin, Y. Liu, Y. Wang, X. Li, R. Ji and L. Ma (2022). "Human striatal organoids derived from pluripotent stem cells recapitulate striatal development and compartments." *PLoS Biology* 20(11): e3001868.

Onorati, M., V. Castiglioni, D. Biasci, E. Cesana, R. Menon, R. Vuono, F. Talpo, R. Laguna Goya, P. A. Lyons, G. P. Bulfamante, L. Muzio, G. Martino, M. Toselli, C. Farina, R. A. Barker, G. Biella and E. Cattaneo (2014). "Molecular and functional definition of the developing human striatum." *Nature Neuroscience* 17(12): 1804-1815.

Wildenberg, G., A. Sorokina, J. Koranda, A. Monical, C. Heer, M. Sheffield, X. Zhuang, D. McGehee and B. Kasthuri (2021). "Partial connectomes of labeled dopaminergic circuits reveal non-synaptic communication and axonal remodeling after exposure to cocaine." *eLife* 10.

Zhang, X., C. T. Huang, J. Chen, M. T. Pankratz, J. Xi, J. Li, Y. Yang, T. M. Lavaute, X.-J. Li, M. Ayala, G. I. Bondarenko, Z.-W. Du, Y. Jin, T. G. Golos and S.-C. Zhang (2010). "Pax6 is a human neuroectoderm cell fate determinant." *Cell Stem Cell* 7(1).

Zhu, Y., X. Zhang, L. Sun, Y. Wang and Y. Zhao (2023). "Engineering Human Brain Assembloids by Microfluidics." *Advanced Materials* (Deerfield Beach, Fla.): e2210083.

Reviewer #2:**Remarks to the Author:**

The manuscript by Reumann et al uses hiPSC-derived region-specific organoids to generate a model of the human dopaminergic system. By fusing separately-generated midbrain, striatum and cortical organoids they generate an assembloid that mimics some aspects of this system in vitro, and can be used to ask questions about transplantation and the effects of drugs. Overall, although the generation of the individual region-specific organoids is not novel, the assembly of these 3 regions together has not been done before and has important implications for research on human-specific aspects of the dopaminergic system. There are, however, a number of specific points that should be addressed:

1. Quantification, analysis and stats

1.1- The authors include the number of organoids used for experiments in the figure legends as a way of quantifying their results. However, in most cases it is unclear what they are quantifying. There is no section in the methods that specifies how the quantifications are done either. For example, in Figure 1e authors say “44-day old organoids display clusters of TH+ and FOX2A+ mDA neurons (n=8/8 organoids of 2 batches and 2 cell lines)”. The figure only shows one immunostaining of one organoid stained with TH and FOX2A. What is the quantification for? Are authors saying that 8 out of 8 organoids they looked at had TH+/FOX2A+ clusters? What do they consider a cluster for this quantification? A cluster could be anything from 3 cells to thousands of cells. And what percentage of the organoids do these clusters correspond to? This should be noted, otherwise the numbers shown have little meaning.

This is one example, but there are many instances of this throughout the manuscript. These

include: Fig 1d, 1e, 1f, 1g, 1i, 1j, 1k, 1l, 2c, 2e, 2f, 2g, 2h, 3a, 3l, 3m, 5b, 5c, 5e and Ext Data Fig 1c, 3i, 3j, 3k, 5g, 5h, 5i, 7a, 7b, 7d, 9u.

Please specify in each case what the quantification denoted in the figure legends for the panels above correspond to. This could be either in the figure legends or as a separate methods section.

We thank the reviewer for his comment. The images we chose to display are of representative nature, meaning if we state e.g. “n=28/32 organoids” this indicates that 28 of the 32 organoids had a comparable observation in comparison to the representative images. We updated the methods by an “Reproducibility” section in the methods, where we try to describe what n numbers mean, and added the description “representative images” to several panels.

We agree with the reviewer that the definition of a “cluster” can be heterogeneous, however, as we show representative examples, we certainly do not include aggregations of 3 cells but focus on similar morphological features to the examples we provide. To address the quantification aspect of this question, we performed additional quantification of TH+ regions in VM in extended data fig. 2 (VM) and extended data fig. 4 for striatal organoids. We believe that showing the different cell lines plus quantifications will solve the question regarding the definition of “clusters”. Alternatively, we suggest we could exchange the term “cluster” with “tissue” or “regions”, if the reviewer would recommend this.

1.2- In general, however, I am confused as to why these quantifications are not part of the figures. If these quantifications were performed, wouldn't they strengthen the manuscript if included in the form of graphs as part of the figures?

We thank the reviewer for this suggestion. We focused on showing quantifications which demonstrate the usability- and efficiency- of the protocol and MISCOs and were following the size limitations of Nature methods (6 main figures and 10 supplemental figures).

1.3- Quantification is missing for some experiments. For example:
- Fig 3e says "TH+ axons generally avoid neurogenic regions" - there is no quantification for this.

We thank the reviewer for this suggestion and added quantifications to the figure legend (n=17/17 rosettes of 8 organoids with ≤ 1 TH⁺ axon per neural rosette).

- Fig 5e says "Striatal tissue displayed denser innervation of grafted cells in comparison with cortical tissues..." - there is not quantification for this

We added quantifications for this statement (Extended Data Fig. 9v).

- In the text for Ext Data Fig 7e it says "...found that most forebrain TH+ cells expressed GFP..." - there is not quantification for this Either quantification should be performed or statements toned down

We toned down the statement ("To investigate their identity, we grew MISCOS with a DLX156-GFP reporter line³³ in the striatal organoids (CtX_{WT}-Str_{DLX156-GFP}-VM_{WT}) and found GFP and TH double positive cells, indicating TH⁺ interneuron or striatal TH⁺ neuron identity (Extended Data Fig. 7e)."

1.4- The authors perform ANOVA analysis in a few instances throughout the manuscript followed by t-tests. This is not standard practice. What is the reason for this? Usually ANOVA tests are followed by tests that correct for multiple comparisons (eg Bonferroni, Dunnett)

We thank the reviewer for his suggestion and updated analysis for more than two groups using Tukey's multiple comparisons test (recommended by Graphpad Prism based on the datasets which we provided).

1.5- P-values and/or details of statistical tests performed are missing in a few instances (see for example Fig 3c, 3g, 3h, 6d, 6e, 6g, 6h and Ext data Fig 3h, 9a-l)

We apologize for our oversight and updated the figure legends with exact p values and statistical tests used.

1.6- For each of the graphs in the manuscript, please state:
- what each of the points corresponds to. For example, in Fig 1n, is each dot an organoid? The legend says "n=8/8 organoids of 2 batches", yet there are only 4 points in the graph that I can see.
- how many sections per organoid are counted in each case (either in legend or in methods).
- how many lines are used per experiment (including RNAseq experiments)

We apologize for the mistake in n numbers and updated accordingly. We now address the number of sections per organoid (one per experiment) as well as the number of cell lines used in experiments in the section "Reproducibility" in the material & methods section.

1.7- While I appreciate that authors characterized their organoid recipes in different lines, I believe quantification (IHC or qPCR) would be needed to claim "We confirmed our findings in organoids from three different hiPSC and three different hESC lines, highlighting the robustness of the protocol." The data supporting this claim is limited to one immunostaining per line.

For ventral midbrain patterning, we indeed only analyzed one staining as the combination of FOXA2 and TH to our understanding is specific for ventral midbrain dopaminergic neurons and both antibodies performed robustly and specifically in our hands. For striatum, we performed more extensive analysis and show LGE patterning (GSX2+) as well as striatal medium spiny neuron identity (DARPP32, GABA and FOXP1 expression), which to our understanding is a specific set of markers to confirm the formation of striatal neurons. However, to address the concern of the reviewer we changed our statement ("We additionally confirmed the capability of this protocol

to produce VM mDAs/striatal neurons on three hiPSC and three hESC lines “) and additionally quantify the percentage of TH⁺ and DARPP32⁺ tissue (as well as % of rosettes expressing GSX2 in the striatal condition) per organoid.

1.8- There are some experiments that appear anecdotal as presented. For example:
- the authors mention they can recapitulate neuronal morphologies within the three brain regions, but I think that this would be stronger if it was paired with immunostainings for relevant markers of those cells.

We thank the reviewer for this suggestion. We tried to perform 3D IHC using celltype specific markers on Arch1-GFP transduced organoids, however we were not successful in performing 3D immunolabeling using cortical (TBR1) or striatal (DARPP32) markers- this is of course frustrating, but many antibodies which work in 2D do not work in 3D IHC. Furthermore, the procedure of 3D IHC disrupted the intricate morphologies of dendritic spines, which is the first message of this panel: we can see the formation of both pre- and postsynaptic structures, which is one of the milestones in neural maturation (formation of structural connections). The second message which we want to transport with this panel is that different neuronal morphologies in brain regions could be recapitulated using this methodology. We agree with the reviewer that the morphologies are in some context anecdotal, however we do not aim here to recapitulate the dozens of different morphologies of neurons which exist in the three different brain regions- we only want to propose that neural morphologies can potentially be studied in MISCOS. To clarify further, we update the description of these morphologies as “putative”.

- in Fig 5g the authors mention “putative axonal transport” with very little evidence (and no quantification). Some other indication that there is transport would be important

We removed this experiment from our manuscript.

2- Functional connectivity

2.1- The authors perform a rabies tracing experiment to show neuronal connectivity in MISCOs. However, the experiment as performed only allows them to show the presence of projections and not neuronal connectivity or functional projections (specifically they say “RV monosynaptic tracing indicated potentially functional mDA long-range projections”). The authors show the presence of mDA neuronal projections from the VM organoid to the striatal and cortical organoids. Injection of the helper virus into the cortical side will transduce mDA projections as well as cortical cells allowing the rabies virus to infect mDA cells directly. In order to claim spread of the rabies virus authors should perform the transduction before assembly.

We thank the reviewer for this comment. We do show that the transduction of starter cells is specific to the site of injection (figure 3l, m, extended data fig. 7j, 7i) and there are no GFP⁺ cells at the site of spread (in the VM), thus we believe that the conclusion which we make is valid. We want to emphasize here that the amount of helper virus which was injected was finely tuned to only transduce cells at the site of injection. We are aware of the discussion in the field of rabies tracing whether rabies virus can only be transmitted through functional synapses, or if structural synapses are sufficient- which is why we state “potentially functional mDA long-range projections”. The question if these long-range connections are functional is then addressed in the next figure (figure 4).

We furthermore want to emphasize that viral transduction before assembly would not be possible to our knowledge: first, on day 20-25, almost none of the neurons which we want to investigate are produced yet (at this stage, brain organoids are almost exclusively comprised of progenitors). Second, for functional analysis we investigate organoids after day 120. As AAVs are transient, we wonder if expression levels would still be sufficient >100 days after transduction.

2.2- Similarly, the authors use an AAV to transduce Chr2 into MISCOs but they do so after assembly, therefore all cells express it. Even if the authors use a targeted approach to stimulate

only VM tissue, Ext Data Fig 7 shows there are reciprocal projections (ie projections from the striatum and cortex to VM). Therefore, light stimulation in VM would also be activating striatal and cortical cells that received the AAV. Based on this experiment the authors cannot say "Together, this data confirms the ability of MISCOs to develop functional neural networks across all regions". If the authors want to claim connectivity between the different organoids they would need to either transduce organoids before assembly or use a soma-restricted Chr2.

We thank the reviewer for this suggestion. With the experiments we performed, antidromic action potentials can indeed not be excluded, however we initially speculated that the light intensity needed for such stimulation must be higher than the threshold for cell bodies.

Because organoid transductions before assembly were not possible given the reasons outlined in response to **reviewer comment 2.1**, we resorted to the use of a soma-restricted Chr2, AAV pAAV9-hSynapsin-soCoChR-GFP (Addgene Cat.# 107708-AAV9). Unfortunately, the serotype performed poorly with low cell transduction and GFP expression more than 2 weeks after transduction (**Reviewer Figure 1h, i**). As a result light stimulation were unable to illicit sufficient action potentials (**Reviewer Figure 1j-k**). When we extracted multi-unit spikes from extracellular silicon-probe recordings +/- 100 seconds from the start of optogenetic stimulation (**see reviewer figure 1e and 1f**), we detected no firing rate changes, or network burst responses following optogenetic stimulation parameters matched with previous experiments (see Figure 4 a-h).

Therefore, to address this comment, we performed a new set of optogenetic experiments while broadly blocking synaptic transmission with a cocktail of D-AP5, CNQX, Gabazine, SCH-23390 and Sulpiride ^{-/-}. We probed functional connectivity in 2 MISCOs by optogenetically stimulating ventral midbrain tissue while recording extracellular signals in forebrain tissue. Synaptic blockers were added to ACSF and perfused into a bath recording chamber for 5-minutes, before repeating the optogenetic stimulation. We found that the synaptic antagonist cocktail abolished the typical forebrain multi-unit firing rates in synchrony with light stimulations of VM (Extended Data Figure 8m, n). This data suggest that network activity during optogenetic stimulus is mediated by the synaptic transmission of functional long-range connections between fused regions.

2.3- Again, authors transduce all regions in MISCOs to show striatal and cortical cells receive dopamine. Even if authors are imaging the presumed striatal and cortical regions, it would be important to at least show some immunostainings to show that the cells that receive the AAV-GRAB-DA2 in the cortex and striatum are the cells they believe.

We performed the suggested immunostainings which show that the AAV-GRAB-DA2 (GFP+) transduces cortical (TBR1+) and striatal (DARPP32+) neurons and display the stainings in extended data fig. 8i and j.

2.4- It would be useful to see some examples of activity-dependent and dopamine-related genes upregulated after assembly in the main text (in paragraph starting on line 314) We added FOS and BDNF as examples of activity-regulated genes to the main text.

2.5- Authors say that Ext Data Fig 8d, e “allows them to study calcium events spreading through VM axons into forebrain tissues”. However, I am struggling to see this from the figure provided. It is not clear what the images show and what fluorescence is quantified. Please provide more details. There is also no quantification in this panel.

We updated the panel to have zoomed in images of calcium events in VM derived axons. We also added a description to panel e where the corresponding images (d' and d'') have been extracted from and updated the text to describe which fluorescence is shown in panel e. We also performed more recordings which generally displayed VM derived calcium events in forebrain tissue and added these numbers to the figure legend.

3. Effects of cocaine on dopaminergic system

3.1- The authors mention “not much is known about the effects of cocaine on the development of the CNS – predominantly because of the lack of an effective model system”. What do they mean by effective? There are mouse and rat models used to study in utero exposure to cocaine

We thank the reviewer for his comment. While it is true there are in utero models for cocaine exposure, there are significant differences between a) the brain regions involved in the dopaminergic circuits (where the striatum and cortex are arguably the most different to mouse and rat), and b) the timespan in which neurogenesis and circuit formation happens in these models versus in humans. Dopaminergic neurons are born within days in mice, whereas the time window of dopaminergic neurogenesis (and circuit formation) in humans spans months. This opens up a large time window in which the establishment of circuits can be impaired, and to date we are not aware of a model system which could model this slow and sensitive process- besides using primates.

We think that MISCOS could be system which can allow to study exactly this question, and we provide indications that we can study the temporal aspects of dopaminergic circuit formation such as by the example of increase of dopaminergic varicosities over the timespan of 120 days (Figure 3g, h). To clarify and be more specific with our statement, we now state that “...not much is known about the effects of cocaine on the development of the human CNS- predominantly because of the lack of an effective **human** model system.”

3.2- The authors say “we used 0.7uM...a concentration well within the physiological range in humans” – what is the physiological range in humans? Please state

The concentration which we use is in the range of IC50 of DAT, for which we found different values ranging from 0.177µM (Aggarwal et al., Biomedicines, 2021) to 0.7µM (Woodward et al., European Journal of Pharmacology 1995). The physiological range in humans can range,

dependent on the frequency of use, however Zheng&Zhan (Plos Computational Biology, 2012) discuss the concentration of cocaine in the brain to reach levels of 2.5 μ Molar. To give another calculation, based on Bravo et al. (Toxins, 2022) a typical dose ("line") of cocaine would contain 50mg cocaine hydrochloride. For a person of 70kg, this would result in a theoretical peak of 2.1 μ M. However, this peak concentration would only occur for a short period of time: the elimination half life of cocaine is between 0.7-1.5h). Cocaine is mostly metabolized by plasma esterases and liver cholinesterases, which are both most likely not present in brain organoids, which would result in relatively stable cocaine levels in the media. Thus, we decided to use the approximate IC50 of cocaine hydrochloride for 1h instead. We decided to not add such a number to the main manuscript as we think the statement of a single number could be misleading- the concentration of cocaine in blood vs. brain is different and based on mostly hypothetical calculations- and undergoes rapid changes in vivo. However, to clarify this issue we added a section in the "Pharmacology" tab of the methods, explaining our calculations in more detail.

3.3- I am confused about the duration of calcium events in Fig 6g. If I understand correctly, each calcium event lasts ~5 seconds? This is not standard for neuronal calcium events using Gcamp. If the authors are using a constitutive Gcamp line to perform these experiments then they could be recording events in progenitors. Treatment with TTX could help establish if events are neuronal (which is important based on the claims made)

We thank the reviewer for this question. We use a neural-specific GCAMP line (Syn-GCAMP), however to additionally confirm that we record neurons we performed a TTX experiment which we added into Extended Data Fig. 10 b and which demonstrates that the majority of activity we record is neuron-derived. However, we want to state here additional potential reasons for the extended GCAMP events. Often, GCAMP activity duration is measured by extracting the width at half height. However, as many neurons in MISCOs were active for an extended period of time - sometimes being continuously active for up to 20 seconds- this function caused quite some issues for us by not catching all events, or splitting one event into many events, which is why we started

using the width at 5% of max intensity (as indicated in Figure 6f). We speculate that this activity could be neurons which are active over a longer period of time (GCAMP6s is too slow to resolve such events individually), or immature neurons. However, as they are a feature of neurodevelopment (and MISCOS) at this stage we did not want to exclude them.

To make the way we measure GCAMP activity clearer, we added a description of how we measured calcium event duration in the methods.

3.4- How are varicosities quantified/detected? Is it just based on morphology? Are they specific to TH cells? How do we know they are not just unhealthy beading axons? Because of this, it would be important to show varicosities in another way other than morphology (eg. DAT colocalization)

Varicosities were identified based on their morphological features (TH⁺ enlargements on axons) which is common in the literature (e.g. Ducrot et al., 2021, Dos Santos et al., 2018). Varicosities (or boutons- in the literature, both terms are used for dopaminergic neurons) are not specific to dopaminergic neurons and can be found generally on axons. We used the term varicosity as it includes unspecific release (thus, not into a synaptic terminal) which dopaminergic neurons do- whereas axonal boutons are usually the presynaptic site of a synapse. To clarify that we quantify TH immunolabeled axons, we changed the figure legend of figure 6b and changed the contrast of "TH" labeling in the panel.

We show TH and DAT colocalization in extended data Fig. 5i.

Minor comments

- There is a typo on line 80 (on are on their way)

We corrected this typo.

- There is a typo on line 96 (stratal)

We corrected this typo.

- Genes in lines 162 and 164 should be in italics

We corrected this typo.

- The term "surface recordings" on line 331 is confusing. Do they mean imaging? The term recording is usually used to mean electrophysiological recordings (see also figure legend for that figure)

We exchanged the term surface recording with whole mount microscopic recording.

Reviewer #3:**Remarks to the Author:**

The manuscript entitled, “In vitro modeling of the human dopaminergic system using spatially arranged ventral midbrain-striatum-cortex assembloids” describes the establishment of a protocol to generate a complex human neural tissue (here termed MISCO) that recreates aspects of the human dopaminergic system through fusing 3 distinct regional tissues (ventral midbrain, VM; striatum, and cortex). The authors present data supporting protocol development, and then apply the MISCO assembloid to test cell engraftment related to parkinson’s disease therapy and study the response to cocaine stimulation. Altogether, this is a very interesting paper that uses diverse methods to characterize the system, and study the functional activities within the assembloid under different conditions. I have several questions/concerns that are related to the specificity and robustness of the Method.

As the first step to generate the three-region fusion, the authors describe modulation of previous protocols to generate the three regions prior to fusion. There have been previous papers that describe the generation of midbrain, striatum, and cortex from human pluripotent stem cells. The authors should provide more details and explanation about what is distinct between previous protocols and the protocol described here. Also, the authors need to ensure that all previous protocols are referenced in the manuscript.

We thank the reviewer for the interest in our work! We tried to cite the papers of protocols which were first in their field, resulted in relevant improvements to the method, or were used as a reference for our work, however given the growth of the field citing all previously published protocols would be above the citation limit provided by Nature Methods. However, if the reviewer believes that important protocols have been overseen and should be added, we would be very happy to include them and apologize for our oversight!

Concerning the VM protocol, the authors state that “we found that a concentration of 300nM SAG from day 4-11, together with dual SMAD inhibition and Wnt activation, was sufficient to introduce maximal FOXA2 expression levels by day 20 (Fig. 1b, d Extended Data Fig. 1a).” Did the authors look beyond day 20 at organoids treated with the different concentrations of SAG? Also, did the authors test treatment for other days before or beyond day 4-11? Finally, did the authors test different initiating sizes of the organoids prior to SAG treatment (e.g. using fewer or more cells to start the culture)? Again, how is this protocol different from previously published ventral midbrain organoid protocols?

We thank the reviewer for these interesting questions. As the main purpose of this experiment was to replace the morphogen SHH (for which we had significant batch to batch variability in the past) with the small molecule SAG, we particularly looked at its ventralization effect for which we used FOXA2 as a readout. We then selected the condition which had satisfactory floor plate induction (300nM SAG) and continued the characterization as described (with testing of multiple cell lines up to day 80, see extended data figure 2).

We did not test different timepoints for SAG as the duration gave us a very high yield in FOXA2⁺ floor plate progenitors- which was the aim of this experiment.

We did also not test different initiating sizes of organoids prior to SAG treatment, as the amount of cells (9000) is a standard which was used in previous work and results in very nice EB formation. However, we want to add here that we speculate that too many cells as starting population might increase unwanted tissue heterogeneity (e.g. by limited diffusion of supplied morphogens and small molecules).

Summarizing, our VM protocol is different by the replacement of the morphogen SHH with the small molecule SAG and by using a basal media (neural induction media) which contains no additional growth factors- however, similar compositions of neural induction media can be found in other protocols.

The authors present bulk RNA-seq data on each of the three tissues, and then compare this data to primary reference datasets showing general similarity to the regions of interest. The authors then use single-cell RNA-seq to characterize cell heterogeneity in the day 60 assembloid by first separating the major regions prior to sequencing. The data is presented in Figure 2j-l, and ED Fig. 6. How many assembloids were used to generate the data? Did the authors generate scRNA-seq data from multiple batches? In addition to the selected feature plots in the ED, the authors should include a heatmap showing marker genes (e.g. 20) for each cluster, as well as a supporting table (apologies if this is included already). It would be helpful if the authors could show the UMAP plot in ED 6b with each condition in separate plots, as well as a stacked bar plot showing the proportion of cells per per cluster broken down by condition. It is difficult to determine if each cluster has contributions from all conditions.

We performed scRNAseq on cells from three MISCOS, however we did not perform scRNAseq on multiple batches of organoids. We added the number of MISCOS to the figure legend.

We thank the reviewer for the suggestion of including marker genes per cluster, which we added (see extended data fig. 6h) and we also added the cells per cluster per condition (extended data fig. 6c).

The authors present a device that can be used for assisting in linear organoid fusion. The authors state, "In consecutive batches, we achieved 96% ($\pm 2.4\%$ SD) fusion efficiency (Fig. 2d)." This quantification appears to be based on data from 3 days after fusion. What is the percentage at a later time point (e.g. 21, 33, 109 as in the images)? It would be helpful if the authors could have an schematic in Figure 2 that shows the time point of fusion and the media composition in the assembloid culture, similar to what is shown in Fig. 1b, as the particular details are likely relevant for the reproducibility of the Method.

We thank the reviewer for his recommendations. For fusion of MISCOS, the first 3 days were the only critical time window: We saw separation of tissue at the timepoint of transfer from embedding mold into 10cm plates, and subsequent breaking events while handling the plates in the first three days. After 3 days, tissue was generally fused well enough that MISCOS generally did not break apart anymore. We added an explaining sentence to the text.

We also thank the reviewer for the suggestion for a schematic. We created a schematic for the generation of MISCOS and added it as panel Extended Data Fig. 5a.

Figure 3 shows an assembloid that has very different sized VM, striatum, and cortex regions. It would be very helpful to **describe the variation in assembloid overall size**, as well as the size of the individual regions. This information will be very helpful to other researchers that may try to reproduce the method.

We thank the reviewer for his suggestion and added the quantification for individual regions as well as MISCOS into extended figure 1d and e.

Reviewer Figure 1

Reviewer figure 1

a, Correlation of striatal and cortical bulkRNAseq (day 60) with the BrainSpan dataset of the developing human brain (PCW20-25). STR...Striatum, NCx...Neocortex, HIP...Hippocampus, DTH...Dorsal Thalamus, CB...Cerebellum, AMY...Amygdala. b, Chimeric MISCOs with CAG-GFP line in striatal tissue. c, Schematic illustrating optogenetic stimulation of VM tissue and extracellular recording for the striatal tissue. Organoids were transduced with the optogenetic construct AAV-RG.AAV-CAG-hChR2H134R-tdtomato. d, Representative active channel raster plots in striatal tissue +/- 5 minutes from the initiation of optogenetic stimulation (orange box). 460-nm LED light

pulses set at an interval of 10 seconds with a 500-millisecond duration (blue). A 20-second interval of the recordings is shown to highlight optogenetic neural population responses (right). Same recording as in manuscript figure 4b. **e**, Schematic illustrating optogenetic stimulation of VM tissue and extracellular recording from the cortical tissue, following the same experimental setup as in (c). **f**, Representative active channel raster plots in cortical tissue \pm 5 minutes from the initiation of optogenetic stimulation (orange box). Same recording as in manuscript figure 4f. **g**, an extracellular signal extracted from a single channel shown with its corresponding absolute band-passed filtered values (left). An example extracellular waveform from a spike detected 5 standard deviations above the noise floor (right). Detected spikes are shown in blue. **h**, Confocal recording of MISCOS with AAV-hChr2-tdtomato 2.5 weeks after transduction. **i**, Confocal recording of MISCOS with AAV-soCoChR-GFP 2.5 weeks after transduction. **j**, Schematic of the stimulation setup for testing functionality of scCoChR-GFP transduced organoids. Activity was recorded in the ventral midbrain and stimulation was performed on the site of recording. **k**, Active channel raster plots \pm 100sec from initiation of optogenetic stimulation (orange box) of VM recordings with VM stimulation after baseline recording in 178 day old MISCOS (blue dashes, same setup as in Manuscript Main Figure 4a-b and e-f). No stimulation dependent population responses were detected.

Decision Letter, first revision:

Dear Juergen,

Thank you for submitting your revised manuscript "In vitro modeling of the human dopaminergic system using spatially arranged ventral midbrain-striatum-cortex assembloids" (NMETH-A51344A). It has now been seen by the original referees and their comments are below. The reviewers find that the paper has improved in revision, and therefore we'll be happy in principle to publish it in Nature Methods, pending minor revisions to satisfy the referees' final requests and to comply with our editorial and formatting guidelines.

I recommend that you add some discussion on why the organoids were not tested with SAG for longer than 20 days (like in your response to Ref 3).

TRANSPARENT PEER REVIEW

Nature Methods offers a transparent peer review option for new original research manuscripts submitted from 17th February 2021. We encourage increased transparency in peer review by publishing the reviewer comments, author rebuttal letters and editorial decision letters if the authors agree. Such peer review material is made available as a supplementary peer review file. Please state in the cover letter 'I wish to participate in transparent peer review' if you want to opt in, or 'I do not wish to participate in transparent peer review' if you don't. Failure to state your preference will result in delays in accepting your manuscript for publication.

Please note: we allow redactions to authors' rebuttal and reviewer comments in the interest of confidentiality. If you are concerned about the release of confidential data, please let us know specifically what information you would like to have removed. Please note that we cannot incorporate redactions for any other reasons. Reviewer names will be published in the peer review files if the reviewer signed the comments to authors, or if reviewers explicitly agree to release their name. For more information, please refer to our <https://www.nature.com/documents/nr-transparent-peer-review.pdf> target="new">FAQ page.

ORCID

IMPORTANT: Non-corresponding authors do not have to link their ORCIDs but are encouraged to do so. Please note that it will not be possible to add/modify ORCIDs at proof. Thus, please let your co-authors

know that if they wish to have their ORCID added to the paper they must follow the procedure described in the following link prior to acceptance:

Sincerely,
Madhura

Madhura Mukhopadhyay, PhD
Senior Editor
Nature Methods

Reviewer #1 (Remarks to the Author):

Dear author and editor:

The reviewer greatly appreciates the efforts of the authors to address the reviewer's main and minor concerns about this manuscript. The revision and response letter almost/fully address the reviewer's concerns. A beautiful work. Congrats!

We have two minor concerns :

1. In extended Fig1C of the revised version, is the label of the X-axis right?
2. To the reviewer's knowledge, the post-injection days differ from the time of organoid in culture. Thus, the reviewer suggests that the post-injection days and the time of organoid in culture should label separately in extended Fig9w of the revised version.

Reviewer #2 (Remarks to the Author):

I thank the authors for their thorough responses, and I appreciate the efforts to address all the concerns. However, there are a few things that I believe still need to be changed.

- Quantification: my main concern is not with the word "cluster", but instead with the subjective nature of all the reported quantification throughout the manuscript. An appropriate criterion for quantification in this particular case would be, for example: "20 out of 20 organoids from 5 individual lines had >80% FOXA2 cells (either over total DAPI or area)". Please either specifically say what is being quantified in

each panel, or change the wording to indicate that quantifications are subjective indications. For example, in figure 1d instead of saying “n=20/20 organoids of 5 cell lines” say “similar results were observed in 20 organoids derived from 5 cell lines”. Please do this throughout manuscript for all panels with representative images that are lacking quantification.

- Rabies tracing experiment: I see that there is a picture in the supplement showing the absence of helper GFP+ cells in the VM. However, I could not find quantification for this or see it mentioned in the text. Please include quantification. I believe this is an important point if claiming connectivity.

- Please change the title for Figure 3. I don't believe the data is showing evidence of “structurally mature neural circuits”. And in general, it would be a good idea to also tone down the "neural circuit" and "maturity" claims in parts of the manuscript (for example: "This has opened new avenues for studying the dopaminergic system in vitro, including morphological and functional maturation of both neurons and formation of neural circuits"). It is unclear what the authors mean by maturity. Neuronal maturity would have to be shown with electrophysiological recordings and in comparison to other mature cells.

Author Rebuttal, first revision:

General response to all reviewers

We would like to thank the reviewers for their feedback. We have followed their recommendations and corrected the points they raised (see also point-by-point responses below).

Point-to-point response to reviewer's comments

Reviewer #1:

Remarks to the Author:

Dear author and editor:

The reviewer greatly appreciates the efforts of the authors to address the reviewer's main and minor concerns about this manuscript. The revision and response letter almost/fully address the reviewer's concerns. A beautiful work. Congrats!

We have two minor concerns :

1. In extended Fig1C of the revised version, is the label of the X-axis right?
2. To the reviewer's knowledge, the post-injection days differ from the time of organoid in culture. Thus, the reviewer suggests that the post-injection days and the time of organoid in culture should label separately in extended Fig9w of the revised version.

Response of the authors:

We thank the reviewer for this positive feedback and are happy to hear that they share the excitement of our work!

We updated the labeling of Fig1c and added the age of injected MISCOs into the figure legend of Fig9W.

Reviewer #2:

Remarks to the Author:

I thank the authors for their thorough responses, and I appreciate the efforts to address all the concerns. However, there are a few things that I believe still need to be changed.

- Quantification: my main concern is not with the word “cluster”, but instead with the subjective nature of all the reported quantification throughout the manuscript. An appropriate criterion for quantification in this particular case would be, for example: “20 out of 20 organoids from 5 individual lines had >80% FOXA2 cells (either over total DAPI or area)”. Please either specifically say what is being quantified in each panel, or change the wording to indicate that quantifications are subjective indications. For example, in figure 1d instead of saying “n=20/20 organoids of 5 cell lines” say “similar results were observed in 20 organoids derived from 5 cell lines”. Please do this throughout manuscript for all panels with representative images that are lacking quantification.

We thank the reviewer for this comment. We adapted the figure legends and now always mention directly with provided n numbers that the presented images are representatives and that similar results were observed in the provided n numbers. For example, we rephrased the cited figure legend (Figure 1d) to “representative image, similar results in n=20/20 organoids of 5 cell lines”.

- Rabies tracing experiment: I see that there is a picture in the supplement showing the absence of helper GFP+ cells in the VM. However, I could not find quantification for this or see it mentioned in the text. Please include quantification. I believe this is an important point if claiming connectivity.

We thank the reviewer for this comment and now provide quantification for this in extended data figure 7k.

Please change the title for Figure 3. I don't believe the data is showing evidence of "structurally mature neural circuits". And in general, it would be a good idea to also tone down the "neural circuit" and "maturity" claims in parts of the manuscript (for example: "This has opened new avenues for studying the dopaminergic system in vitro, including morphological and functional maturation of both neurons and formation of neural circuits"). It is unclear what the authors mean by maturity. Neuronal maturity would have to be shown with electrophysiological recordings and in comparison to other mature cells.

We updated the title for figure 3 to "MISCOs form structural features of maturation" and adapted the text where applicable. We agree with the reviewer that stating the presence of "mature" neurons would be problematic and we apologize for this unclear phrasing. However, in our manuscript we particularly focus on the process of neural maturation (thus, the development of an immature neuron to a – hypothetical- mature neuron). In this developmental trajectory, neurons will first migrate, then build out morphological features (axonal and dendrite outgrowth), then form structural connections with target neurons (by the formation of pre (dendritic spine)- and postsynaptic (axonal boutons/varicosities) terminals) and then start to become functionally active and interact with target neurons. Of course, just because a neuron is functionally active, it does not mean it has reached a fully mature state yet (which, in humans, might take years), and its features and activity might change drastically in its maturation trajectory. We went through the manuscript and edited, where we think we were not clear enough about referring to the process of maturation, and where it could have been confused with referencing to a mature state.

Reviewer #3:

None

Final Decision Letter:

Dear Jürgen,

I am pleased to inform you that your Article, "In vitro Modeling of the Human Dopaminergic System using Spatially Arranged ventral Midbrain-Striatum-Cortex Assembloids", has now been accepted for publication in Nature Methods. Your paper is tentatively scheduled for publication in our December print issue, and will be published online prior to that. The received and accepted dates will be 30th Dec, 2022 and 10th Oct, 2023. This note is intended to let you know what to expect from us over the next month or so, and to let you know where to address any further questions.

Over the next few weeks, your paper will be copyedited to ensure that it conforms to Nature Methods style. Once your paper is typeset, you will receive an email with a link to choose the appropriate publishing options for your paper and our Author Services team will be in touch regarding any additional information that may be required.

You will receive a link to your electronic proof via email with a request to make any corrections within 48 hours. If, when you receive your proof, you cannot meet this deadline, please inform us at rjsproduction@springernature.com immediately.

Please note that *Nature Methods* is a Transformative Journal (TJ). Authors may publish their research with us through the traditional subscription access route or make their paper immediately open access through payment of an article-processing charge (APC). Authors will not be required to make a final decision about access to their article until it has been accepted. [Find out more about Transformative Journals](https://www.springernature.com/gp/open-research/transformative-journals)

Your paper will now be copyedited to ensure that it conforms to Nature Methods style. Once proofs are generated, they will be sent to you electronically and you will be asked to send a corrected version within 24 hours. It is extremely important that you let us know now whether you will be difficult to

contact over the next month. If this is the case, we ask that you send us the contact information (email, phone and fax) of someone who will be able to check the proofs and deal with any last-minute problems.

If, when you receive your proof, you cannot meet the deadline, please inform us at rjsproduction@springernature.com immediately.

Once your manuscript is typeset and you have completed the appropriate grant of rights, you will receive a link to your electronic proof via email with a request to make any corrections within 48 hours. If, when you receive your proof, you cannot meet this deadline, please inform us at rjsproduction@springernature.com immediately.

Once your paper has been scheduled for online publication, the Nature press office will be in touch to confirm the details.

Once your paper has been scheduled for online publication, the Nature press office will be in touch to confirm the details.

Content is published online weekly on Mondays and Thursdays, and the embargo is set at 16:00 London time (GMT)/11:00 am US Eastern time (EST) on the day of publication. If you need to know the exact publication date or when the news embargo will be lifted, please contact our press office after you have submitted your proof corrections. Now is the time to inform your Public Relations or Press Office about your paper, as they might be interested in promoting its publication. This will allow them time to prepare an accurate and satisfactory press release. Include your manuscript tracking number NMETH-A51344B and the name of the journal, which they will need when they contact our office.

About one week before your paper is published online, we shall be distributing a press release to news organizations worldwide, which may include details of your work. We are happy for your institution or funding agency to prepare its own press release, but it must mention the embargo date and Nature Methods. Our Press Office will contact you closer to the time of publication, but if you or your Press Office have any inquiries in the meantime, please contact press@nature.com.

To assist our authors in disseminating their research to the broader community, our SharedIt initiative provides you with a unique shareable link that will allow anyone (with or without a subscription) to read

the published article. Recipients of the link with a subscription will also be able to download and print the PDF.

Nature Portfolio journals [encourage authors to share their step-by-step experimental protocols](https://www.nature.com/nature-research/editorial-policies/reporting-standards#protocols) on a protocol sharing platform of their choice. Nature Portfolio 's Protocol Exchange is a free-to-use and open resource for protocols; protocols deposited in Protocol Exchange are citable and can be linked from the published article. More details can found at www.nature.com/protocolexchange/about.

Best regards,
Madhura

Madhura Mukhopadhyay, PhD
Senior Editor
Nature Methods